# Variations of global and continental water balance components as impacted by climate forcing uncertainty and human water use

**H. Müller Schmied[1,2], L. Adam[1], S. Eisner[3], G. Fink[3], M. Flörke[3], H. Kim[4], T. Oki[4], F. T. Portmann[1], R. Reinecke[1], C. Riedel[1], Q. Song[1], J. Zhang[1] and P. Döll[1]**

[1]{Institute of Physical Geography, Goethe-University Frankfurt, Germany}

[2]{Senckenberg Biodiversity and Climate Research Centre (BiK-F), Frankfurt, Germany}

[3]{Center for Environmental Systems Research (CESR), University of Kassel, Germany}

[4]{Institute of Industrial Science, The University of Tokyo, Tokyo, Japan}

Correspondence to: H. Müller Schmied (hannes.mueller.schmied@em.uni-frankfurt.de)

## Abstract

When assessing global water resources with hydrological models, it is essential to know about methodological uncertainties. The values of simulated water balance components may vary due to different spatial and temporal aggregations, reference periods and applied climate forcings and due to the consideration of human water use, or the lack thereof. We analyzed these variations over the period 1901-2010 by forcing the global hydrological model WaterGAP 2.2 (ISI-MIP 2.1) with five state-of-the-art climate data sets, including a homogenized version of the concatenated WFD/WFDEI data set. Absolute values and temporal variations of global water balance components are strongly affected by the uncertainty in the climate forcing, and no temporal trends of the global water balance components are detected for the four homogeneous climate forcings considered (except for human water abstractions). The calibration of WaterGAP against observed long-term average river discharge Q significantly reduces the impact of climate forcing uncertainty on estimated Q and renewable water resources. For the homogeneous forcings, Q of the calibrated and non-calibrated regions of the globe varies by 1.6% and 18.5%, respectively, for 1971-2000. On continental scale, most differences for P and Q estimates occur in Africa and, due to snow undercatch of rain gauges, also in the data-rich continents Europe and North America. Variations of Q at the grid-cell scale are large except in a few grid cells upstream and downstream of calibration stations, with an average variation of 37% and 74% among the four homogeneous forcings in calibrated and non-calibrated regions, respectively. Considering only the forcings GSWP3 and WFDEI_hom, i.e. excluding the forcing without undercatch correction (PGFv2.1) and the one with a much lower SWD than the others (WFD), Q variations are reduced to 16% and 31% in calibrated and non-calibrated regions, respectively. These simulation results support the need for extended Q measurements and data sharing for better constraining global water balance assessments. Over the 20$^{th}$ century, the human footprint on natural water resources has become larger. On 11–18% of the global land area, change of Q between 1941-1970 and 1971-2000 was driven more strongly by change of human water use including dam construction than by change in precipitation, while this was true on only 9-13% of the land area from 1911-1940 to 1941-1970.

## 1 Introduction

Assessment of global-scale water resources and water balance components is of importance for water resources management at global, continental and river basin scales (Vörösmarty et al. 2015). Many data-based, model-based, and hybrid approaches exist in order to quantify macro-scale water balance components (Baumgartner and Reichel, 1975; Fekete et al., 2002; Haddeland et al., 2011; Müller Schmied et al., 2014; Oki and Kanae, 2006). For water resources management, especially the estimation of renewable freshwater resources (long-term average runoff or river discharge) is of importance, as it is the source for both human and ecosystem needs. As adequate discharge observations are available only at selected locations (see the catalogue of the Global Runoff Data Centre (GRDC), http://grdc.bafg.de/), model-based or hybrid (i.e. incorporating historical discharge observations) approaches to estimate discharge and other water balance components are of increasing importance. Since the 1980s, global hydrological models (GHMs) have been developed to calculate the water balance on global and/or continental scales. Recent reviews of such models are presented by Bierkens (2015), Sood and Smakhtin (2015) and Trambauer et al. (2013).

All GHMs are driven by climate forcing input data sets (hereafter called climate forcings), based on station observations (e.g. for precipitation and air temperature), reanalysis (global circulation models for numerical weather prediction which assimilate all available up-to-date data for current time step) and/or remote sensing data (e.g. for radiation). Within the last two decades, numerous climate forcings were developed with a current standard of at least daily temporal resolution and 0.5° by 0.5° spatial resolution (the common GHM spatial resolution), providing data from as early as 1901 until recent years. These climate forcings differ among each other and thus may lead to different water resources estimates by GHMs.

Humans have altered the global water cycle with an increasing intensity, e.g. due to irrigation or industrial water use (Döll and Siebert, 2002; Döll et al., 2012; Flörke et al., 2013; Siebert et al., 2015; Wada et al., 2010). A number of GHMs (but not all) are able to incorporate human water use in their calculations (see Table 2 in Bierkens, 2015). Neglecting anthropogenic water consumption prevents meaningful water resources assessments, at least in regions with high water consumption relative to renewable resources (e.g. High Plains Aquifer, Indus, Ganges-Brahmaputra). For example, groundwater depletion as observed by falling groundwater heads in wells and by GRACE satellite observations of gravity variations can only be modelled when human water use is considered (Döll et al., 2014).

Simulated water balance components vary considerably due to various uncertainties of GHMs (Haddeland et al., 2011; Schewe et al., 2014) including GHM ability to include human water use, model improvements over time (e.g. see the different results of the Water Global Assessment and Prognosis (WaterGAP) model in Müller Schmied et al. (2014), their Table 5), climate forcing (Biemans et al., 2009; Voisin et al., 2008) as well as uncertainties in discharge observations (Coxon et al., 2015; McMillan et al., 2012). In addition to these uncertainties, water resources estimates differ due to different reference periods (Wisser et al., 2010).

This study contributes to the assessment of water balance components on a global and continental scale by answering the following research questions.

1. What is the impact of climate forcing uncertainty on water balance components at global, continental and grid-cell scale?
2. What is the variation of estimated global water balance components for different temporal aggregations: year, decade, 30 years, and century?
3. What determines variations of long-term average river discharge between consecutive 30-year periods more strongly – either change of precipitation or change of human water use and dam construction creating reservoirs and regulated lakes ("anthropogenic impact")?

To answer these questions, we conducted a modeling experiment. The model, data and methods are described in Sect. 2. Results are presented and discussed in Sect. 3. Finally, conclusions are drawn and an outlook is given.

## 2 Data and methods

In this study, the global water availability and water use model WaterGAP (Alcamo et al., 2003; Döll et al., 2003) was applied in a modified version of WaterGAP 2.2 (Müller Schmied et al., 2014) in two water use and management variants (including and excluding anthropogenic effects). The model was driven by four state-of-the-art climate forcings provided by the Inter-Sectoral Impact Model Intercomparison Project (ISI-MIP) in its current phase 2.1 (https://www.pik-potsdam.de/research/climate-impacts-and-vulnerabilities/research/rd2-cross-cutting-activities/isi-mip/about/isi-mip2) and a fifth homogenized forcing.

## 2.1 GHM WaterGAP 2.2 (ISI-MIP 2.1)

The spatial resolution of WaterGAP is 0.5° by 0.5° (approx. 55 km by 55 km at the equator), and the model uses daily time steps for calculation. The WaterGAP water use models compute water use estimates for five sectors (irrigation, domestic, manufacturing, cooling water for electricity generation, and livestock) that are processed by the Ground Water Surface Water Use (GWSWUSE) submodule to quantify both net water abstractions from surface water and from groundwater (Fig. 1 in Müller Schmied et al., 2014). Taking into account the net abstractions, the WaterGAP Global Hydrology Model (WGHM) calculates changes in water storage compartments as well as water flows between these compartments based on water balance equations, including groundwater recharge, evapotranspiration and river discharge. A description of model version WaterGAP 2.2 can be found in Müller Schmied et al. (2014). The version used for this study is named "WaterGAP 2.2 (ISI-MIP 2.1)", and differences to WaterGAP 2.2 mainly consider requirements of the ISI-MIP project phase 2.1 as described in Appendix A.

## 2.2 Calibration of WaterGAP 2.2 (ISI-MIP 2.1) against observed streamflow

The purpose of WaterGAP has been to provide a best estimate of renewable water resources worldwide. To obtain meaningful estimates of water resources despite different sources of uncertainty related to GHMs, a calibration routine was applied (see Döll et al., 2003; Hunger and Döll, 2008; Müller Schmied et al., 2014). The calibration routine in WaterGAP 2.2 (ISI-MIP 2.1) forces the long-term annual simulated river discharge (Q) to be equal (within ±1%) to observed long-term annual discharge at grid cells representing calibration stations, for the period of observations (with a maximum of 30 years of observation being considered). With alternative climate forcings, basin-scale differences in Q and (subsequent) actual evapotranspiration (AET) therefore occur especially in catchments without calibration stations or during years without observed discharge. Figure 1 shows the land grid cells that are affected by calibration in this study, incorporating around 54% of the global land surface (excluding Antarctica and Greenland). We calibrated the model for each of the four climate forcings GSWP3, PGFv2, WFD and WFDEI_hom (descriptions of acronyms in Sect. 2.3) against mean annual discharge at 1319 discharge observation stations from the Global Runoff Data Centre (GRDC) catalogue, except for WFD, where due to the earlier end of the forcing time series, only 1312 stations could be used. The calibration parameters of WFDEI_hom were then used for the WFD_WFDEI forcing. Observation stations were selected such that

the upstream area had a minimum of 9000 km². To avoid including stations that are located very close to each other along a river, the minimum interstation catchment area was set to 30 000 km². Furthermore, a station was selected only if a minimum of four complete years of data was available.

## 2.3 Climate forcing data sets

Within the ISI-MIP project phase 2.1, four state-of-the-art climate forcings were made available through the coordinating Potsdam Institute for Climate Impact Research (PIK): GSWP3, PGFv2, WFD, and WFD_WFDEI. For each forcing, daily values of the variables surface-level (raingage-level) precipitation (P), 2 m air temperature (T), shortwave downward radiation at the surface level (SWD) and longwave downward radiation at the surface level (LWD) were used to run WaterGAP. Due to inhomogeneity problems during overlapping periods of WATCH Forcing Data based on ERA-40 (WFD data set, 1901-2001) and WFD methodology applied to ERA-Interim (WFDEI data set, 1979-2010), a data homogenization method was applied. This resulted in a fifth homogenized climate forcing (WFDEI_hom). The name of the climate forcing is used to name the model variant. In all data sets, daily precipitation estimates were obtained by bias-correcting output of weather models by monthly precipitation data sets that had been derived from monthly precipitation observed at raingages. These monthly data sets were optimized for spatial coverage, i.e. using, for each month, the available number of gauging stations. The temporally variable number of precipitation observations makes the applied precipitation data sets less suitable for the analysis of temporal variations. While a temporally homogeneous data set of observation-based monthly precipitation exists at least for the time period 1950-2000, it is based on less than 10 000 gauging stations and therefore provides a spatially less accurate representation of global-scale precipitation (Beck et al., 2005) than the data sets used in this study, which include up to 50 000 gauging stations (Schneider et al., 2015).

### 2.3.1 Global Soil Wetness Project 3 (GSWP3)

For the third phase of Global Soil Wetness Project (GSWP), a century-long (1901-2010) high-resolution global climate data was developed (http://hydro.iis.u-tokyo.ac.jp/GSWP3). The 20th Century Reanalysis (20CR) done with the NCEP atmosphere land model (Compo et al. 2011) which has a relatively low spatial resolution (~2.0°) and long-term availability (140 years) was dynamically downscaled into the global T248 (~0.5°) resolution using Experimental

Climate Prediction Center (ECPC) Global Spectral Model (GSM) by spectral nudging data assimilation technique (Yoshimura and Kanamitsu, 2008). Also, Global Precipitation Climatology Centre (GPCC) version 6 (for P), Climate Research Unit (CRU) TS3.21 (for T) and Surface Radiation Budget (SRB, for SWD/LWD) were used for bias correction to reduce model-dependent uncertainty. Wind-induced P undercatch correction is applied depending on gauge type and their global distribution according to Hirabayashi et al. (2008).

### 2.3.2 Princeton Global Meteorological Forcing Dataset (PGFv2.1)

The Princeton Global Meteorological Forcing Dataset, version 2 (PGFv2) is an update of the forcing described by Sheffield et al. (2006). It blends reanalysis data (NCEP-NCAR) with station and satellite observations and covers in its current form the period 1901-2012 (http://hydrology.princeton.edu/data.pgf.php). P is bias corrected to monthly CRU TS3.21 but is not undercatch corrected (different to its previous version 1). Daily T is adjusted to match CRU TS3.21 monthly values by shifting. SWD is adjusted for systematic biases at monthly scale (using a product from the University of Maryland (by Rachel Pinker) developed within the NASA Measures project) and then for trends using CRU TS3.21 cloud cover. LWD is scaled to match the mean and variability of the University of Maryland data (see SWD) but retains the year-to-year variation of the NCEP data. All information on PGFv2 is based on personal communication with J. Sheffield in 2015. During the first review process of this manuscript, we were informed about an error in the T data for the period 1901-1947 for certain regions. We therefore present below results of a WaterGAP run driven by the corrected version PGFv2.1 but with calibration parameters determined by using PGFv2 as no significant effect of the erroneous T data on calibration is expected, because calibration periods start after 1947 except for 21 basins that are all located in regions where the error effect is small.

### 2.3.3 WATCH Forcing Data (WFD)

The WATCH Forcing Data (WFD) was developed by Weedon et al. (2010, 2011) in the scope of the European FP6-funded Water and Global Change (WATCH) project (www.eu-watch.org). The data-set is based on the European Centre for Medium-Range Weather Forecasts (ECMWF) 40-year reanalysis product (ERA-40) for the period 1958-2001 and on the reordered ERA-40 data for the period 1901-1957. The variables from ERA-40 are interpolated (taking into account elevation) and some are corrected to monthly observation

data, e.g. P is corrected using GPCC version 4 observations (details in Weedon et al., 2010, 2011). Monthly P is corrected for wind-induced undercatch according to Adam and Lettenmaier (2003). Monthly T is corrected to CRU TS2.1 and SWD is corrected to cloud cover of CRU TS2.1 whereas LWD is not bias corrected (Weedon et al. 2010).

### 2.3.4 Combined WFD and WFDEI (WFD_WFDEI)

The WFDEI dataset was created by applying the WFD methodology to the newer ERA-Interim reanalysis data of ECMWF which is improved compared to ERA-40, especially for SWD (Weedon et al., 2014). WFDEI is available for the period 1979-2010, with P bias corrected to GPCC version 5 (and version 6 for 2010) and using ratios from Adam and Lettenmaier (2003) for correction of P undercatch. SWD in WFDEI is almost everywhere on the globe larger than SWD of WFD, with differences between 15 and 100 $W/m^2$ in most of Africa and Europe, due to changes in aerosol distribution in ERA-Interim as compared to ERA-40 (Dee et al., 2011; Weedon et al., 2014). Monthly values for T are bias corrected to CRU TS3.1/3.21 and SWD to cloud cover of CRU TS3.1/3.21. WFD_WFDEI as provided by ISI-MIP 2.1 is a simple time-consecutive combination of WFD (1901-1978) and WFDEI (1979-2010), which can be problematic when not checking for offsets (Weedon et al., 2014). Müller Schmied et al. (2014) used the same concatenating approach and found considerable offsets in WaterGAP simulated water balance components. Due to the strong global increase in SWD in WFDEI relative to WFD for overlapping periods (1979-2001), global AET increased by ~5 000 $km^3\ yr^{-1}$, which affects resulting water storages and global sums of Q (Müller Schmied et al., 2014).

### 2.3.5 Homogenized combined WFD and WFDEI (WFDEI_hom)

To overcome the offset in selected climatic variables between WFD and WFDEI, a homogenization approach analog to the bias correction approach in Haddeland et al. (2012) was applied to the daily data for three climatic variables SWD, LWD, and T. For SWD and LWD, a multiplicative approach was applied (Eq. 1), whereas T was homogenized with an additive approach due to possible zero values (Eq. 2) and P was not homogenized as only marginal differences in continental and global sums occur (Table 4).

$$V_{hom} = V_{WFD} * \frac{\overline{V_{WFDEI}(m)}}{\overline{V_{WFD}(m)}} \quad (1)$$

$$V_{hom} = V_{WFD} + \overline{V_{WFDEI}}(m) - \overline{V_{WFD}}(m) \quad (2)$$

with $V_{\text{hom}}$ being the homogenized daily variable (1901-2001), $V_{WFD}$ the original daily variable from WFD (1901-2001), $\overline{V_{WFDEI}}(m)$ and $\overline{V_{WFD}}(m)$ the long-term mean monthly variable from WFDEI and WFD for the overlapping time period 1979-2001, applied to the current month. The final homogenized daily WFDEI_hom time series consists of homogenized WFD data until 1979 and of WFDEI data afterwards. As the averages of SWD and T during the overlapping period are larger for WFDEI than for WFD, WFDEI_hom values until 1978 are larger than respective original WFD values, also included in WFD_WFDEI time series. The opposite is true for LWD which is furthermore only slightly adjusted compared to SWD.

## 2.4 Calculation of spatial averages and indicators

### 2.4.1 Calculation of spatial averages

The calculation of global averages for climate forcing variables as well as water balance components are based on all land grid cells excluding Antarctica (not represented), Greenland, and those grid cells that represent inland sinks. For T, SWD and LWD, area-weighted averages were calculated. Q was calculated for global totals by summing up Q of all grid cells that are outflow cells into the ocean according to the drainage direction map DDM30 (Döll and Lehner, 2002) and Q into all grid cells that represent inland sinks. The same procedure was used for the continental assessment (with all of the Russian Federation considered to belong to Europe in this study). For the calibrated and non-calibrated regions, the sum of net cell runoff (Q flowing out of the grid cell minus Q flowing into the grid cell) was used.

### 2.4.2 Indicator for relative dominance of precipitation or anthropogenic impact on discharge variability

To answer research question 3, i.e. to determine whether the change of long-term average discharge between two consecutive 30-year periods is caused mainly by the change of P in the upstream river basin or by the change of anthropogenic impact on Q by human water use and dam construction, two indicators were developed and combined. In the equations below, Q represents simulated discharge under anthropogenic conditions, whereas Qnat is the discharge

that would occur with neither human water use nor reservoirs or regulated lake regulation by dams.

First, we assume that P change cannot be more dominant a driver than change of anthropogenic impacts if P increases while Q decreases (and vice versa), expressed by the ratio of differences in Eq. 3. Furthermore, the runoff coefficient scales the ratio. Thus, indicator $A_n$ is computed as

$$A_n = C_{QP,n} \frac{P_{bas(n),t2} - P_{bas(n),t1}}{Q_{n,t2} - Q_{n,t1}} \tag{3}$$

where $A_n$ [-] is the indicator for dominance of P of grid cell $n$ with $P_{bas(n)}$ [km$^3$ yr$^{-1}$] as sum of P for the upstream area (contributing basin area) and $Q_n$ [km$^3$ yr$^{-1}$] the simulated river discharge of the grid cell between the time periods $t1$ (e.g. 1941-1970) and $t2$ (e.g. 1971-2000). The runoff coefficient $C_{QP,n}$ [-] is calculated as the averaged mean runoff coefficient of the two time periods under consideration

$$C_{QP,n} = avg\left(\frac{Qnat_{n,t1}}{P_{bas(n),t1}}, \frac{Qnat_{n,t2}}{P_{bas(n),t2}}\right), \tag{4}$$

where $Qnat_n$ [km$^3$ yr$^{-1}$] is the simulated river discharge of the grid cell of the model runs without human water abstractions and reservoir operation. The runoff coefficient is independently calculated for the two time periods.

If changes of P and Q have the same sign, $A_n$ is positive, and the change in P may be a significant driver of the Q change. If $A_n$ is negative, it can be excluded that the change of P is a dominant driver of the change in Q.

Indicator $B_n$ quantifies the anthropogenic impact on river discharge, expressed as the change in the difference between Q and Qnat compared to the change in Q. An increasing difference between Q and Qnat between the periods should lead to a decrease of Q.

$$B_n = \frac{\left(Q_{n,t2} - Qnat_{n,t2}\right) - \left(Q_{n,t1} - Qnat_{n,t1}\right)}{Q_{n,t2} - Q_{n,t1}} \tag{5}$$

where $B_n$ [-] is the indicator for dominance of anthropogenic impact on river discharge ranging from negative values, zero (for Q = Qnat) to positive values. If e.g. Q increases

between the two time periods but the difference between Q and Qnat decreases e.g. due to decreased human water use among the time periods, $B_n$ becomes negative, indicating that anthropogenic effects cannot be the dominant driver of change in Q.

The larger $A_n$ ($B_n$), the more likely P (anthropogenic effects) is the dominant driver of Q change, since the change in P (anthropogenic effects) is large. Consequently, P is more dominant a driver than change in anthropogenic impact if $A_n > B_n$ and $A_n > 0$. The change in anthropogenic impact is the more dominant a driver than change in P if $B_n > A_n$ and $B_n > 0$. If both $A_n <= 0$ and $B_n <= 0$, changes in Q are neither consistent with changes in P nor with changes in anthropogenic impact, and Q change is caused by other drivers, e.g. T. No assessment is possible, if there is no change in Q. To illustrate the indicator of relative dominance approach, Table 1 lists indicator values and underlying data for the example of four grid cells representing discharge of large rivers near the outlet to the ocean.

## 3 Results and discussion

### 3.1 Water balance components as impacted by climate forcing uncertainty

In this section, uncertainties of climate forcing are described first, followed by uncertainties of model output variables stemming from climate forcing uncertainty (Fig. 2). Spatial scales range from global (Table 2) to continental (Table 3) and to grid cells (Fig. 3). In addition, we differentiate between calibrated and non-calibrated regions (Table 5). Finally, values of water balance components are compared to values from other studies (Table 6).

#### 3.1.1 Uncertainty of global climate forcings

The 1971-2000 global P differs among the model forcing variants, with the largest difference found between CRU based product (PGFv2.1) and the GPCC based products (all other forcings) amounting up to 7500 km$^3$ yr$^{-1}$ (Table 2). Even the GPCC-based forcings vary by up to 1400 km$^3$ yr$^{-1}$ (exceeding the amount of actual water consumption WCa). Oceania (with the lowest absolute value) has the lowest deviation among the forcings (Table 3). The largest deviations are found in North America, Europe, and Africa. In North America and Europe, where the station density is comparably high and GPCC versions agree very well (Table 4) but in winter precipitation falls often as snow (with strong undercatch in gauging devices), the

different approaches to undercatch correction of P lead to large P deviations among the climate forcings. In case of WFD and WFDEI, monthly precipitation data are undercatch corrected according to Adam and Lettenmaier (2003); in case of GSWP3 a correction described in Hirabayashi et al. (2008) is applied, while there is no undercatch correction in PGFv2.1. While the calibrated grid cells cover 53.9% (53.7% for WFD) of global land area (excluding Antarctica and Greenland), they receive 61.0-61.5% of P (for all forcings, Table 5). The variation among the forcing variants, calculated as (maximum P – minimum P) / mean P, is with 7.5% in calibrated basins slightly higher than in non-calibrated basins (6.1%) (Table 5).

Global averages of T for 1971-2000 are very similar for all forcings, which is not surprising as all of them are bias corrected to (different) versions of the CRU time series. Global annual averages over the 30 years differ between the warmest (PGFv2.1) and coldest (WFD) forcing by only 0.08°C.

SWD is the forcing variable which has large differences throughout the forcings (Fig. 2). Remarkably lower values are found for WFD (compared to GSWP3 and PGFv2.1) which is a result of the underlying reanalysis and affects dominantly Africa and Europe (Sect. 2.3.4, Table 3, Weedon et al., 2014). The concatenation approach (which is also used in the ISI-MIP project phase 2.1) of WFD_WFDEI leads to a very strong increase (by on average ~15 W $m^{-2}$) starting in 1979. Homogenizing WFD eliminates this effect (WFDEI_hom, Fig. 2). Variations of global LWD are rather low (Table 3).

### 3.1.2 Uncertainty of simulated water balance components due to climate forcing uncertainty

Climate forcing uncertainty propagates to all water balance components simulated by WaterGAP. For the period 1971-2000, global Q varies among the five forcings by about 3400 $km^3$ $yr^{-1}$ (Table 2). On the continental scale, the strongest climate forcing-induced variation of Q occurs in Africa (Table 3). Here, some areas with high amounts of P (and Q) are in non-calibrated regions (e.g. Madagascar, see Fig. 3). Besides, the runoff coefficient (Q/P) of Africa is with 0.21 the lowest compared to all other continents, varying between 0.34 (Oceania) and 0.47 (Europe). A low runoff coefficient leads to the translation of a small precipitation deviation (in percent of mean) to a relatively large discharge deviation, as can also be seen for Oceania (Table 3).

While calibrated basins cover 54% of the global land area excluding Greenland and Antarctica (Fig. 1), 53-58% of global Q flows out of calibrated basins (Table 5). Most of the Q from non-calibrated basins is simulated to occur in tropical regions, particularly in Indonesia and other parts of South-East Asia. As expected, the sum of Q from all non-calibrated basins varies more strongly among the forcing variants (18.4%) than the sum of Q from all calibrated basins (2.8%, Table 5). Variation of Q from non-calibrated regions is reduced to 10.5% if the PGFv2.1 variant (the only forcing without precipitation undercatch correction) is excluded, while Q variation in the calibrated regions remains the same. If only the four homogeneous forcings (without WFD_WFDEI) are considered, Q varies by 18.5% for the non-calibrated region and by 1.6% for the calibrated one (Table 5, row 3). Q variation in calibrated basins is due to various reasons. Calibration forces the simulated mean annual discharges in the cells with discharge stations to be equal (within 1%) to the observed ones for the calibrated period. Outside the calibrated period, the different forcings cause the computed Q to vary. In case of the homogeneous forcings that are undercatch corrected and bias corrected against GPCC data (GSWP3, WFD, WFDEI_hom), Q differs by only 0.1% in calibrated regions and 10.5% in non-calibrated regions. The low value for the calibrated regions indicates neglectable influence of the different calibration periods and the smaller number of calibration stations in the case of WFD. The Q variation for the discharge produced in non-calibrated regions appears to be large in particular because all forcings are bias-corrected against monthly observations of temperature from CRU, and of P from GPCC (Table 4). This indicates a dissimilar spatio-temporal distribution of SWD and LWD radiation components. The larger deviation of WFD_WFDEI in calibrated regions (Table 5, row 3) can be explained by the fact that in order to deal with the offset problem in the WFD_WFDEI forcing, the WFDEI_hom calibration parameters were also used for the model variant that was driven by WFD_WFDEI (see Appendix A). Due to the much lower values of SWD in WFD as compared to WFDEI, GWSP or PGFv2.1, Q as computed with WFD has the highest value of all variants in non-calibrated regions, and is 11% larger than Q computed with WFDEI_hom for 1971-2000 (Table 5). One may conclude that GHMs without a calibration routine overestimate Q if driven by WFD; this may be one reason for the comparably high multi-model Q estimate of 42 000 – 66 000 $km^3$ $yr^{-1}$ reported in Haddeland et al. (2011) which is much higher than previous estimates (e.g. Baumgartner and Reichel, 1975; Fekete, 2002) or this study.

Figure 3 shows the uncertainty range of Q at grid-cell level for calibrated and non-calibrated regions caused by the four homogeneous forcings. In both calibrated and non-calibrated regions, the highest absolute differences occur in cells with large discharge, either in the downstream part of large rivers (e.g. Nile in Fig. 3a) or in areas with high precipitation (e.g. coast of Alaska or in Papua New Guinea in Fig. 3b). The lowest relative differences in the calibrated regions occur up- and downstream of the 1319 discharge gauging stations that were used for model calibration (Fig. 3c). The effect of calibration is also visible in non-calibrated regions downstream of a gauging station, e.g. in the Amazon downstream of Obidos (Fig. 3d). Even in most areas of the globe which are calibrated, i.e. in grid cells upstream of calibration stations, relative Q variations due to variations in climate forcings exceed 10% (Fig. 3c). In many cells not only in dry regions, variations exceed 50%. In non-calibrated regions, grid cells with relative Q variations below 10% are very rare unless they are located downstream of calibration station (Fig. 3d). In general, relative variations of Q are often higher in non-calibrated (Fig. 3d) than in calibrated regions (Fig. 3c) mainly because dry areas are less likely to have calibration stations. However, humid Iceland, for example, also exhibits simulated Q variations of more than 50%. When averaged over all grid cells globally (with Q > 0), variation of Q due to variation of the four homogeneous forcings is 55% (1.3 $km^3$/yr). For calibrated regions, the variation reduces to 37% (1.6 $km^3\ yr^{-1}$), while it increases to 74% (1.0 $km^3\ yr^{-1}$) in non-calibrated regions. When considering net cell runoff R in all cells with positive values, i.e. the runoff added to upstream discharge within a cell, variations due to the climate forcings grow to an average of 64% in calibrated regions and an average of 92% in non-calibrated regions. When considering only GSWP3 and WFDEI_hom, i.e. additionally excluding the forcings without undercatch correction (PGFv2.1) and with a much lower SWD than the others (WFD), the Q (runoff) variations are reduced to 16% (27%) and 31% (38%) in calibrated and non-calibrated regions, respectively. Reduction due to excluding PGFv2.1 is larger than reduction due to excluding WFD.

Global AET is the variable with the highest relative uncertainty due to climate forcing (Table 2). As Q within the calibrated region is forced to be nearly equal for all climate data sets, different values of P (as well as T and radiation) lead to large differences in aggregated AET (with higher absolute differences than P differences, or 12.2%). In contrast, AET differs by only 8.8% (and lower absolute differences than the P differences) in non-calibrated regions (both numbers for all forcings, Table 5). 63-67% of AET occur in calibrated regions (Table 5, row 2). In WFD forcing, the low global values for SWD lead to relatively low AET and

higher Q (2000 $km^3$ $yr^{-1}$) compared to the homogenized forcing WFDEI_hom. PGFv2.1 has the lowest global AET but the highest WCa of all five forcings (Table 2), even though WCa includes mainly evaporation of irrigation water that is driven by the same climatic variables as AET. This reflects the variations in the spatial pattern of the climatic variables among the five forcing data sets.

For the period 1971-2000, global WCa varies among the five forcings by 45 $km^3$ $yr^{-1}$ (Table 2), i.e. the range is less than 5%. 56.0 to 56.9% of global WCa occurs in calibrated regions (Table 5, row 4). Among all forcing variants, deviation of WCa is higher in calibrated regions (5.9%) than in non-calibrated regions (4.4%) (Table 5). WCa uncertainty due to climate forcings differs strongly among the continents (Table 3). For Asia, the continent with the highest water use, variation among the model variants is very low, indicating good agreement of climate forcing for the irrigation sub-model and/or averaging out differences in climate forcings over the large number of grid cells in Asia with irrigation water use. Again, Europe and North America have high uncertainties in continental assessments due to climate forcing uncertainty/variability.

### 3.1.3 Comparison with other studies

Global sums of AET and Q for the five climate forcings used in this study are within the range of estimates reported in the literature (see values from various sources in Müller Schmied et al. (2014), their Table 5). Values for AET of this study (64 400 – 70 800 $km^3$ $yr^{-1}$ including WCa) are well within this range. Global values for Q (39 200 – 42 200 $km^3$ $yr^{-1}$) are at the upper end of values from literature (except Haddeland et al., 2011).

Table 6 shows a comparison to global and continental estimates of AET and Q of this study to four recent reference studies. Time span and spatial coverage of WaterGAP results is the same as in the respective references, and the climate forcing variant of WaterGAP was selected such that studies using P without undercatch were compared to results of WaterGAP using PGFv2.1. Wisser et al. (2010) used the WBMplus model with CRU forcing plus three different precipitation data sets for an uncertainty analysis. Even though their P was not undercatch corrected and also scaled to CRU observations (like PGFv2.1), global P of PGFv2.1 is 2.7% lower for the time period 1901-2002 (and lower between 2.2 and 3.1% in the different time periods analyzed). For this period, WaterGAP simulates around 7.2% less

AET and 4.2% more Q compared to Wisser et al. (2010), but differences are varying for the other time periods analyzed (AET: 6.0-8.7%, Q: 2.6-6.6%).

Hanasaki et al. (2010) used a climate forcing that is scaled to CRU TS 2.1 and not undercatch corrected for the time span 1985-1999. Therefore, results of WaterGAP driven by PGFv2.1 were used for the comparison in Table 6. Their values included probably also Antarctica as they mention a land area of 144 000 km², so a direct comparison is not straightforward. Based on the assessment of Rodell et al. (2015) (see next paragraph), Antarcticas's share in global P, AET and Q is about 2.1%, 0.2% and 4.9%, respectively,, and these percentages were added to the WaterGAP results. Surprisingly, global P of PGFv2.1 is 7.4% lower than P of Hanasaki et al. (2010). As a consequence, AET (by 12.1%) and Q (by 2.3%) are also lower for WaterGAP forced with PGFv2.1 compared to Hanasaki et al. (2010) (Table 6).

Rodell et al. (2015) provide an optimized consistent set of global and continental water fluxes during 2000-2010 by combining satellite products and outputs from a number of models in an optimization routine that enforced multiple water and energy budget constraints simultaneously. Compared to WFDEI_hom (this study), global P is nearly equal (0.8% lower). WaterGAP simulated AET slightly higher (1.5%) but Q is 6.0% lower compared to Rodell et al. (2015) (Table 6). As the definition of continents differs partly between Rodell et al. (2015) and this study, only North and South America as well Africa can be compared. PGFv2.1 continental estimates for P are 1.5% and 3.3% higher for North America and Africa, and 1.5% lower for South America, with WaterGAP AET being higher (1.6 to 4.1%). Large differences occur for Q, where WaterGAP estimated 19.5% and 6.2% lower values for North and South America, and 5.2% higher values for Africa. North America and Africa are the continents, which show high variations in Q also in this study among the forcings (Table 3).

Considering the water balance component values of Wisser et al. (2010), Hanasaki et al. (2010) and Rodell et al. (2015), there is no water balance component for which WaterGAP values are consistently too high or too low. Even when the climate forcings used in these studies are similar to one of the climate forcings used here (e.g. regarding undercatch and bias correction), global P values differ, which in itself leads to different model output. Therefore the approach of the many model intercomparison studies to use the same climate forcing for all models helps to assess the differences of the models themselves (Haddeland et al., 2012).

The WaterGAP 2.2 (ISIMIP 2.1) water balance components using WFD_WFDEI climate input (Table 2) differ from those of the STANDARD WaterGAP 2.2 model runs that was also

driven by WFD_WFDEI as presented in Müller Schmied et al. (2014) (their Table 2) due to the seven model modifications listed in Appendix A. Global P is insignificantly affected by the different landmasks. Global AET and Q are comparable and differ only by 1 to 2% between both studies (Table 6). Due to the assumed deficit irrigation in groundwater depletion areas (Sect. 2.1), global WCa during 1971-2000 is estimated as 936 km$^3$ yr$^{-1}$ as compared to 1031 km$^3$ yr$^{-1}$ in STANDARD. Deficit irrigation also explains the smaller decrease of groundwater storage in this study, with an average of 75 km$^3$ yr$^{-1}$ during the period 1971-2000 compared to 125 km$^3$ yr$^{-1}$ in STANDARD (Müller Schmied et al., 2014, their Table 3). In the applied WaterGAP 2.2 (ISIMIP 2.1) version, reservoirs are filled up with water in their construction year. This leads to a net increase of reservoir storage (53 km$^3$ yr$^{-1}$) compared to a decrease of 43 km$^3$ yr$^{-1}$ in STANDARD, where reservoirs are assumed to have been in operation over the entire simulation period. Thus, total water storage decreased less than in STANDARD, with 74 km$^3$ yr$^{-1}$ instead of 215 km$^3$ yr$^{-1}$.

## 3.2 Variation of estimated global water balance components across temporal aggregation and reference periods

Figure 2 shows the importance of temporal aggregation and reference periods for the assessment of global-scale climatic variables and water balance components during the time period 1901-2010 (2001 for WFD, 2012 for PGFv2.1). Even for globally aggregated components, there are strong year-to-year fluctuation. To assess (next to the visual interpretation) the importance of the choice of temporal aggregation for the different climatic variables or water balance components on their variability during the simulation period, the ranges of their global values at temporal aggregations of 1, 10 and 30 years were first computed as the difference between the maximum and the minimum value during the whole time period. Then the effect of temporal aggregation was quantified by calculating the ratio of the ranges at the different temporal aggregations. For all climate variables and water balance components except those with a significant trend, the ranges (Fig. 2) and ratios vary strongly among the forcing variants. To achieve an approximate but robust representation of the effect of temporal aggregation on variability, we present only the median of the ratios among the four homogeneous forcings. Regarding the radiation variables SWD and LWD, their range is approximately halved when going from 30 to 10 years or from 10 to 1 year, and consequently reduced by a factor of 3-4 when going from 30 to 1 year. Global P and AET range is reduced by a factor of about 2 when going from 30 to 10 years or by a factor of 3 when going from 10

to 1 year. Regarding global discharge, the corresponding ratios are approximately 2 and 4. Here, the variation among the four forcings is 1.6-2.7 for the reduction of variability when going from 30 to 10 years and 3.0-5.4 when going from 10 to 1 years. Quantifying temporal variability of global WCa, which has a significant trend (Fig. 2), the range of 1-year and 10-year aggregates is very similar, while the range is reduced by a factor of 1.6 when going from 30 to 10 years. Considering the variability of T, the ranges during the simulation period are around 1.5 °C (1 year), 1.1 °C (10 years) and 0.4 °C (30 years).

Regarding the choice of reference period, its importance is obvious in case of the variables with a strong temporal trend like T and WCa. Increase of global averages of T during the last three decades is comparable among the five climate forcings as they are all bias corrected to almost the same observation-based product (CRU TS, but different versions). Large differences occur for 100 year average SWD, for which WFD forcing shows an offset of around -15 $Wm^{-2}$. This also affects the combined WFD_WFDEI, resulting in an implausible discontinuity from 1978 to 1979. The monthly homogenized series (WFDEI_hom) reduces this offset, but the (smaller) offset within WFD since 1973 (integration of first NOAA VTPR satellite data, Uppala et al., 2005) cannot be reduced by this method. LWD shows different variations among the climate forcings at annual, decadal and 30-year aggregations (e.g. between GSWP3 and PGFv2.1), while the 100-year averages are relatively close to each other. Again, in WFD (and consequently WFD_WFDEI and WFDEI_hom) the usage of satellite data in the ERA-40 reanalyses from 1973 onwards leads to an offset in LWD which is clearly visible in the 30-year averages (1971-2000) in all three forcings. Except PGFv2.1, all climate forcings indicate an increase of LWD in the last decades which fits to increasing T. Using land surface parameters and T, WaterGAP calculates the outgoing components of radiation and subsequently net radiation which is then used to calculate potential evapotranspiration. In WFD, net radiation is much lower than in the other data sets (century mean 72 $Wm^{-2}$ compared to 83 $Wm^{-2}$ for WFDEI_hom and 86 $Wm^{-2}$ for GSWP3 and PGFv2.1) (Fig. 2). Considering the four homogeneous forcings only, temporal variations of net radiation are low but rather different among the forcings, and there is no significant trend except for PGFv2.1 with a decreasing trend in the last 30 years. Global PET has an even smaller variation, and no trend during the century either. Global P seems to be slightly smaller before 1940 than afterwards but this may be due to the lower number of rain gauges available during this time period. After 1940, 30-year averages of global P are almost constant in time. This is supported by Beck et al. (2005) who found no significant trend in global P for 1950-

2000 when utilizing observations from the same set of rain gauges over the whole analysis period.

Neither can trends of global AET or Q be detected. The decadal or 30-year variations vary strongly among the forcings. For Q (AET), the inhomogeneity in WFD_WFDEI leads to an implausible decrease (increase) of around 5000 $km^3$ $yr^{-1}$. Among the homogeneous forcings, WFDEI_hom shows low Q (high AET) during the last three decades as compared to the previous decades and as compared to the other forcings, even though PET of all those forcings does not show a trend. This might be related to differences in spatial patterns among the forcings. The results of this study confirm the finding of the IPCC Fifth Assessment Report that “the most recent and most comprehensive analyses of river runoff do not support the IPCC Fourth Assessment Report (AR4) conclusion that global runoff has increased during the $20^{th}$ century” (Stocker et al, 2013, p. 44). Century means of global Q from GSWP3 and WFDEI are very similar (like their P and PET values), while Q is smaller in case of PGFv2.1 due to lower P (compare section 3.1) and higher in WFD due to lower SWD (and thus PET).

WCa is the only water balance component with a strong temporal trend (strong increase since the 1950s) and only a small variation of annual values around the trend that is mainly caused by expansion of irrigated land. Interannual variability is due to climate variability affecting irrigation water use. Temporal aggregation over a decade appears to be appropriate to clearly show the trend. The separation of total water use into the different sectors as well as into water withdrawals and consumptive use is presented by Müller Schmied et al. (submitted to Proceedings of the International Association of Hydrological Sciences (PIAHS), their Fig. 3).

When comparing the output of different GHMs, the climate forcing used as model input is a very strong determinant of model output (see section 3.1). When GHMs driven by (more or less) the same climate forcing are compared (see comparison of WaterGAP to Wisser et al., 2010, in Table 6), the choice of reference period matters. Differences for global P, AET and Q among the four roughly 25-year time periods are 3.2, 2.5 and 4.9%, respectively, for WaterGAP in this study and 2.6, 3.0 and 5.5% for Wisser et al. (2010).

## 3.3 Dominant drivers of temporal variations of 30-year mean annual river discharge: precipitation or human water use and dam construction

Figure 4 shows where the change of long-term average Q between the time period 1941-1970 and the time period 1971-2000 is either caused mainly by the change of P in the upstream

river basin (blue colours) or by the change of the anthropogenic impact on Q by human water use and dam construction (red colours, see Sect. 2.4.2). Results for WaterGAP as driven by each of the four homogeneous climate forcings GSWP3, PGFv2.1, WFDEI_hom and WFD are shown. In most regions, change in P is the more important driver of change in Q than change in the anthropogenic impact. It is areas with high water consumption or/and the construction of dams where change in anthropogenic impact is more important than change in P for explaining temporal Q changes. Note that the developed indicators only compare the relevance of two drivers of change. Even in the blue and red grid cells, other variables such as T or radiation may be even stronger drivers of the simulated change in Q. In grid cells where indicators $A_n$ and $B_n$ are both negative or zero (green colours), however, other drivers (and not P or anthropogenic effects) are certainly the main reason for changes in Q.

Changes in long-term average Q between the time periods 1911-1940 (t1) and 1941-1970 (t2) are, in most world regions, less dominated by changes in the anthropogenic impact on river discharge (Fig. 5). Anthropogenic impact increases in the time period 1941-1970 and 1971-2000 (t3), which is consistent with the acceleration of human water use (Fig. 2) and dam construction throughout the 20$^{th}$ century. In the earlier analysis period, anthropogenic activity dominates Q change only in small parts of North America and Asia (around the North China Plain and inflows to the Caspian Sea). It is only in the later analysis period that anthropogenic impact dominates over P impact in India, Southeast China, Spain and Turkey (compare Figs. 4 and 5). Taking India and GSWP3 forcing as an example, P increases for both time steps (t1-t2: +49 km$^3$ yr$^{-1}$, t2-t3: +30 km$^3$ yr$^{-1}$). However, Q (+9 km$^3$ yr$^{-1}$) and WCa (+26 km$^3$ yr$^{-1}$) increases between t1 and t2, while between t2 and t3 Q (-39 km$^3$ yr$^{-1}$) decreases and WCa (+81 km$^3$ yr$^{-1}$) increases more strongly than between t1 and t2. In India, the intensified water use and changed signs between P and Q lead to the indication that anthropogenic effects dominate the change in Q (compare Figs 4 and 5).

Human water use and dam construction is the dominant driver for changes in long-term Q averages on 9-13% of land area for the time period 1911-1940 and 1941-1970, and increases to 11-18% of land area for the time period 1941-1970 and 1971-2000. The fraction with P domination decreases, from 82-84% to 77-82%. At the same time, the area for which the indicators $A_n$ and $B_n$ cannot be calculated (due to similar long-term Q averages and thus zero in the denominators of Eqs.3 and 4) is rather constant (1.1% to 0.9%). The land fractions

where neither driver dominates decreases slightly from 6% to 5%. Figs. B1 and B2 in Appendix B shows $A_n$ and Figs. B3 and B4 shows $B_n$.

The four climate forcings affect the spatial pattern of dominance. They lead to different changes of P and different changes of human water use as the globally dominant irrigation water use is computed as a function of climate. For example, with the forcings based on ECMWF reanalyses (WFDEI_hom, WFD, Fig 4c-d), large parts in southeast Australia are driven by anthropogenic effects whereas for the forcings based on NCEP reanalyses this is not (PGFv2.1, Fig. 4b) or to a lesser extent (GSWP3, Fig. 4a) the case. For WFDEI_hom, the anthropogenic dominance is considerably higher in Mexico (Fig. 4). If using PGFv2.1 forcing, the area around the North China Plain is dominated by P changes whereas in the other forcings it is dominated by anthropogenic effects (Fig. 5). Even if mean global values, e.g. for P and Q, compare well (Fig. 2, Table 2), regional differences in the climate forcings (and underlain reanalysis) result in these different spatial patterns of GHM output.

The effects of human water use and dam construction on Q variations cannot be separated by the applied indicator approach. While dam construction leading to new reservoirs decreases long-term average Q (e.g. due to additional evaporation), human water consumption is expected to be more important in most grid cells (see also Döll et al., 2009).

## 4 Conclusions

This study presents a model-based assessment of water balance components considering different temporal (year to century) and spatial (0.5° grid cell to global) aggregations. The GHM WaterGAP 2.2 (ISI-MIP 2.1) was forced with an ensemble of four (plus one homogenized) state-of-the-art climate forcings with daily data. These forcings differ by the underlying reanalyses, the observational data sets used for bias correction and whether precipitation observations were corrected for undercatch. At global scale and for 1971-2000, P differs among the forcing by 7500 $km^3$ $yr^{-1}$ and Q about 3000 $km^3$ $yr^{-1}$. Estimated Q differs most among climate forcings where WaterGAP cannot be calibrated due to a lack of river discharge observations in the GRDC database, in particular in South-East Asia (Indonesia and Papua New Guinea). Variations among the four homogeneous forcings (GSWP3, PGFv2.1, WFD, WFDEI_hom) result, for 1971-2000, in a variation of long-term average Q aggregated over all non-calibrated areas of 18.5% but only in a variation of 1.6% for the calibrated areas. This supports the many calls for extending (or maintaining) in situ Q observations (e.g. Fekete et al., 2015) and for sharing the already available Q data (e.g. Hannah et al., 2011). Certainly,

satellite observations have the potential to support river discharge estimation (Tang et al., 2009). The Surface Water and Ocean Topography (SWOT) mission, for example, proposes discharge observations for river widths > 50 m but all remote sensing methods for deriving Q strongly rely on in situ measurements (Pavelsky et al., 2014).

On continental scale, most differences for P and Q among the homogeneous forcings (GSWP3, PGFv2.1, WFD, WFDEI_hom) occur in Africa and, due to snow undercatch of rain gauges, also in the data-rich continents Europe and North America. Variations of Q at the grid-cell scale due to uncertainty in meteorological data are large except in a few grid cells upstream and downstream of calibration stations, with on average 37% and 74% variation among the four homogeneous forcings. These large forcing induced uncertainties are disturbing because actual forcing data set uncertainty may not fully be represented by the ensemble and uncertainty due to the choice of hydrological model and its parameters is neglected.

The study underlined that the level of temporal aggregation of water balance components is of importance, such that for comparison purposes, the same temporal aggregation and identical reference periods should be used. However, for all variables except T and WCa, due to the uncertainty of climate data, the choice of the climate forcing affects climate variables and water balance components computed by GHMs more strongly than the choice of reference period. For global variables that (until now) show no significant trend (like P and Q), the widely used 30-year aggregation period is suitable for comparison purposes, while for variables showing a strong trend, i.e. T and WCa, decadal aggregation is recommended. Ranges of climate forcing variables and water balance components are reduced roughly by a factor of 2 when going from 30 to 10 years (and 10 to 1 years) and consequently by a factor of 3-4 when going from 30-year to 1-year assessment.

Homogenization of climate forcing is required when concatenating time series of meteorological variables from different sources, as in the case of WFD and WFDEI (which are based on two different reanalyses) are combined to cover the time period since 1901 until recent times. Even within the homogenized WFDEI_hom climate forcing there remains an offset in SWD and LWD data in 1973 that stems from the ERA-40 reanalysis; therefore it is recommended to start analysis if possible only after 1978 when ERA-Interim data are available. Anyway none of the four homogeneous climate forcings appears to be suitable for trend analyses as they are all bias-corrected against gridded monthly data derived from

observations of precipitation and temperature where the number of observation stations varies over time.

Humans affect the global water cycle increasingly. Comparing global sums of human water consumption to river discharge into oceans and internal sinks (or to renewable water resources), human impact seems to be small (Table 2). However, on 9-18% of global land area, human water consumption and dam construction was a more important driver of change in river discharge in the 20th century than precipitation (Figs. 4 and 5). In this study, however, only the impact on long-term averaged discharge was analyzed, while possible seasonal impacts e.g. due to reservoir operation were not considered (Adam et al., 2007; Döll et al., 2009).

For future water resources modeling studies (see also Döll et al., 2016), the impact of the uncertainty of meteorological variables should be considered by applying various (equally) plausible climate forcings. Using more than one GHM may add additional robustness. Such model intercomparison projects are currently on the way (e.g. ISI-MIP 2.1, eartH2Observe (www.earth2observe.eu/), The Agricultural Model Intercomparison and Improvement Project AgMIP (http://www.agmip.org/), Land Surface, Snow and Soil Moisture Model Intercomparison Project LS3MIP (http://www.climate-cryosphere.org/activities/targeted/ls3mip) or already finished (e.g. WATCH model intercomparison (Haddeland et al., 2011), ISI-MIP Fast Track (Schewe et al., 2014)). They may improve the quantification of the world's water resources and guide investigation of various sources of uncertainty. Development of an improved method for correcting the global state-of-the-art precipitation products, by building on the work of Fuchs et al. (2001), would enable a better quantification of global precipitation.

## Data availability

The WaterGAP output will become freely available for the public within the framework of the ISI-MIP project phase 2.1 but it is not yet known where the data will be hosted (please check https://www.pik-potsdam.de/research/climate-impacts-and-vulnerabilities/research/rd2-cross-cutting-activities/isi-mip/for-modellers/isi-mip-phase-2 for updates). The homogenized climate forcing WFDEI_hom is not included within the ISI-MIP 2.1a project. All model outputs used in this study are available on request from the corresponding author.

## Acknowledgments

The authors thank the Global Runoff Data Centre (GRDC, http://grdc.bafg.de), 56068 Koblenz, Germany, for providing the discharge data used in this study. We are also grateful to the ISI-MIP coordination team as well as the leaders of the ISI-MIP water sector (Simon Gosling and Rutger Dankers) for providing the climate forcings and for their support. Furthermore, we thank Wolfgang Grabs for organizing the international conference "Water Resources Assessment & Seasonal Prediction" (13-16 October 2015 in Koblenz, Germany) where some content of this paper was presented. Finally, we thank three anonymous reviewers for their valuable suggestions that improved the manuscript significantly.

## Appendix A: Modification of WaterGAP 2.2 (ISI-MIP 2.1) compared to WaterGAP 2.2

- A new land cover input based on MODIS data from the year 2004 (using the dominant land cover class per 0.5° cell instead of the land cover class at the grid centre).
- Updated lake and wetland inputs based on the Global Lakes and Wetlands Database (GLWD) (Lehner and Döll, 2004) and the Global Reservoir and Dam database (GRanD) version 1.01 (Lehner et al., 2011) as well as information on operation years from available electronic resources.
- Different ocean-land mask: while WaterGAP 2.2 uses the ocean-land mask from the IMAGE model (Alcamo et al., 1998), being the standard for WaterGAP development and covering 66 896 grid cells, here the WATCH-CRU ocean-land mask with 67 420 grid cells is used. Main differences occur in coastal areas (for which static attributes, such as soil moisture capacity, of the standard land mask are transferred to the new neighbouring cell, while some other coastal cells disappeared), and the inclusion of many more islands in the Pacific Ocean (that obtained attributes values from nearest grid cells).
- Deficit irrigation based on Döll et al. (2014), with only 70% of irrigation water demand in grid cells which have a groundwater depletion of at least 5 mm $yr^{-1}$ during

1980-2009 and where the fraction of water withdrawals for irrigation is larger than 5% of total water withdrawals for the same time period.

- Man-made reservoirs are no longer assumed to exist over the whole simulation period but only from the year of their construction onward. This includes also regulation of the outflow of natural lakes by dams.
- For lakes, reduction of evaporation due to decreasing lake area is calculated according to Eq. 1 in Hunger and Döll (2008), resulting in a lower but more realistic lake area and thus evaporation reduction with decreasing lake storage.
- For WaterGAP calibration, we used observed streamflow data from 30 years. For GSWP3, PGFv2.1 and WFD, we used data from 1971-2000 if available for the time period. Due to the offset in radiation of WFD_WFDEI forcing (and consequences for model results, see Müller Schmied et al., 2014), we calibrated WFDEI_hom using preferably the period 1980-2009 and used these calibration parameters for the WFD_WFDEI simulation.

## Appendix B: Indicators $A_n$ and $B_n$

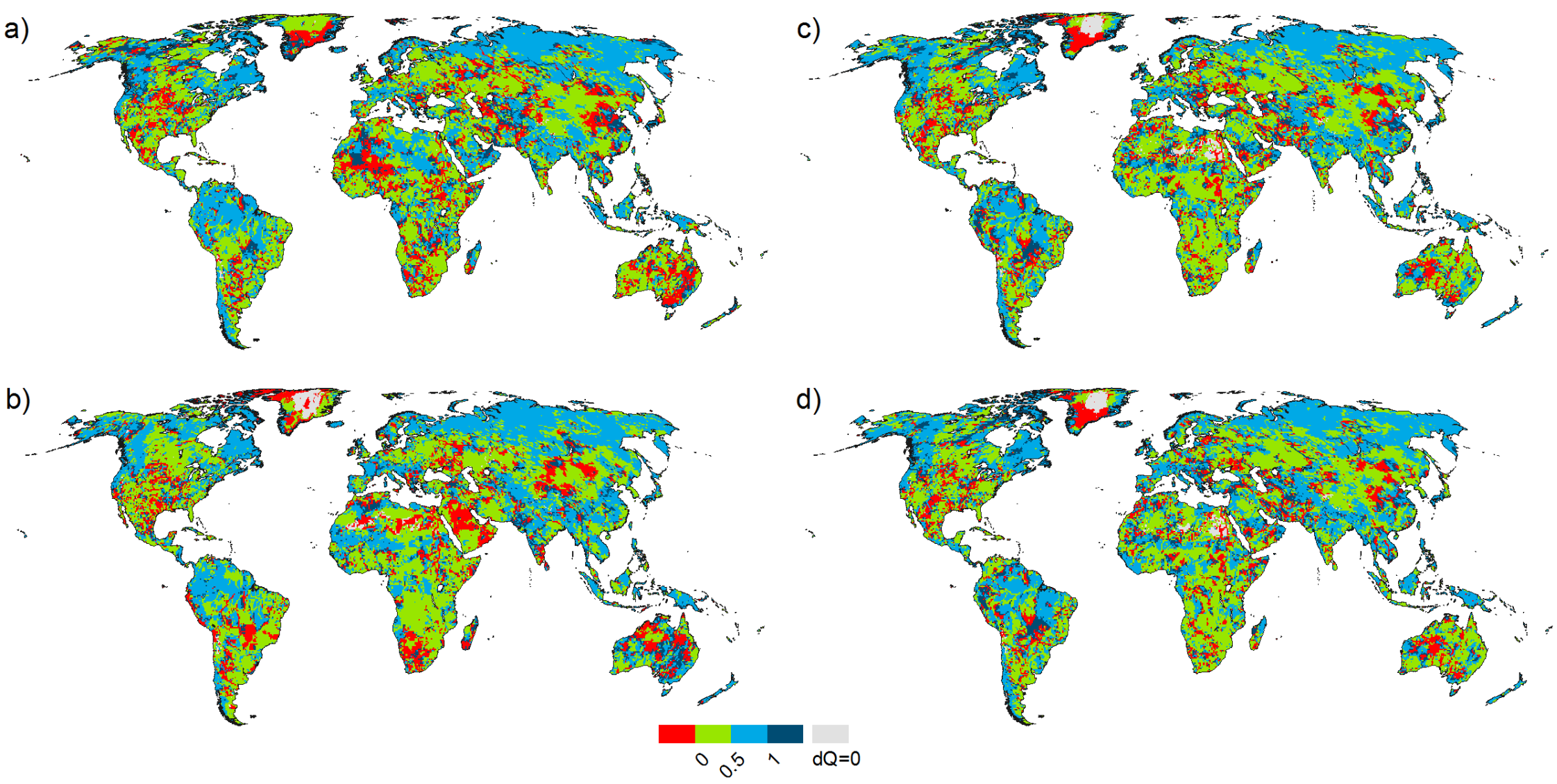

Figure B1: Indicator $A_n$ representing dominance of the change of P for the change in Q (Eq. 3) from 1911-1940 to 1941-1970. Grey colour indicates that the change in Q is zero such that $A_n$ cannot be computed. Red colour indicate areas where $A_n$ is negative, i.e. change in P had

the opposite sign of the change in Q; therefore, P was not the dominant driver for change in Q. Results are shown for WaterGAP as driven by the climate forcings GSWP3 (a), PGFv2.1 (b), WFDEI_hom (c) and WFD (d).

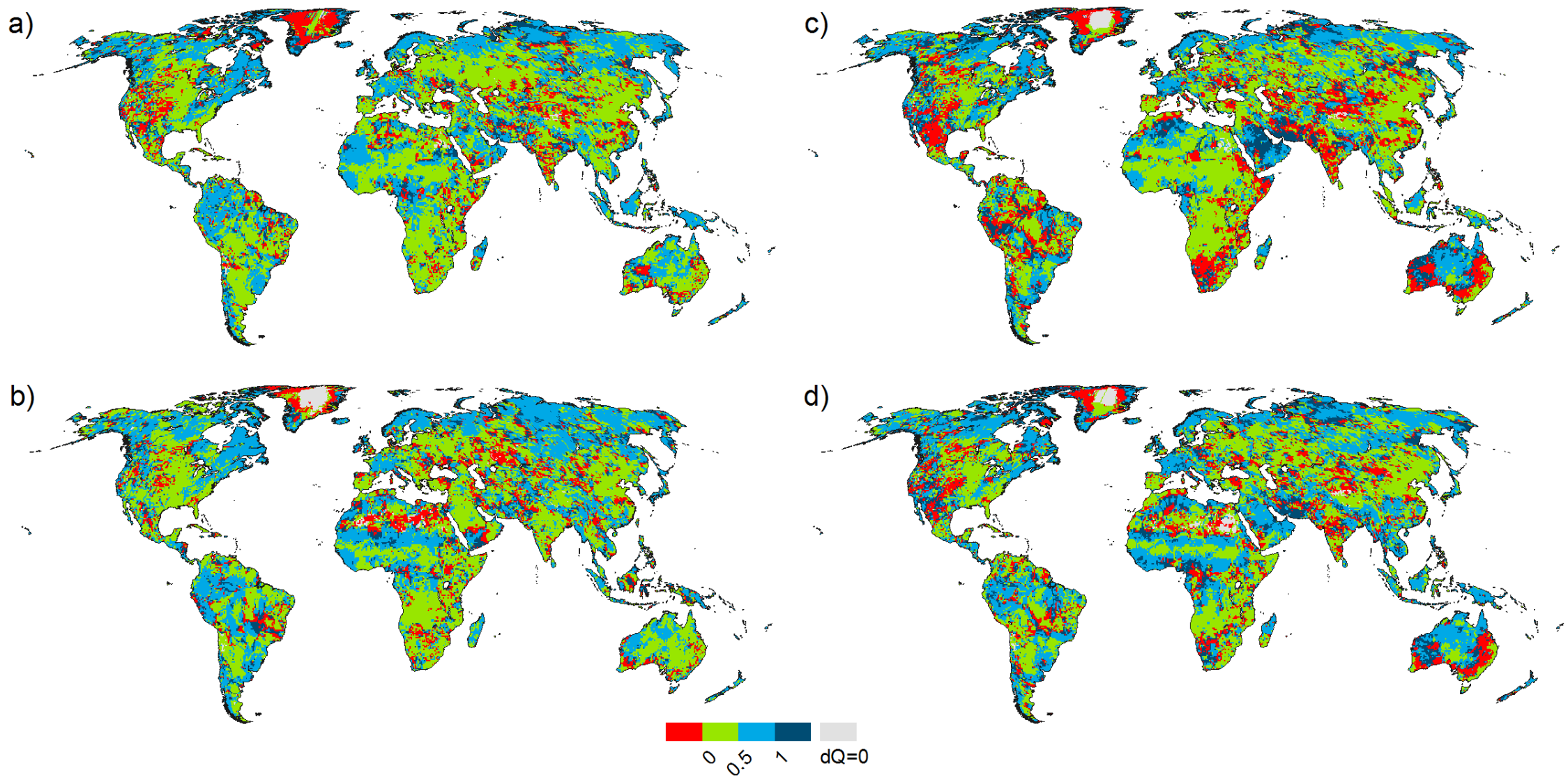

Figure B2: Indicator $A_n$ representing dominance of the change of P for the change in Q (Eq. 3) from 1941-1970 to 1971-2000. Grey colour indicates that the change in Q is zero such that $A_n$ cannot be computed. Red colour indicate areas where $A_n$ is negative, i.e. change in P had the opposite sign of the change in Q; therefore, P was not the dominant driver for change in Q. Results are shown for WaterGAP as driven by the climate forcings GSWP3 (a), PGFv2.1 (b), WFDEI_hom (c) and WFD (d).

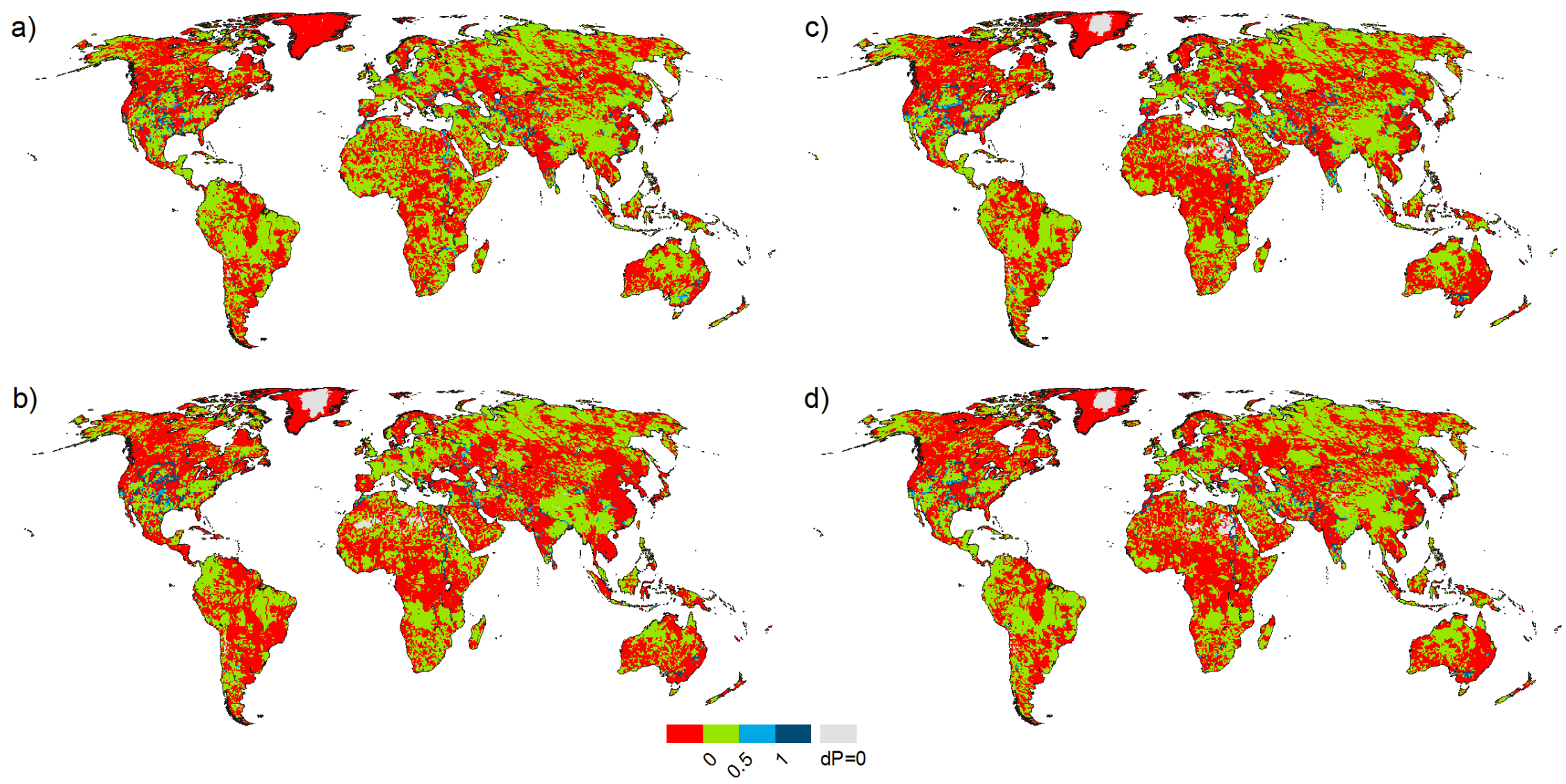

Figure B3: Indicator $B_n$ quantifying the relative dominance of anthropogenic impact on Q change (i.e. Q-Qnat) as compared to the change in Q (Eq. 5) from 1911-1940 to 1941-1970. Grey colour indicates that the change in Q is zero such that $B_n$ cannot be computed. Red colour indicates areas where $B_n$ is less than 0, and the change in anthropogenic impact is not consistent with the change in Q; therefore, the anthropogenic impact is not the dominant driver for change in Q. Results are shown for WaterGAP as driven by the climate forcings GSWP3 (a), PGFv2.1 (b), WFDEI_hom. (c) and WFD (d).

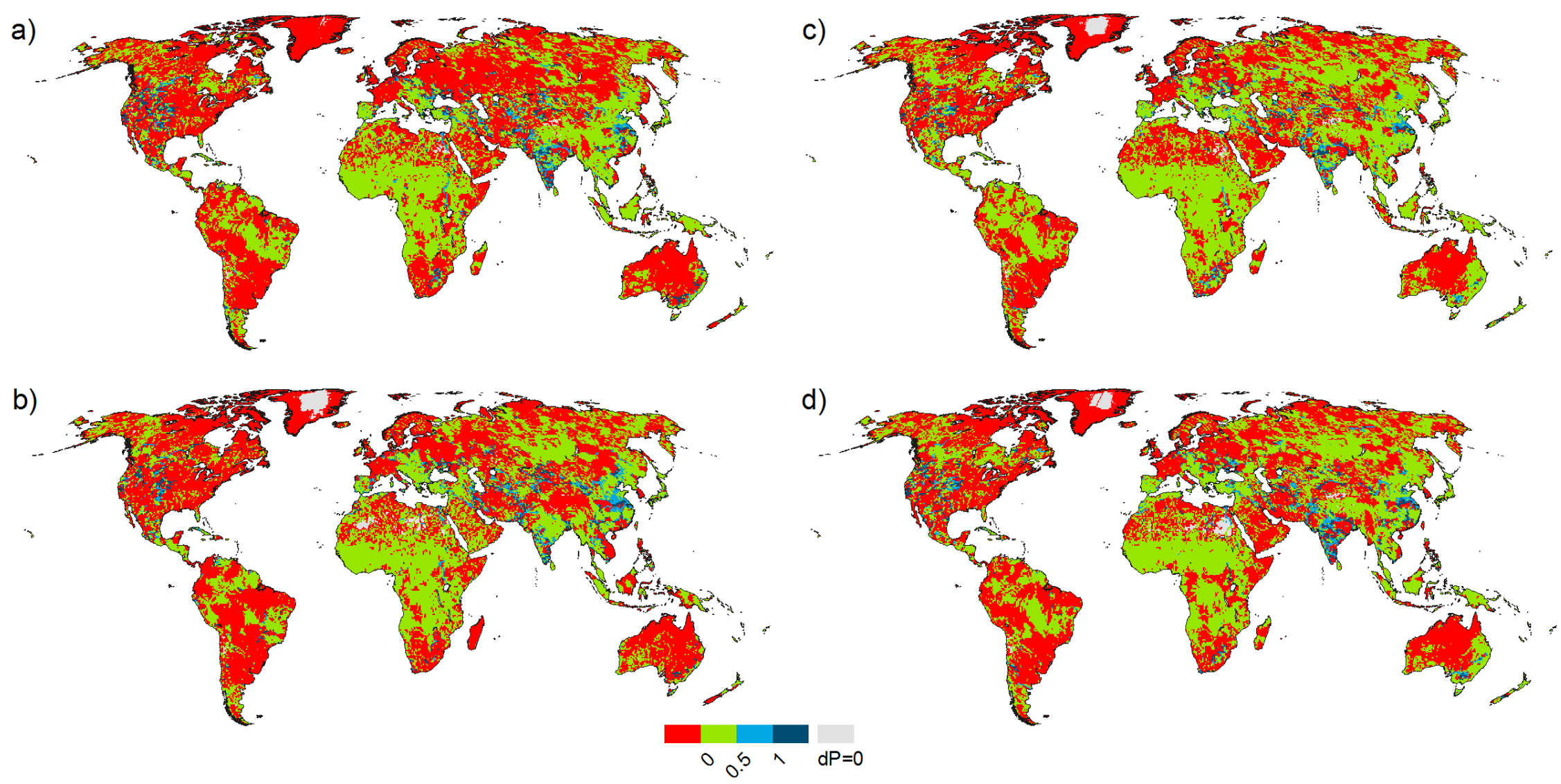

Figure B4: Indicator $B_n$ quantifying the relative dominance of anthropogenic impact on Q change (i.e. Q-Qnat) as compared to the change in Q (Eq. 5) from 1941-1970 to 1971-2000. Grey colour indicates that the change in Q is zero such that $B_n$ cannot be computed. Red colour indicates areas where $B_n$ is less than 0, and the change in anthropogenic impact is not consistent with the change in Q; therefore, the anthropogenic impact is not the dominant driver for change in Q. Results are shown for WaterGAP as driven by the climate forcings GSWP3 (a), PGFv2.1 (b), WFDEI_hom. (c) and WFD (d).

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

Table 1: Examples of indicator calculation (Sect. 2.4.2) for four large river basins at grid cells located near the outflow to the ocean for the forcing GSWP3 and changes from 1941-1970 (t1) to 1971-2000 (t2). Values for latitude and longitude in decimal degrees, values for $A_n$ and $B_n$ are dimensionless, other numbers in $km^3$ $yr^{-1}$. Explanations of variables other than Lat and Lon see Sect. 2.4.2.

| | Rhine River | Congo River | Colorado River | Yellow River |
|---|---|---|---|---|
| Lat | 4.25 | 12.25 | -114.75 | 133.25 |
| Lon | 52.25 | -6.25 | 31.75 | 48.25 |
| $P_{bas(n),t1}$ | 169.36 | 5735.52 | 191.24 | 771.92 |
| $P_{bas(n),t2}$ | 176.43 | 5469.11 | 206.56 | 771.91 |
| $Q_{natn,t1}$ | 69.27 | 1370.46 | 1.53 | 215.28 |
| $Q_{natn,t2}$ | 75.19 | 1251.09 | 1.92 | 209.94 |
| $Q_{n,t1}$ | 67.83 | 1370.46 | 0.62 | 213.41 |
| $Q_{n,t2}$ | 72.63 | 1250.67 | 0.10 | 203.68 |
| $A_n$ | 0.61 | 0.52 | -0.26 | 0.00 |
| $B_n$ | -0.23 | 0.00 | 1.76 | 0.45 |
| Dominant driver | $A_n > B_n$ & $A_n$>0: Precipitation | | $B_n > A_n$ & $B_n$>0: Human impact | |

Table 2. Global sums of water balance components for land area (except Antarctica and Greenland) [$km^3 yr^{-1}$] from WaterGAP (same sorting as Table 2 in Müller Schmied et al. 2014) for the five model variants and the years 1971-2000. Cells representing inland sinks were excluded but discharge into inland sinks was included.

| No. | Component | GSWP3 | PGFv2.1 | WFD | WFDEI_hom | WFD_WFDEI |
|---|---|---|---|---|---|---|
| 1 | Precipitation P | 109 631 | 103 525 | 110 690 | 111 050 | 111 050 |
| 2 | Actual evapotranspiration AET[a] | 68 026 | 63 416 | 67 588 | 69 907 | 68 887 |
| 3 | Discharge into oceans and inland sinks Q[b] | 40 678 | 39 173 | 42 200 | 40 213 | 41 298 |
| 4 | Water consumption (actual) (rows 5 + 6) WCa | 933 | 960 | 915 | 949 | 932 |
| 5 | Net abstraction from surface water (actual)[c] | 1050 | 1071 | 1023 | 1070 | 1044 |
| 6 | Net abstraction from groundwater[d] | -117 | -111 | -108 | -121 | -112 |
| 7 | Change of total water storage dS/dt[e] | -14 | -30 | -20 | -25 | -74 |
| 8 | Long-term-averaged yearly volume balance error (P-AET-Q-WCa-dS/dt) | 6 | 6 | 7 | 6 | 6 |

[a]AET does not include evapotranspiration caused by human water use, i.e. actual water consumption WCa; [b]Taking into account anthropogenic water use; [c]Satisfied demand from surface waters; [d]Negative values indicate that return flows from irrigation with surface water exceed groundwater abstractions; [e]Total water storage (TWS) of 31 December 2000 minus TWS of 31 December 1970, divided by the number of 30 years.

Table 3. Continental climate forcing variables (T, SWD, LWD, P) and water balance components (AET, Q, WCa). Ensemble mean and min/max deviation from mean (in %, also for T) over all four homogenous forcing variants (GSWP3, PGFv2.1, WFD, WFDEI_hom), for six continental regions and the global total and for the time period 1971-2000.

| | | Africa | Asia | Europe[1] | NAmerica | Oceania | SAmerica | Global |
|---|---|---|---|---|---|---|---|---|
| T [°C] | Mean | 24.1 | 14.6 | -1.6 | 4.2 | 21.7 | 22.1 | 13.6 |
| | Δmin | -0.2 | 0.0 | -5.3 | -3.0 | -0.2 | -0.1 | -0.4 |
| | Δmax | 0.1 | 0.1 | 14.0 | 1.3 | 0.4 | 0.2 | 0.2 |
| SWD [W m$^{-2}$] | Mean | 229 | 196 | 117 | 156 | 229 | 197 | 185 |
| | Δmin | -11.1 | -5.6 | -9.9 | -1.9 | -2.4 | -6.8 | -6.7 |
| | Δmax | 3.9 | 2.9 | 4.5 | 2.3 | 1.2 | 4.4 | 2.5 |
| LWD [W m$^{-2}$] | Mean | 365 | 322 | 266 | 285 | 351 | 381 | 326 |
| | Δmin | -0.2 | -0.7 | -1.0 | -1.2 | -0.9 | 1.4 | -0.3 |
| | Δmax | 0.4 | 1.0 | 0.9 | 1.2 | 0.6 | 1.5 | 0.3 |
| P [km$^3$ yr$^{-1}$] | Mean | 20 457 | 24 501 | 13 026 | 16 177 | 5939 | 28 623 | 108 724 |
| | Δmin | -5.6 | -3.0 | -7.0 | -6.9 | -2.4 | -4.0 | -4.8 |
| | Δmax | 3.9 | 2.6 | 3.7 | 3.7 | 1.9 | 2.1 | 2.1 |
| AET [km$^3$ yr$^{-1}$] | Mean | 16 194 | 13 506 | 6942 | 9573 | 3887 | 17 132 | 67 234 |
| | Δmin | -5.5 | -4.0 | -8.7 | -8.0 | -3 0 | -5.2 | -5.7 |
| | Δmax | 4.7 | 3.7 | 3.7 | 5.4 | 3.2 | 3.4 | 4.0 |
| Q [km$^3$ yr$^{-1}$] | Mean | 4183 | 10 276 | 6138 | 6507 | 2033 | 11 428 | 40 566 |
| | Δmin | -5.9 | -3.6 | -5.0 | -5.4 | -3.4 | -2.2 | -3.4 |
| | Δmax | 10.9 | 4.4 | 4.0 | 3.8 | 5.0 | 2.5 | 4.0 |
| WCa [km$^3$ yr$^{-1}$] | Mean | 71 | 612 | 87 | 121 | 16 | 32 | 939 |
| | Δmin | -5.9 | -1.0 | -8.5 | -5.3 | -2.8 | -2.8 | -2.6 |
| | Δmax | 2.5 | 1.7 | 3.6 | 5.7 | 2.5 | 2.8 | 2.2 |

[1] includes all of Russian Federation

Table 4. Average density of precipitation gauging stations and P sums [$km^3\ yr^{-1}$] for 1971-2000 of the original P data that were used for bias correction (WFD: GPCCv4, WFDEI: GPCCv5, GSWP3: GPCCv6, PGFv2.1: CRU TS3.21) and P outputs of WaterGAP using the undercatch adjusted forcings (except PGFv2.1 which is not adjusted).

| variable | continent | Africa | Asia | Europe | NAmerica | Oceania | SAmerica | global |
|---|---|---|---|---|---|---|---|---|
| Stations per 0.5° grid cell | CRU TS3.21 | 0.12 | 0.09 | 0.06 | 0.12 | 0.17 | 0.06 | 0.09 |
| | GPCCv4 | 0.30 | 0.23 | 0.61 | 0.32 | 1.05 | 0.61 | 0.44 |
| | GPCCv5 | 0.31 | 0.30 | 0.66 | 0.54 | 1.82 | 0.63 | 0.57 |
| | GPCCv6 | 0.31 | 0.32 | 0.68 | 0.60 | 1.85 | 0.71 | 0.60 |
| P totals (without undercatch correction) | CRU TS3.21 | 19 595 | 24 040 | 12 128 | 15 160 | 5958 | 27 611 | 104 492 |
| | GPCCv4 | 19 745 | 24 062 | 11 858 | 15 073 | 5732 | 28 135 | 104 605 |
| | GPCCv5 | 19 729 | 24 044 | 11 852 | 15 095 | 5688 | 28 201 | 104 610 |
| | GPCCv6 | 19 724 | 24 066 | 11 861 | 15 116 | 5694 | 28 085 | 104 546 |
| P totals (WaterGAP) | PGFv2.1 | 19 318 | 23 756 | 12 112 | 15 065 | 5799 | 27 475 | 103 525 |
| | WFD | 21 102 | 24 519 | 13 232 | 16 732 | 5960 | 29 146 | 110 690 |
| | WFDEI_hom | 21 250 | 24 597 | 13 256 | 16 779 | 5945 | 29 223 | 111 050 |
| | GSWP3 | 20 160 | 25 133 | 13 505 | 16 131 | 6053 | 28 649 | 109 631 |

Table 5. Global sums of water balance components for land area [km$^3$ yr$^{-1}$] (except Antarctica, Greenland, and inland sinks) (component numbers as in Table 2) for the model variants and the years 1971-2000, divided in calibrated and non-calibrated grid cells.

| | calibrated regions | | | | | non-calibrated regions | | | | |
|---|---|---|---|---|---|---|---|---|---|---|
| No. | GSWP3 | PGFv2.1 | WFD | WFDEI_hom | WFD_WFDEI | GSWP3 | PGFv2.1 | WFD | WFDEI_hom | WFD_WFDEI |
| 1 | 66 825 | 63 290 | 68 039 | 68 288 | 68 288 | 42 806 | 40 235 | 42 651 | 42 762 | 42 762 |
| 2 | 43 996 | 40 112 | 45 232 | 45 482 | 44 903 | 24 031 | 23 303 | 22 356 | 24 425 | 23 984 |
| 3 | 22 291 | 22 619 | 22 286 | 22 269 | 22 893 | 18 388 | 16 554 | 19 915 | 17 944 | 18 405 |
| 4 | 523 | 546 | 515 | 531 | 523 | 411 | 414 | 400 | 418 | 410 |
| 5 | 582 | 598 | 572 | 594 | 581 | 468 | 473 | 451 | 476 | 463 |
| 6 | -59 | -52 | -58 | -62 | -59 | -57 | -59 | -50 | -58 | -53 |
| 7 | 18 | 15 | 9 | 9 | -28 | -32 | -44 | -29 | -33 | -46 |
| 8 | -3 | -3 | -2 | -3 | -2 | 9 | 8 | 9 | 8 | 8 |

Table 6. Global and continental estimates of WaterGAP water balance components compared to literature values [$km^3 yr^{-1}$]. WaterGAP results are analyzed for the same time span and spatial coverage as the reference and are comparable in terms of precipitation undercatch (see footnotes).

| Source | Coverage | time span | P | | AET | | Q | |
|---|---|---|---|---|---|---|---|---|
| | | | WaterGAP | reference | WaterGAP | reference | WaterGAP | reference |
| Wisser et al. (2010) | global, w G | 1901-1925 | 102 110[a] | 105 298 | 63 319[a,b] | 68 274 | 37 974[a] | 36 888 |
| | | 1926-1950 | 102 653[a] | 105 675 | 63 081[a,b] | 67 826 | 38 837[a] | 37 092 |
| | | 1951-1975 | 105 444[a] | 108 081 | 64 693[a,b] | 68 550 | 39 914[a] | 38 864 |
| | | 1976-2002 | 104 436[a] | 106 764 | 64 337[a,b] | 69 917 | 39 421[a] | 36 813 |
| | | 1901-2002 | 103 676[a] | 106 461 | 63 867[a,b] | 68 480 | 39 044[a] | 37 401 |
| Hanasaki et al. (2010) | global, w G, w A | 1984-1999 | 106 012[a,c] | 113 900 | 64 281[a,b,d] | 72 080 | 40 876[a,e] | 41 820 |
| Rodell et al. (2015) | global, w/o A | 2000-2010 | 113 341[f] | 114 300 | 71 554[b,f] | 70 500 | 41 309[f] | 43 800 |
| | NAmerica, w G | | 17 983[f] | 17 717 | 10 339[b,f] | 9911 | 6604[f] | 7894 |
| | SAmerica | | 29 153[f] | 29 587 | 17 573[b,f] | 17 286 | 11 579[f] | 12 301 |
| | Africa | | 21 323[f] | 20 629 | 17 307[b,f] | 16 809 | 4029[f] | 3820 |
| Müller Schmied et al. (2014) | global, w/o G, w/o A | 1971-2000 | 111 050[g] | 111 070[h] | 69 819[b,g] | 70 576[b,h] | 41 298[g] | 40 458[h] |

[a]PGFv2.1, [b]including WCa, [c]including Antarctica (as 2.1% of global value), [d]including Antarctica (as 4.9% of global value), [e]including Antarctica (as 0.2% of global value, all percentages based on Rodell et al., 2015), [f]WFDEI_hom, [g]WFD_WFDEI, [h]STANDARD model variant; G in column coverage: Greenland, A in column coverage: Antarctica.

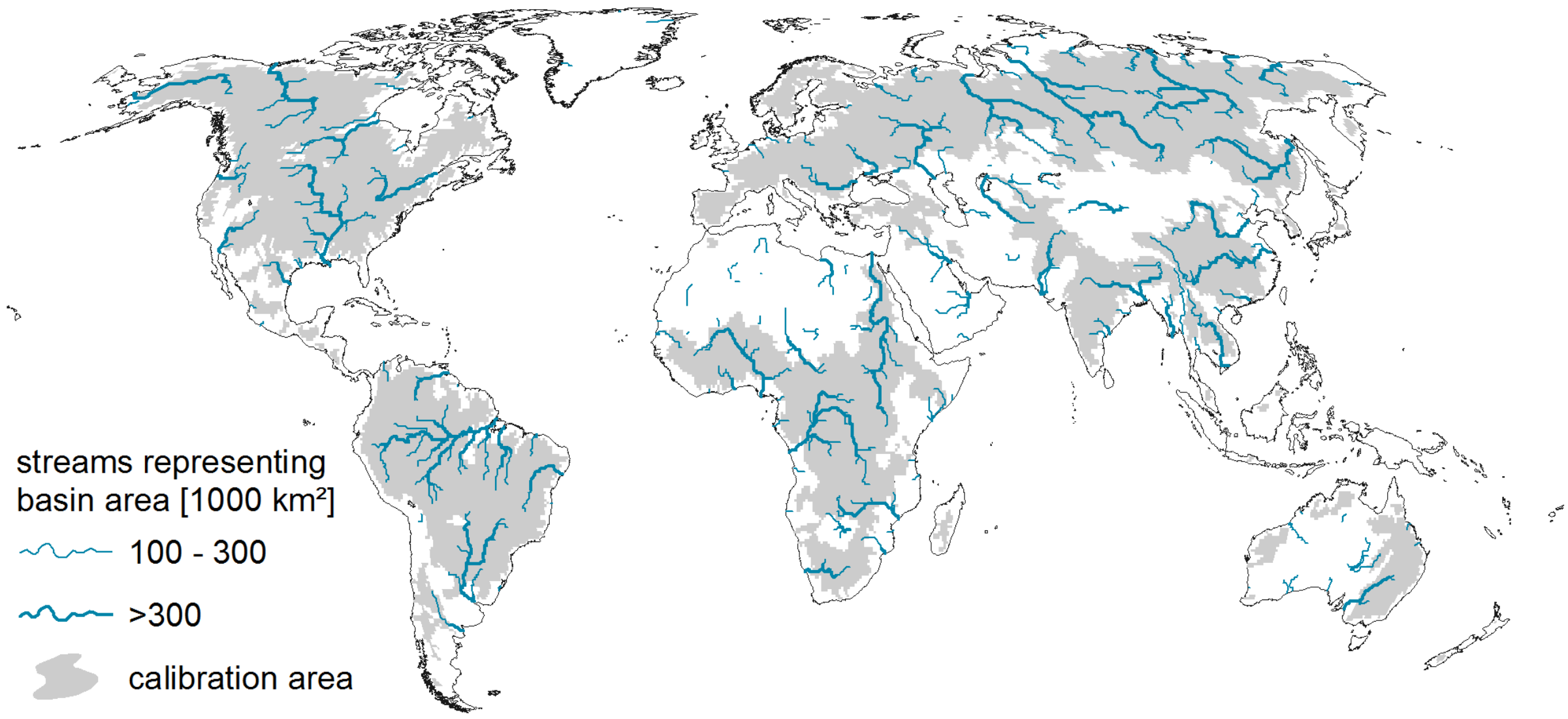

Figure 1. Global land area affected by WaterGAP 2.2 (ISI-MIP 2.1) calibration (grey shading) against observed long-term average river discharge. Streamflow directions and flow accumulation are based on the drainage direction map DDM30 with 0.5° resolution (Döll and Lehner, 2002).

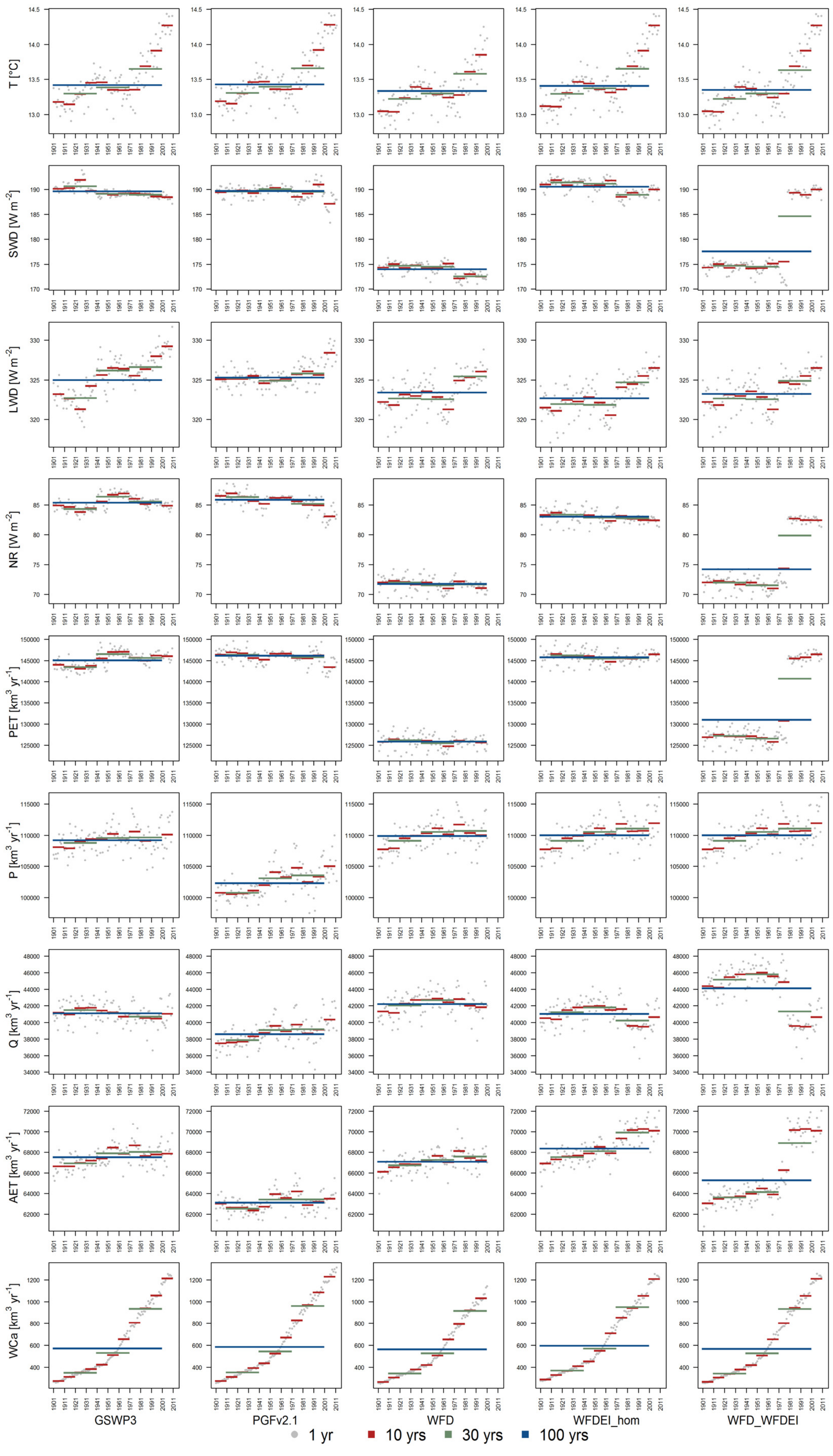

T [°C]
SWD [W m-2]
LWD [W m-2]
NR [W m-2]
PET [km3 yr-1]
P [km3 yr-1]
Q [km3 yr-1]
AET [km3 yr-1]
WCa [km3 yr-1]
GSWP3
PGFv2.1
WFD
WFDEI_hom
WFD_WFDEI
1 yr
10 yrs
30 yrs
100 yrs

Figure 2. Global sums (means) of climatic variables and water balance components for five climate forcings (GSWP3: 1901-2010, PGFv2.1: 1901-2012, WFD: 1901-2001, WFDEI_hom: 1901-2010, WFD_WFDEI: 1901-2010) for different temporal aggregation periods of 1, 10, 30, and 100 years. Displayed are: temperature (T), shortwave downward radiation (SWD), longwave downward radiation (LWD), precipitation (P), discharge into the ocean or inland sinks (Q), actual evapotranspiration (AET) and (actual) water consumption from surface water resources (which could be smaller than the demand, depending on water availability) and groundwater resources (WCa).

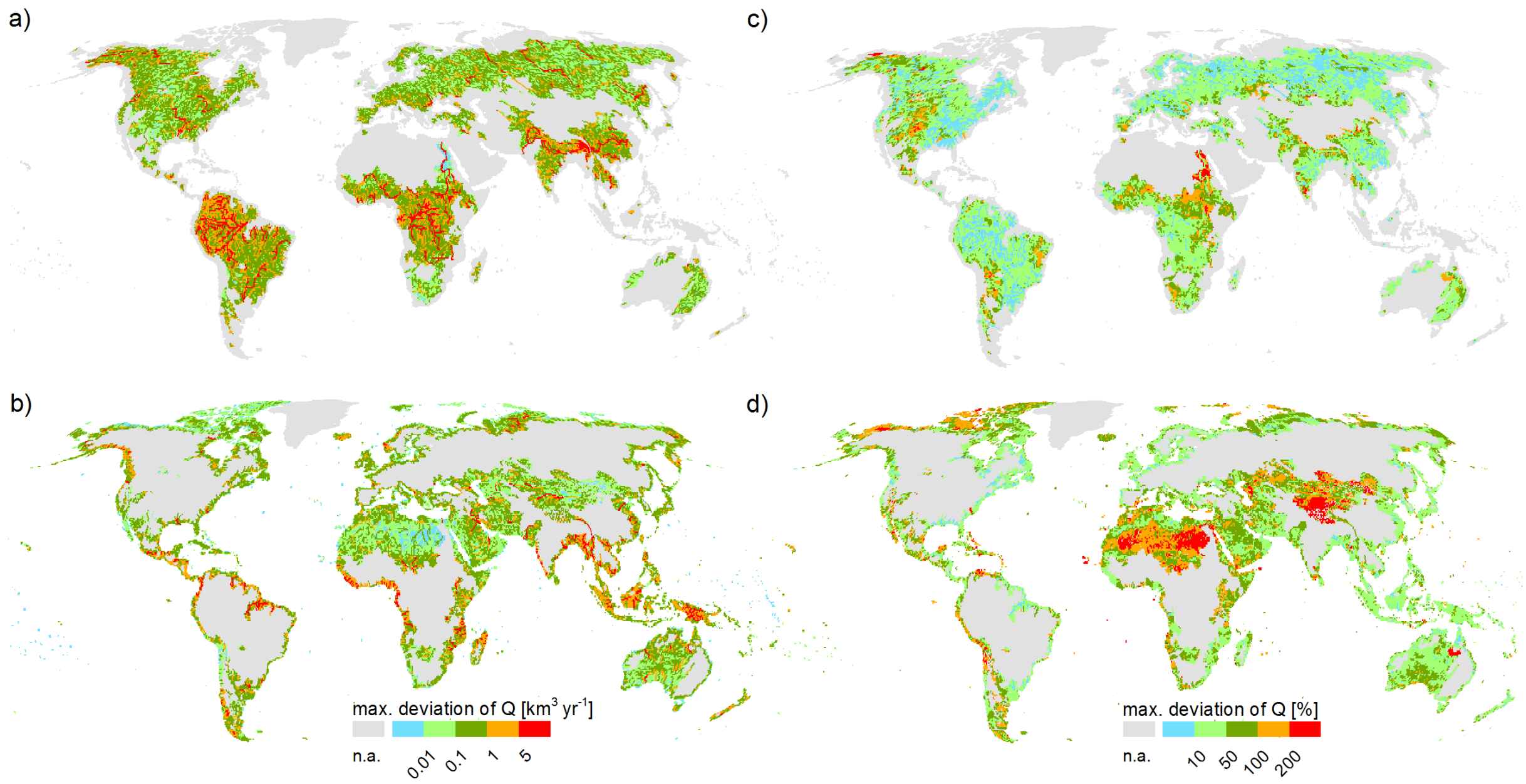

Figure 3. Spatial distribution of the maximum difference of long-term average (1971-2000) Q among the four homogeneous climate forcings (GSWP3, PGFv2.1, WFD, WFDEI_hom), expressed as absolute deviation [$km^3$ $yr^{-1}$] (a, b) and relative deviation (c, d) separately for calibrated (a, c) and non-calibrated (b, d) regions. Grey areas contain either no discharge or are outside the region of interest, i.e. non-calibrated regions are grey in a) and c) and vice versa.

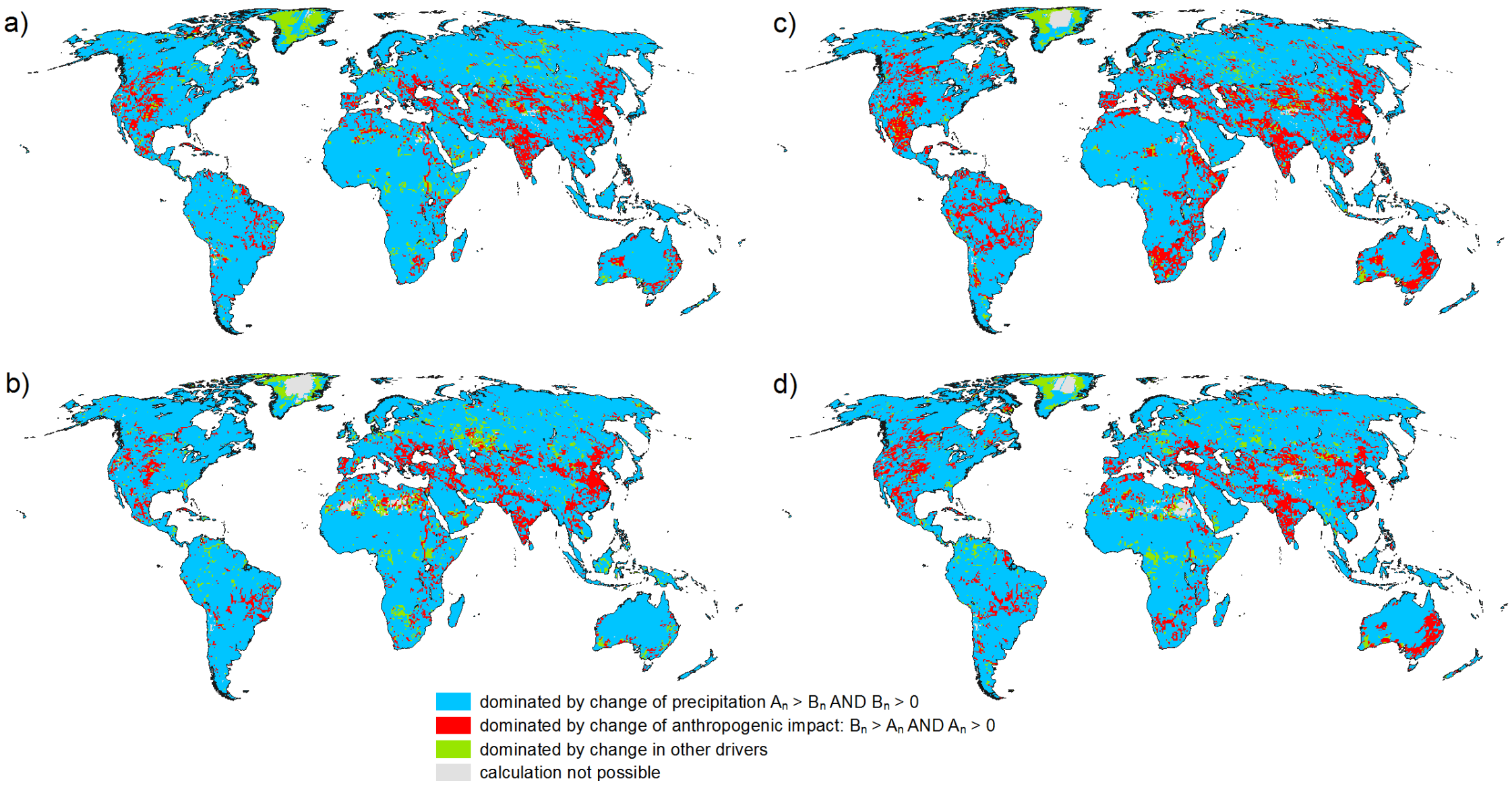

Figure 4. Relative dominance of drivers of change of long-term average Q between 1941-1970 and 1971-2000 (Sect. 2.4.2). Blue indicates that change in P is more dominant than change in anthropogenic impact due to water abstraction and dam construction, red indicates the opposite. In green areas, other drivers are dominant. In grey areas a calculation is not possible as the denominator of indicators $A_n$ and $B_n$ is zero (no change in long-term average Q). Results are shown for WaterGAP as driven by the meteorological forcings GSWP3 (a), PGFv2.1 (b), WFDEI_hom (c) and WFD (d).

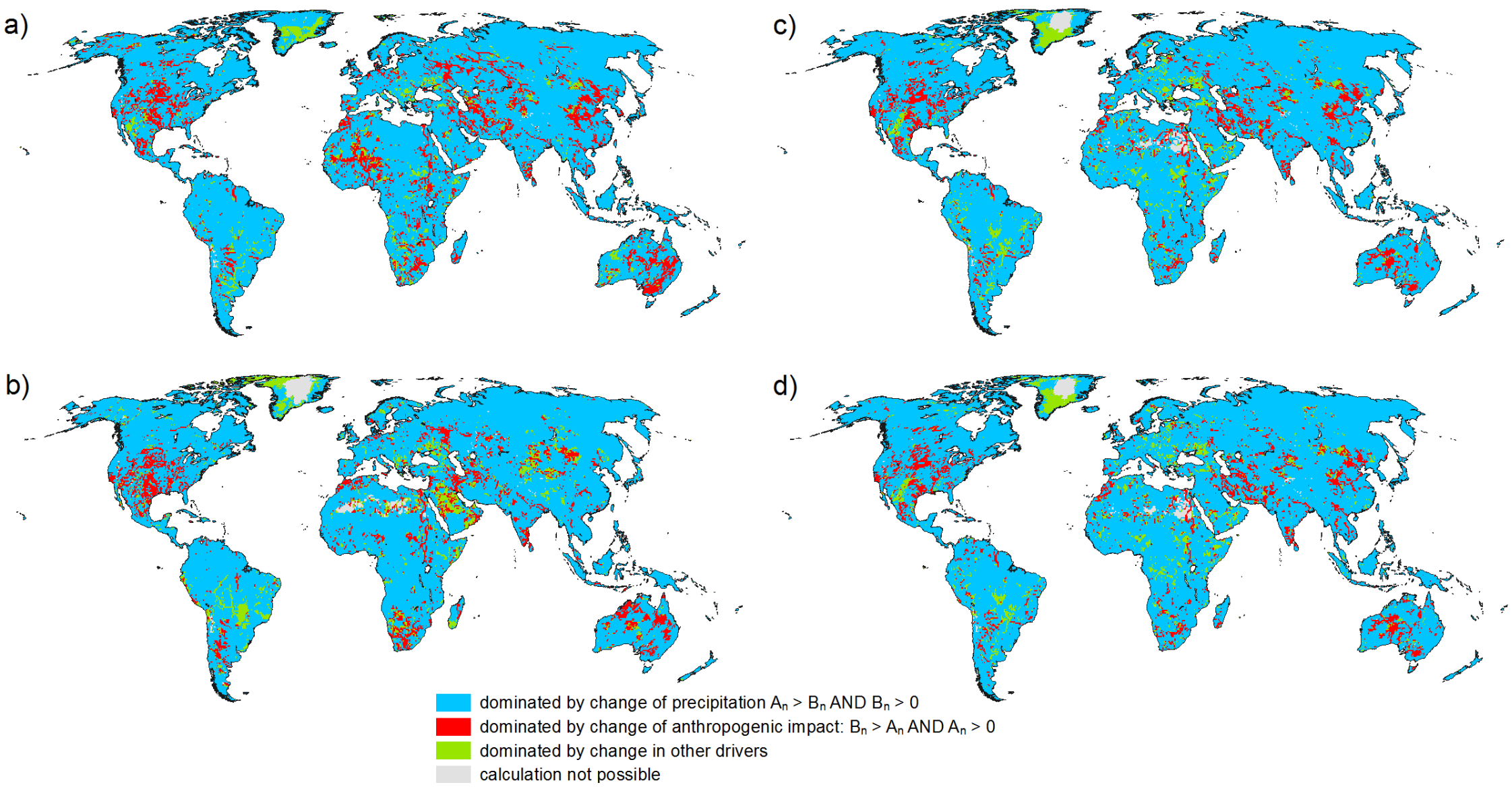

Figure 5. Relative dominance of drivers of change of long-term average Q between 1911-

1940 and 1941-1970 (Sect. 2.4.2). Blue indicates that change in P is more dominant than change in anthropogenic impact due to water abstraction and dam construction, red indicates the opposite. In green areas, other drivers are dominant. In grey areas a calculation is not possible as the denominator of indicators $A_n$ and $B_n$ is zero (no change in long-term average Q). Results are shown for WaterGAP as driven by the meteorological forcings GSWP3 (a), PGFv2.1 (b), WFDEI_hom (c) and WFD (d).