# Peer review of "Variations of global and continental water balance"

_Hydrology and Earth System Sciences, 2015_

## Referee Comment (RC1) · Anonymous Referee #1 · 13 Feb 2016

This m/s used alternative climate forcings in a global hydrological model to determine the resulting uncertainty in water resource estimates, and compares that uncertainty with the estimated impact of human modifications on water resources.

Overall Assessment:

In summary, I am not convinced of the original contribution of this m/s.

The overall methodology seems sound enough and the first 2 questions asked (page 4) appear answered by the analysis. The 3rd question seems ill-formulated within the context of this m/s, as it does not clearly distinguish between real changes in precipitation and apparent changes that are artefacts in the data.
[Figure]

However the concern I have with this m/s is the answers to those first 2 questions are already known and the results unsurprising. Regards (1): from numerous previous studies (including those cited but many more – including high level reports such as from IPCC etc) we already have a fair idea of temporal variations in global water cycle components. Regards (2): we already know that global precipitation analysis in particular diverge considerably and of course that will propagate through a hydrological model. Regards (3): This m/s shows that depending on the precipitation data set chosen we see the human impact on the water cycle less or more clearly, but always in places where we know it exists.

Furthermore, on page 19/20 the authors refer to an earlier study in 2014 that sounds like it had very similar objectives to the present m/s. It is not clear to me what new insights this m/s adds to that previous study, given its title makes explicit reference to the sensitivity of the model to input data.

The above does not mean that the analysis presented cannot be used to draw some interesting new conclusions. In particular, the authors draw attention to the unexpectedly large uncertainty in North America and Europe, which they attribute to undercatch corrections. I thought that was a very interesting finding which could probably be the topic of a m/s in its own right.

Specific comments:

- page 6: It sounds like you essentially treat land use as unchanged during the model period. That has a precedent of course but is still a limitation, pls discuss.

- page 12: This section requires proper statistical treatment, which some type of significance testing, not the type of 'binary' heuristics you use here. For starters, you clearly show that the forcing data are uncertain (not to mention the model) so that error needs to be considered in testing.

---

## Referee Comment (RC2) · Anonymous Referee #2 · 16 Feb 2016

In this manuscript, the authors assessed the variations of global and continental water balance components as impacted by climate forcing uncertainty and human water use. In addition to that, the authors evaluated the impact of climate forcing uncertainty and the effect of temporal aggregations on the different water components at local, regional, and global scales.

The paper addresses relevant scientific questions within the scope of HESS. Although it is not the first paper that evaluates the impacts of human interventions versus hydro-meteorological changes on hydrological parameters, the research angle chosen as well as the approach could be of added value to the research domain of water resources research.
[Figure]

Overall, the paper reaches some interesting conclusions- especially regarding the uncertainty of meteorological forcing data-sets and with respect to the dominant factors of change- backed-up with figures and numbers (results) and substantiated with some existing research (discussion). I would therefore support the manuscript for publication but with substantial revisions taking into account the following general and technical comments/suggestions:

General comments:

1. Locally redundant text, especially when it comes to the introductory parts and descriptions of data in different sections:

a. The introduction of section 2 (Data and Methods) shows much overlap with the last paragraphs of section 1 (Introduction).

b. Page 6, line:2-26: Move the list of modifications to the appendix.

c. Page 16, line:7-9: Here the authors refer to Müller Schmied et al (2015) for a comparison of AET, Q, WCa, and total water storage to the five climate forcings. Since the different climate forcings are already described in this paper, the authors can shorten all text with reference to these climate forcings (section 2.3).

d. De results and discussion section show a lot of overlap. In its current form, the discussion section is basically a repetition of/part of the results section but with some possible explanations for the observations given. Please avoid any unnecessary repetition and try to substantiate/discuss the found results with previous research where possible.

e. The authors use their research questions as titles for the discussion section, please choose shorter title names.

2. The results presented in the results section do not follow an intuitive order. Section 3.1 (temporal aggregations) is not a very logical section to start with. Given the order GCM(-impact)-GHM(-impact)-outputs I would start with the GCM uncertainty discussion (3.1 - make a separate section for this topic), then continue with the human impacts section (3.2), and finally end with the impact of aggregation section (3.3). Results for the global and continental scales could be discussed in sub-sections. Change the order of the research questions and the discussion accordingly.

3. In the discussing the impact of climate forcing uncertainty clearly distinguish between the modeling spread in forcing data (P, T, LWD) and the modelling spread in WaterGAP output (Q, AET, WCa) and elaborate a bit more on the discussion whether the initial spread in forcing data increases/decreases when feeding it into WaterGAP.

Specific/Technical comments:

4. Page 3, line: 13-16: "The international .. data-sparse regions": Would leave this sentence out. Does not add much to the introduction

5. Page 7, line: 5-6: "The initial .. well-being": Incomplete sentence

6. Page 7, line: 24: Please specify (or with reference) why you applied this minimum and maximum catchment area size

7. Page 12, line: 13-14: I would say that -despite A being negative- change in P might still be a significant driver to the change in Q (although not the largest/dominant driver of change). E.g. without a positive change in P, Q might have been even lower than already observed due to human activity.

8. Page 12, line: 21-24: Isn't it technically possible that there is (+/-) change from natural to human whilst there is a change in the other direction when comparing period 1 with period 2? Again, although anthropogenic factors would in that case not be the major driving factor, the influence can still be significant.

9. Page 12, equations: I think it would be good to provide the reader with a map (appendix) that shows the results of the consistency indicators individually. From figures 3 & 4 I cannot deduct for the red/green areas which share of this land area is red/green simply due a negative result for the inconsistency indicator of the other parameter.

10. Page 13, line: 7-8: I had to read this a few times before understanding. With the 0.5 you basically take the mean of the ratio between Q and Precip over period 1 and period 2. Would be more clear if you show something with the SUM/AVERAGE symbol in the equation. Moreover, the line afterwards "assuming that the runoff coefficient remains constant over the two time-periods" is a bit strange in this context: the runoff coefficient is namely estimated using data from the two time-periods.

11. Page 13, line: 7-8: Wouldn't it –w.r.t. to comment 3- be more logical to estimate the $I_{varprec,n}$ with the use of the runoff coefficients ($C_{qp,n}$) under both natural and human conditions in time period 1 and 2 only, rather than combining it again with Q and P?

12. Page 13, line: 7-8: I would say that it is especially this runoff coefficient that is changing due to human interventions. Wouldn't in that sense the change in runoff coefficient be responsible for the share of Q that is impacted by changes in P under a human impacts run?

13. Page 13, line: 15: Would be good to show a figurative example (with numbers) to show how/whether different P and Q scales end up to fall within the same range in $I_{varant}$/$I_{varprec}$.

14. Page 15, line: 22: "1971-2000": The values in the table refer to long-term means Please mention in text.

15. Page 16, line: 7-9: "Müller Schmied .. Table 2": Is this correct? It seems to me that the numbers are swapped. Moreover, could you think of an explanation why the homogeneous forcing performs worse than all forcings (although prob. no significant difference)?

16. Page 16, line: 22: "10.5%": Where does this number come from?

17. Page 17, line: 7: "of": should be "in"

18. Page 17, line: 8-14: This piece of text is a bit fuzzy. Starts with antrophogenic water use, then on to Q, finally back to WCa again. Please reformulate.

[Figure]

19. Page 17, line: 15: "different .. AET": Can you relate AET also with differences in T or Radiation? And how could this possibly influence your irrigation water demand estimates?

20. Page 18, line: 8: "leads" should be "lead"

21. Page 18, line: 24: "More likely" is not equal to "more impotant", please apply terminology consistently

22. Page 19, line: 10-11: "This is.. 20th century": At first glance, this statement seems contradictory to the previous sentence. Please add some text about the changes towards T3 (1971-2000) to clarify this.

23. Page 19, line: 14-15: "The fraction …later period". Had to read this sentence a few times before I understood what was meant. Please clarify what is meant with the later period and specify that with 1911-1940/1941-1970 is mean between/from period T1 to period T2.

24. Page 19, line: 16-21: I thought that the areas "which cannot be calculated"(l:19) and "where both, P and human water use is not the dominant driver"(l:20) are the same areas. But here they have different fractions associated.

25. Page 20, line: 2: "STANDARD": I haven't seen this model reference before whilst you refer to this specific version of model in earlier paragraphs. Please use consistent namings.

26. Page 20, line: 13-16: "In addition … in STANDARD". Incomplete sentence, please reformulate

27. All of section 4.2 and many parts of 4.3/4.4 are results, no discussion. Please reshuffle.

28. Page 21, line: 21-22: "the effect .. time aggregation". Was this specifically evaluated? If yes, where (I cannot find the associated results section).

29. Page 23, line: 26: Would dam construction indeed lead to significant decreases in long-term Q? I would think that dam construction would mainly influence the timing of runoff peaks/lows. Could you give a reference for this statement?

30. Page 24, line: 2:8: "For example... "anthropogenic effects": Do you expect that under the different climate forcings the absolute value of the anthropogenic impacts (mainly irrigation I would say) also changes or is this difference in outcome (relative contribution) only determined by changes in P)

31. Page 25, line: 20-21: "e.g. ISI-MIP2.1". Could you mention more model intercomparison projects? Think of Agmip, Earth2Observe.

―――――――――――――――

---

## Referee Comment (RC3) · Anonymous Referee #3 · 26 Feb 2016

This manuscript presents a quantitatively useful update on our ability to marry terrestrial models of water and energy balance—their uncertain physical processes and tuning parameters, with records of hydrometeorological forcing that are fragmentary and have significant time-dependent biases. These forcing biases result from an amalgam of diverse satellite and in situ data as well as inputs from atmospheric reanalyses. The analysis presented here covers time scales from annual to century and spatial scales from 0.50 degree lat/lon grids to global land means. While the formalism here is a largely a standard approach there are two specific aspects I feel are noteworthy:

1) The first is the identification of how specific differences in forcing data propagate through the water balance and, by virtue of tuning / calibration on observed discharge,

affect gauged and ungauged contributions differently. Both the low precipitation in the PGFv2 data (as a result primarily of no snowfall undercatch correction) and the low values of downward SW radiation in the WATCH forcing (resulting from the older NCEP reanalysis values) are cases in point. In gauged regions, discharge (Q) is controlled by calibration and so AET responds to the differences in precipitation and radiative forcing. (Note the large spread in global AET in Table 3, focused in Europe and N. America where snowfall is significant.) Conversely, in ungauged regions the resulting uncertainty from forcing data sets in discharge is over 18% (Table 2). These results quantify the effects of forcing data quality and (un)availability on regional and global results.

2) The other noteworthy aspect of this paper is the attempt to identify the relative roles of climate forcing versus anthropogenic effects on changing water balance. Diagnostic indicators involving the relative changes in P versus Q and changes in the actual versus "naturalized" P and Q changes are considered. The resulting maps (Figs 3,4) are valuable, I think, not only for their consistency but also their differences between forcing data sets. One is not surprised to see the western US and Europe exhibit significant increases in diversions /extractions through the middle of the 20th Century. The growth of these impacts throughout southern Asia in the second half of the 20th Century is reasonably consistent across the four homogeneous data sets. Nevertheless there are significant regional differences in the estimated growth of anthropogenic effects—Australia, China and Mexico are discussed as examples where the forcing data set differences have significant interpretive consequences.

So, basically, my sense is that this paper is a valuable assessment of where our diagnostic modeling capabilities for water balance stand. That said, there are some aspects of the presentation that need improvement:

i. The discussion in section 2.4.2 on the construction of the indicators for anthropogenic effects was difficult to follow. The reasoning behind An and Bn seems clear enough, but I had a difficult time trying to understand how An and Bn were incorporated in Figs

3 and 4. What combinations of $A_n > 0$, $B_n < 0$ and $I_{varpredoc,n} > 0$ make up red or blue areas in those Figs? Presumably $A_n$ and $B_n$ are of opposite sign in both blue and read areas? Perhaps some schematic picture would be useful.

ii. There were numerous places in the manuscript where I was unsure as to what Figures or Tables the discussion related to. Does sevtion4.2 refer to the information in Fig 2? Does the discussion in section 4.3 relate to Table 2 and 3? Does section 4.4 refer to Figures 3 and 4? Alluding to the appropriate graphic needs to be added.

iii. I don't really see a lot of value of section 4.1. It largely discusses differences with the standard version WaterGAP 2.2. Perhaps mentioning earlier estimates from papers such as Oki and Kanae (2006), Haddeland et al, (2011) and Rodell et al, (2015) would give some context outside the WaterGAP model.

I recommend accpetance after attention to these these three areas of concern.

———————————

---

## Author Comment (AC1) · 17 Mar 2016

**Authors' response to referee comments and planned modification of the manuscript**

First of all, we are grateful to the referees for their suggestions. We would like to answer each comment by providing first the referee's comment (normal font, introduced by RC#x), the author's response (italic font, introduced by AR) and the possible modification of the manuscript (bold font, introduced by MM).

General statement of possible modifications of the manuscript: During answering the referee's comments, we have read through the manuscript carefully and found several parts of the manuscript where text can be improved for a better readability and motivation of the study. Esp. referee#2 asked to reduce the overlap of the results and discussion section. We plan to join the results and discussion section as then the overlap is minimized and the aspects can be discussed directly after the results. This also improves in our opinion the overall structure of the manuscripts. This would lead to major changes in that (new) section 3, also due to a modified order of the first two research questions and subsequent re-organized Subsections. In addition, we found out that we have wrongly displayed runoff and not discharge in Fig. 5 – hence we would update this Figure and the belonging numbers in the text. We would also add global assessments of net radiation and potential evapotranspiration to Fig. 2 as we can better explain differences among the water balance components of the model variants.

Please note also the changes which we would do due to a reported error in PGFv2 climate forcing (see the extra author's answer on this).

RC#1 This m/s used alternative climate forcings in a global hydrological model to determine the resulting uncertainty in water resource estimates, and compares that uncertainty with the estimated impact of human modifications on water resources.

*AR: Thank you for the nice summary the manuscript.*

**MM: none**

RC#1: In summary, I am not convinced of the original contribution of this m/s.

The overall methodology seems sound enough and the first 2 questions asked (page 4) appear answered by the analysis. The 3rd question seems ill-formulated within the context of this m/s, as it does not clearly distinguish between real changes in precipitation and apparent changes that are artefacts in the data.

*AR:. You are correct that we do not distinguish between real temporal changes and apparent changes that are artefacts of the data. We actually are not in the position to do so as users of forcing data that have been produced by various data producers. This problem also concerns research questions 1 and 2, and is, in our opinion outside the scope of the manuscript. With the analysis of the (dominant) impact of precipitation changes or human impact on the temporal development of river discharge research question 3) we do a new combined analysis for distinguish the precipitation changes and human impact taking into account the uncertainty of precipitation data; this in our opinion fits well into the context of the manuscript (compare the title of the manuscript). Your comment, however, reminded us that we should mention in the manuscript that monthly precipitation observations used in producing the applied precipitation data sets are heterogeneous in time, i.e. that the number of stations is not constant over time, which makes them less suitable for the analysis of temporal trends. The applied GPCC and CRU monthly precipitation data sets are optimized for best spatial coverage.*

**MM: We would add the following sentence to section 2.3 (as last line of introductory paragraph):**

**"In all data sets, daily precipitation estimates were obtained by bias-correcting output of weather models by monthly precipitation data sets that had been derived from monthly precipitation observed at gauging stations. These monthly data sets were optimized for spatial coverage, i.e. using, for each month, the available number of gauging stations. The temporally variable number of considered precipitation observations makes the applied precipitation data sets less suitable for the analysis of temporal variations. While a homogeneous data set of observation-based monthly precipitation exists at least for the time period 1950-2000, it is based on less than 10 000 gauging stations and therefore provides a spatially less accurate representation of global-scale precipitation (Beck et al. , 2005) than the data sets used in this study, which include up to 50 000 gauging stations (Schneider et al., 2015)."**

RC#1: However the concern I have with this m/s is the answers to those first 2 questions are already known and the results unsurprising. Regards (1): from numerous previous studies (including those cited but many more – including high level reports such as from IPCC etc) we already have a fair idea of temporal variations in global water cycle components.

*AR: Indeed, previous work was done to assess temporal variations in global water components, and we certainly have missed a study when preparing the first submission. Wisser et al. (2010) evaluated temporal variation in 25yr steps of modeled global water balance components (their Table 2). They used a model approach, CRU TS 2.1 as basis data and some precipitation products for the uncertainty analysis. In the submitted manuscript, we present a consistent assessment of more than a century with different temporal aggregation steps for 4 + 1 state-of-the-art climate forcings (with more than varying precipitation variable) that are used frequently in global hydrological modeling and this was, to our knowledge, not done until now in this comprehensive manner. Most other previous work considered specific time steps (e.g. 1961-1990). Regarding knowledge assessed in the IPCC reports (mainly Chapter 2 in the WG I report 2013), the focus is on long-term trends, while we present different temporal aggregations. And while IPCC considers knowledge on each water balance component separately, we wish to show a consistent picture of the temporal development of all components.*

**MM: We would refer to the study of Wisser et al. (2010) in the introduction.**

**P4, line 17 would now read as: In addition to these uncertainties, water resources estimates differ due to different reference periods (Wisser et al., 2010).**

**And we would relate our results regarding the long-term trend of Q to IPCC results.**

**P15, line 10 would read as: The results of this study confirm the finding of the IPCC Fifth Assessment Report that "the most recent and most comprehensive analyses of river runoff do not support the IPCC Fourth Assessment Report (AR4) conclusion that global runoff has increased during the 20th century." (Stocker et al., 2013, p. 44)**

RC#1: Regards (2): we already know that global precipitation analysis in particular diverge considerably and of course that will propagate through a hydrological model.

*AR: There are studies available showing differences e.g. in precipitation products (e.g. Biemans et al., 2009; Voisin et al., 2008; Wisser et al., 2010) and also the correspondent changes in river discharge, but they mainly stay at the level of different precipitation data. To our knowledge, the information we*

present in Table 4 is novel as it compiles for the precipitation products the station density as well as the effect of undercatch correction on continental and global level. In addition, we did not only consider precipitation, but also other climate variables, which can have a huge impact on model results (e.g. short-wave radiation, see the combination of WFD and WFDEI) and may overwhelm the effect of different precipitation products.

**MM: We would add the reference to Voisin et al., 2008 to the manuscript and modify the sentence slightly. P4, l11 would read now: Simulated water balance components vary considerably due to various uncertainties of GHMs (Haddeland et al., 2011; Schewe et al., 2014) including GHM ability to include human water use, model improvements over time (e.g. see the different results of the Water Global Assessment and Prognosis (WaterGAP) model in Müller Schmied et al. (2014), their Table 5), climate forcing (Biemans et al., 2009, Voisin et al., 2008) as well as uncertainties in discharge observations (Coxon et al., 2015; McMillan et al., 2012).**

RC#1: Regards (3): This m/s shows that depending on the precipitation data set chosen we see the human impact on the water cycle less or more clearly, but always in places where we know it exists.

*AR: We agree in the sense that our analysis is based on data on irrigated areas and dams (where we know they exist). The dominance patterns in Figure 4 obviously reflect these data, but they are more complex than the underlying data (and partially surprising). It is true that precipitation is one driver for this method and we used precipitation as it is has probably the key role, but for the human impact indicator $I_{\mathrm{varant}}$, we are using river discharge, where precipitation is not the only driver (but also radiation and temperature). The attempt was not to present new regions of human impact on water cycle, but to distinguish between climatic (data) changes and (simulated) human impact. We show this assessment for two time spans and four forcings which gives an idea of the impact of forcing (which can be huge) as well as the considered time span(s). We believe that this assessment can be of value for future indicators / assessments to not stay with one forcing.*

*Maybe the previous comment "seems ill-formulated" regarding the 3rd question refers to the detailed research question. As our modeling approach is not able to e.g. reflect human changes in land use, we cannot simplify the question to " – either change of precipitation or change of human impact".*

**MM: none**

RC#1: Furthermore, on page 19/20 the authors refer to an earlier study in 2014 that sounds like it had very similar objectives to the present m/s. It is not clear to me what new insights this m/s adds to that previous study, given its title makes explicit reference to the sensitivity of the model to input data.

*AR: This section is also criticized by the other Referees, and would be revised. This manuscript is referring to the sensitivity of input data to model results a follow on to Müller Schmied et al. (2014) (referred from now on as MS14). MS14 evaluated the sensitivity of climate forcing (comparing two forcings, which one of them (WFD_WFDEI) is used here also), land cover (two land cover inputs), human water use (included, not included), model structure (two model versions) and calibration (or no calibration). MS14 is more general and showed that calibration has the largest effect, followed by climate forcing. In this manuscript, we are focusing on the effect of climate forcing (4 state-of-the-art forcings), applied a methodology for improving the combination of two timely consecutive forcings (WFD, WFDEI) plus applied the forcings to quantify the dominance indicator. Furthermore, the assessment of the used climate forcings is of interest for the many modeling groups involved in the Inter-Sectoral Impact Model Intercomparison (ISI-MIP) activities.*

**MM: We would join results and discussion section and rewrite most of the content in the new section 3.1. which would be called "Water balance components as impacted by climate forcing uncertainty", also including now a new Table with comparisons to Wisser et al. (2010), Hanasaki et al. (2010) and Rodell et al. (2015).**

RC#1: The above does not mean that the analysis presented cannot be used to draw some interesting new conclusions. In particular, the authors draw attention to the unexpectedly large uncertainty in North America and Europe, which they attribute to undercatch corrections. I thought that was a very interesting finding which could probably be the topic of a m/s in its own right.

*AR: Thank you for this statement and suggestion. We agree that assessing undercatch correction (or not) in a thorough way should be done in a separate study. In our manuscript we highlight the impact of precipitation undercatch (e.g. P23, l1; P23, l6), and in the revised manuscript we added a sentence in the conclusions that can be understood as a call in this direction.*

**MM: In the revised manuscript, we would add a sentence on P25, l24:**

**A thorough analysis of available precipitation undercatch correction methods and development of an improved method for correcting the global state-of-the-art precipitation products, by building on the work of Fuchs et al. (2001), would enable a better quantification of global precipitation.**

RC#1: page 6: It sounds like you essentially treat land use as unchanged during the model period. That has a precedent of course but is still a limitation, pls discuss.

*AR: You are absolutely right. Land use / land cover is changing in reality within the model time period (20th century and early 21st century), for example because of human activities. Within the WaterGAP model we do not consider dynamic land use as cropland or pasture (which could be e.g. covered using HYDE database (Klein Goldewijk et al. 2011)), but having the IGBP classification of land cover types included (see MS14, Table A1 and A2). To our knowledge, there is no data source available that reflects IGBP classification of the 20th century at the required spatial resolution. Since MODIS satellites are in orbit (since the beginning of the 21st century), remote sensing based land cover classification can be done, and is included in the model as a snapshot. Currently, we do not see an advantage to change year-by-year the land cover e.g. based on MODIS, as this reflects only the last ~15 years. In addition, research has to be done to include e.g. yearly changed land cover classification in a consistent manner. To change, for example, land cover at each 1st of January can be problematic e.g. for modeled leaf area index and rooting depths, which could then result in jumps of water storages (e.g. in canopy or soil).*

**MM: none, as we do not feel that a discussion of this model assumption would not fit into the scope of the manuscript.**

RC#1: page 12: This section requires proper statistical treatment, which some type of significance testing, not the type of 'binary' heuristics you use here. For starters, you clearly show that the forcing data are uncertain (not to mention the model) so that error needs to be considered in testing.

*AR: We think that given the large but unknown uncertainties (e.g. regarding the existence of irrigated areas, or of net radiation), it is not possible to quantify errors in a meaningful way and to do significance testing.*

**MM: none**

RC#2: In this manuscript, the authors assessed the variations of global and continental water balance components as impacted by climate forcing uncertainty and human water use. In addition to that, the authors evaluated the impact of climate forcing uncertainty and the effect of temporal aggregations on the different water components at local, regional, and global scales.

The paper addresses relevant scientific questions within the scope of HESS. Although it is not the first paper that evaluates the impacts of human interventions versus hydrometeorological changes on hydrological parameters, the research angle chosen as well as the approach could be of added value to the research domain of water resources research.

Overall, the paper reaches some interesting conclusions- especially regarding the uncertainty of meteorological forcing data-sets and with respect to the dominant factors of change- backed-up with figures and numbers (results) and substantiated with some existing research (discussion). I would therefore support the manuscript for publication but with substantial revisions taking into account the following general and technical comments/suggestions.

*AR: Thank you for the nice summary and the general evaluation.*

RC#2: 1. Locally redundant text, especially when it comes to the introductory parts and descriptions of data in different sections:

*AR: Thank you for pointing out redundancy so clearly.*

RC#2: a. The introduction of section 2 (Data and Methods) shows much overlap with the last paragraphs of section 1 (Introduction).

*AR: We have modified the last paragraph of the introduction.*

**MM: P5, first sentence would now read as: To answer these questions, we conducted a modeling experiment.**

RC#2: b. Page 6, line:2-26: Move the list of modifications to the appendix.

*AR: Thanks, done as suggested.*

**MM: We would move the bullet points at P6 and P7 to the appendix, and the sentence at P6,l2 would read now as: "The version used for this study is named WaterGAP 2.2 (ISI-MIP 2.1), and differences to WaterGAP 2.2, mainly done to fulfil requirements of the ISI-MIP project phase 2.1 are described in the Appendix A." In addition, we would add a bullet point that describes differences in the calibration setup for WFDEI_hom forcing: "For WaterGAP calibration, we used observed streamflow data from 30. For GSWP3, PGFv2 and WFD, we used data from 1971-2000 if available for the time period. Due to the offset in radiation of WFD_WFDEI forcing (and consequences for model results, see Müller Schmied et al., 2014), we calibrated WFDEI_hom using using preferably the period 1980-2009 and used these calibration parameters for the WFD_WFDEI simulation." We would also reformulate the bullet point regarding the reduction factor of lake evaporation to: "For lakes, reduction of evaporation due to decreasing lake area is calculated according to Eq. 1 in Hunger and**

**Döll (2008), resulting in a lower but more realistic lake area and thus evaporation reduction with decreasing lake storage."**

RC#2:c. Page 16, line:7-9: Here the authors refer to Müller Schmied et al (2015) for a comparison of AET, Q, WCa, and total water storage to the five climate forcings. Since the different climate forcings are already described in this paper, the authors can shorten all text with reference to these climate forcings (section 2.3).

*AR: Thanks for this suggestion. In the submitted conference proceeding, we are referring to this HESSD-manuscript and providing only one table (Table 1) with the main characteristics of the climate forcings.*

**MM: As we now feel that this sentence does not fit to the content of this paragraph, we would remove it.**

RC#2:d. The results and discussion section show a lot of overlap. In its current form, the discussion section is basically a repetition of/part of the results section but with some possible explanations for the observations given. Please avoid any unnecessary repetition and try to substantiate/discuss the found results with previous research where possible.

*AC: When separating results and discussion, at least some overlap cannot be avoided. We would join the results and discussion in order to reduce the overlap and included some more references.*

**MM: The results section would be now structured as follows: Sect 3.1: "Water balance components as impacted by climate forcing uncertainty" with subsections 3.1.1 Uncertainty of global climate forcings, 3.1.2 Uncertainty of simulated water balance components due to climate forcing uncertainty and 3.1.3 Comparison with other studies. This section would include current sections 3.2, 3.3, 4.1 and 4.3 (answering research question 2, now 1). Sect. 3.2 would be now: "Temporal variation of global water balance components for different temporal aggregations" (includes sections 3.1, 4.2 and answering research question 1 and 2). Sect. 3.3 would be now: "Dominant drivers of temporal variations of 30-year mean annual river discharge: precipitation or human water use and dam construction" (includes sections 3.4, 4.4 and answering research question 3). We additionally would re-arrange the research questions (interchanged the position of 1 and 2). Consequently, we would also modify the conclusion section according to the rearranged research questions. We feel that this revising would improve the readability and shape of the manuscript significantly.**

RC#2: e. The authors use their research questions as titles for the discussion section, please choose shorter title names.

*AR: We intended to allow a quick search for the answers to the research questions and have chosen therefore the research questions as titles for the discussion section. Anyhow, we would try to find titles which match to the research questions in the newly joined results and discussion section.*

**MM: Section title 3.1 would now reads as: "Water balance components for the time period 1971-2000 as impacted by climate forcing uncertainty". Section title 3.2 would now reads as: "Variation of estimated global water balance components for different temporal aggregations". Section title 3.3 would now reads as: "Dominant drivers of temporal variations of 30-year mean annual river discharge: precipitation or human water use and dam construction".**

RC#2: 2. The results presented in the results section do not follow an intuitive order. Section 3.1 (temporal aggregations) is not a very logical section to start with. Given the order GCM(-impact)-GHM(-impact)-outputs I would start with the GCM uncertainty discussion (3.1 - make a separate section for this topic), then continue with the human impacts section (3.2), and finally end with the impact of aggregation section (3.3). Results for the global and continental scales could be discussed in sub-sections. Change the order of the research questions and the discussion accordingly.

*AR: Thank you for this suggestion. We do not have included any GCM (if Global Climate Models are meant) in this study, but we believe that you mentioned the forcing data uncertainty. We do not agree that the human impacts section should be before the global and continental scale as well as the temporal variation section as it is affected by the temporal variation. But we agree that it might be better to firstly describe the global and continental scale first, followed by temporal aggregation afterwards. In order to separate global and continental assessment we will stay with the subdivision in sections.*

**MM: We would modify the order or research questions (changed position of 1 and 2) and are would start the results and discussion section with describing the uncertainty of climate forcing to model results, followed by a section about the effect of temporal variations and finally the human impacts.**

RC#2: 3. In the discussing the impact of climate forcing uncertainty clearly distinguish between the modeling spread in forcing data (P, T, LWD) and the modelling spread in WaterGAP output (Q, AET, WCa) and elaborate a bit more on the discussion whether the initial spread in forcing data increases/decreases when feeding it into WaterGAP.

*AR: Thank you for pointing out that we need to separate both uncertainties. Regarding the second point (if initial spread in forcings increases/decreases within WaterGAP), WaterGAP reduces initial spread of the forcings due to the calibration approach. Using observed river discharge for ~54% of the land surface, the model balances out spread in forcing data (see Table 2, e.g. row 2 (river discharge) for calibration regions), but for non-calibration regions, spread is basically transformed to model outputs. We have described this using percentage of spread at several positions in the manuscript, e.g. at P2, l19, P25, l3, Table 2). As translation of climate forcing (and its spread) into model outputs is non-linear, we feel that it is not possible to further discuss that point on a solid basis.*

**MM: As written in earlier answers, we would combine the results and discussion section and followed in the new Sect. 3.1 the order 1) climate forcing uncertainty, 2) impact of climate forcing uncertainty to model outputs, 3) comparison to other estimates. We would be consistently going from global to continental to calibration/non calibration regions and to grid cell scale, so we hope the reader is now guided better through the results section. In addition, we would add climate forcing variables to the continental assessment in Table 3.**

RC#2: Specific/Technical comments:

4. Page 3, line: 13-16: "The international .. data-sparse regions": Would leave this sentence out. Does not add much to the introduction

*AR: Thanks for the suggestion.*

**MM: we would delete this sentence.**

RC#2: 5. Page 7, line: 5-6: "The initial .. well-being": Incomplete sentence

*AR: Thanks, we would modify the sentence*

**MM: This sentence would read now as: "The purpose of WaterGAP has been to provide a best estimate of renewable water resources worldwide."**

RC#2: 6. Page 7, line: 24: Please specify (or with reference) why you applied this minimum and maximum catchment area size

*AR: both numbers are minimum sizes. The 9000 km² are minimum upstream area which corresponds to at least 3 grid cells at the Equator as minimum. In many cases, more than one discharge observation station is located within a basin. With the 30000 km² minimum area between two calibration stations at one basin, we want to allow a fair chance of WaterGAP calibration to be efficient. This was introduced by Hunger and Döll 2008, and we have increased the interstation area in MS14.*

**MM: We would modified the sentence to: "Observation stations were selected such that the upstream area had a minimum of 9 000 km². To avoid including stations that are located very close to each other along a river, the minimum interstation catchment area was set to 30 000 km². Furthermore, a station was selected only if a minimum of four complete years of data was available."**

RC#2: 7. Page 12, line: 13-14: I would say that -despite A being negative- change in P might still be a significant driver to the change in Q (although not the largest/dominant driver of change). E.g. without a positive change in P, Q might have been even lower than already observed due to human activity.

*AR: Thank you for pointing this out, you are right, P can still be significant. However, it cannot be the dominant driver according to our approach.*

**MM: We would change the word "significant" to "dominant" in this sentence.**

RC#2: 8. Page 12, line: 21-24: Isn't it technically possible that there is (+/-) change from natural to human whilst there is a change in the other direction when comparing period 1 with period 2? Again, although anthropogenic factors would in that case not be the major driving factor, the influence can still be significant.

*AR: Same answer to previous comment.*

**MM: We would change the word "significant" to "dominant" in this sentence.**

RC#2: 9. Page 12, equations: I think it would be good to provide the reader with a map (appendix) that shows the results of the consistency indicators individually. From figures 3 & 4 I cannot deduct for the red/green areas which share of this land area is red/green simply due a negative result for the inconsistency indicator of the other parameter.

*AR: Thanks for this suggestion.*

**MM: We would include the maps of An and Bn for the four forcings and the two time steps into the Appendix B. We found out that we interchanged the figure caption of Fig. 3 and 4. c) is WFDEI_hom and d) is WFD. We would update the positions in the text where it is necessary and revise Figure**

**captions in order to improve readability. In addition (and according to the Referee#3 we would reformulate and re-structure the section 2.4.2 and add a more descriptive legend into Figs 3 and 4.**

RC#2: 10. Page 13, line: 7-8: I had to read this a few times before understanding. With the 0.5 you basically take the mean of the ratio between Q and Precip over period 1 and period 2. Would be more clear if you show something with the SUM/AVERAGE symbol in the equation. Moreover, the line afterwards "assuming that the runoff coefficient remains constant over the two time-periods" is a bit strange in this context: the runoff coefficient is namely estimated using data from the two time-periods.

*AR: Thanks for the suggestion regarding the equation. You are right, runoff coefficient is the mean between two periods, but is then applied for both periods (and thus assumed to be constant throughout the time).*

**MM: We would reorganize this section and modify P13, line 6 to: "…calculated as the mean runoff coefficient of the two periods under consideration.". In addition, we would modify P 13 line 8 to: "…and assumed this value for the two time periods."**

RC#2:11. Page 13, line: 7-8: Wouldn't it –w.r.t. to comment 3- be more logical to estimate the Ivarprec,n with the use of the runoff coefficients (Cqp,n) under both natural and human conditions in time period 1 and 2 only, rather than combining it again with Q and P?

*AR: One beauty of the indicators is their absolute units. If we would use only runoff coefficients, we cannot compute an absolute Ivarprecdom,n indicator.*

**MM: none**

RC#2: 12. Page 13, line: 7-8: I would say that it is especially this runoff coefficient that is changing due to human interventions. Wouldn't in that sense the change in runoff coefficient be responsible for the share of Q that is impacted by changes in P under a human impacts run?

*AR: The runoff coefficient can be influenced by both, climate variation or human interventions. To exclude human interventions, we are calculating the runoff coefficient by using the naturalized river discharge. Taking the change of runoff coefficient as indicator of human change is an interesting thought, but we will stay at the current form of usage due to the absolute unit.*

**MM: none**

RC#2: 13. Page 13, line: 15: Would be good to show a figurative example (with numbers) to show how/whether different P and Q scales end up to fall within the same range in Ivarant/Ivarprec.

*AR: Thank you for this suggestion.*

**MM: We would add a new Table 1 showing examples of four grid cells covering four cases of the GSWP3 forcing for the time steps 1941-1970 and 1971-2000. Section 2.4.2 would end now with: "To illustrate the indicator of relative dominance approach, Table 1 lists indicator values and underlying data for the example of four grid cells representing discharge of large rivers near the outlet to the ocean."**

RC#2: 14. Page 15, line: 22: "1971-2000": The values in the table refer to long-term means. Please mention in text.

*AR: Thanks for this suggestion*

**MM: After restructuring the results and discussion section, we feel it would be now not necessary to introduce the tables in the text. From the Table caption, it should be clear that long-term means are meant.**

RC#2: 15. Page 16, line: 7-9: "Müller Schmied .. Table 2": Is this correct? It seems to me that the numbers are swapped. Moreover, could you think of an explanation why the homogeneous forcing performs worse than all forcings (although prob. no significant difference)?

*AR: Thanks, as Müller Schmied et al., 2015 is not yet available for the reader and to avoid confusion, we have deleted this sentence.*

**MM: We would delete this sentence.**

RC#2: 16. Page 16, line: 22: "10.5%": Where does this number come from?

*AR: We calculated the percentage difference in this case as (max(GSWP3, WFD, WFDEI_hom) – min(GSWP3, WFD, WFDEI_hom)) / mean(GSWP3, WFD, WFDEI_hom) * 100 (same methodology as described in P16, l13).*

**MM: none, calculation method already provided in P16, l13.**

RC#2: 17. Page 17, line: 7: "of": should be "in"

*AR: Thanks!*

**MM: fixed**

RC#2: 18. Page 17, line: 8-14: This piece of text is a bit fuzzy. Starts with anthropogenic water use, then on to Q, finally back to WCa again. Please reformulate.

*AR: Thank you.*

**MM: We would reformulate this in the new section 3.1 and tried to avoid such fuzzy paragraphs by separating the variable into paragraphs.**

RC#2: 19. Page 17, line: 15: "different .. AET": Can you relate AET also with differences in T or Radiation? And how could this possibly influence your irrigation water demand estimates?

*AR: Sure, AET is also related to differences in T and radiation. T is relatively consistent between the forcings (Fig. 2). Radiation is for sure one large driver for AET. As global Q is well constrained by observations in calibration regions, and the water balance is closed, differences in meteorological*

*forcings need to be balanced by the other water balance components. As P is obviously the most important variable of the water balance, differences in P are translated to AET differences.*

**MM: This sentence (and the following) would read now as: "As Q within the calibration region is forced to be nearly equal for all climate data sets, different values of P (as well as T and radiation) lead to large differences in aggregated AET (with higher absolute differences than the P differences, or 12.2%). In contrast, AET differs by only 8.8% (and lower absolute differences than the P differences)."**

RC#2: 20. Page 18, line: 8: "leads" should be "lead"

*AR: Thanks.*

**MM: would be fixed**

RC#2: 21. Page 18, line: 24: "More likely" is not equal to "more impotant", please apply terminology consistently

*AR: Thanks.*

**MM: we would modify the "either more likely due to" to "caused mainly by the" at P18, l23, l24; in addition we would modify it at P11, l 20, l 21**

RC#2: 22. Page 19, line: 10-11: "This is.. 20th century": At first glance, this statement seems contradictory to the previous sentence. Please add some text about the changes towards T3 (1971-2000) to clarify this.

*AR: Thanks for pointing this out. We have written differences at P19 l16-21 but we would shortly introduce it here.*

**MM: We would modify this sentence to: "Anthropogenic impact increases in the time period 1941-1970 and 1971-2000, which is consistent with the …"**

RC#2: 23. Page 19, line: 14-15: "The fraction… later period". Had to read this sentence a few times before I understood what was meant. Please clarify what is meant with the later period and specify that with 1911-1940/1941-1970 is mean between/from period T1 to period T2.

*AR: Thanks, you interpreted it correctly, we need to be more specific.*

**MM: We would modify this sentence to: "Human water use and dam construction is the dominant driver for changes in long-term Q averages at 10-13% of land area for the time period 1911-1940 and 1941-1970, and increases to 13-20% of land area for the time period 1941-1970 and 1971-2000."**

RC#2: 24. Page 19, line: 16-21: I thought that the areas "which cannot be calculated"(l:19) and "where both, P and human water use is not the dominant driver"(l:20) are the same areas. But here they have different fractions associated.

*AR: The fraction "which cannot be calculated" refers the area were the denominator of indicators An and/or Bn are zero. In contrast, "where both, P and human water use is not the dominant driver" relates to the grid cells, where An and Bn are both negative.*

**MM: The whole paragraph would be restructured as follows: "Human water use and dam construction is the dominant driver for changes in long-term Q averages at 10-13% of land area for the time period 1911-1940 and 1941-1970, and increases to 13-20% of land area for the time period 1941-1970 and 1971-2000. The fraction with P domination increases, too, from 53–54% to 58–65%. At the same time, the area for which the indicators A and B cannot be calculated (due to zero in the denominators of Eqs. 3 and 4) decreases from 30 to 20% mainly because the human impact on river discharge becomes more prevalent over time (Figs. B1 and B2 in Appendix B regarding A and Figs. B3 and B4 regarding B). The land fractions where neither driver is dominant remain around 4% in both periods."**

RC#2: 25. Page 20, line: 2: "STANDARD": I haven't seen this model reference before whilst you refer to this specific version of model in earlier paragraphs. Please use consistent namings.

*AR: In earlier paragraphs we are referring to the whole set of model variants of MS14, and in this specific paragraph we refer to this specific model variant.*

**MM: none**

RC#2: 26. Page 20, line: 13-16: "In addition…in STANDARD". Incomplete sentence, please reformulate

*AR: Thanks.*

**MM: We would split up this sentence and modify it to: "In the applied WaterGAP 2.2 (ISIMIP 2.1) version, reservoirs are filled up with water in their construction year. This leads to a net increase of reservoir storage (53 km³ yr$^{-1}$) compared to a decrease of 43 km³ yr$^{-1}$ in STANDARD, where reservoirs are assumed to have been in operation over the entire simulation period."**

RC#2: 27. All of section 4.2 and many parts of 4.3/4.4 are results, no discussion. Please reshuffle.

*AR: Thanks.*

**MM: In the revised manuscript we would join results and discussion.**

RC#2: 28. Page 21, line: 21-22: "the effect .. time aggregation". Was this specifically evaluated? If yes, where (I cannot find the associated results section)

*AR: Thanks for pointing this out. We based this evaluation on Fig. 2.*

**MM: We would revise this part.**

RC#2: 29. Page 23, line: 26: Would dam construction indeed lead to significant decreases in long-term Q? I would think that dam construction would mainly influence the timing of runoff peaks/lows. Could you give a reference for this statement?

*AR: Due to increased evaporation from the reservoir surface (which was not there before dam construction), a decrease in mean Q is expected. Döll et al. (2009) calculated the decrease of mean Q solely due to dam construction to be 0.8% globally. In conjunction with dam construction, water is used often e.g. for irrigation, which also reduces mean Q. Anyhow, you are right, effect of timing is more substantial.*

**MM: We would add the reference Döll et al. (2009).**

RC#2: 30. Page 24, line: 2:8: "For example…"anthropogenic effects": Do you expect that under the different climate forcings the absolute value of the anthropogenic impacts (mainly irrigation I would say) also changes or is this difference in outcome (relative contribution) only determined by changes in P)

*AR: Yes, irrigation water use depends on the climate forcings (all variables).*

**MM: We would add a sentence to P24, l2: "They lead to different changes of P and different changes of human water use as the globally dominant irrigation water use is computed as a function of climate."**

RC#2: 31. Page 25, line: 20-21: "e.g. ISI-MIP2.1". Could you mention more model intercomparison projects? Think of Agmip, Earth2Observe.

*AR: Thanks for this suggestion.*

**MM: we would include AgMIP, Earth2Observe and LS3MIP to the list of current model intercomparison projects.**

RC#3: This manuscript presents a quantitatively useful update on our ability to marry terrestrial models of water and energy balance & their uncertain physical processes and tuning parameters, with records of hydrometeorological forcing that are fragmentary and have significant time-dependent biases. These forcing biases result from an amalgam of diverse satellite and in situ data as well as inputs from atmospheric reanalyses.

*AR: Thank you for this very positive general statement.*

RC#3: The analysis presented here covers time scales from annual to century and spatial scales from 0.50 degree lat/lon grids to global land means. While the formalism here is a largely a standard approach there are two specific aspects I feel are noteworthy:

1) The first is the identification of how specific differences in forcing data propagate through the water balance and, by virtue of tuning / calibration on observed discharge affect gauged and ungauged contributions differently. Both the low precipitation in the PGFv2 data (as a result primarily of no

snowfall undercatch correction) and the low values of downward SW radiation in the WATCH forcing (resulting from the older NCEP reanalysis values) are cases in point. In gauged regions, discharge (Q) is controlled by calibration and so AET responds to the differences in precipitation and radiative forcing. (Note the large spread in global AET in Table 3, focused in Europe and N. America where snowfall is significant.) Conversely, in ungauged regions the resulting uncertainty from forcing data sets in discharge is over 18% (Table 2). These results quantify the effects of forcing data quality and (un)availability on regional and global results.

2) The other noteworthy aspect of this paper is the attempt to identify the relative roles of climate forcing versus anthropogenic effects on changing water balance. Diagnostic indicators involving the relative changes in P versus Q and changes in the actual versus "naturalized" P and Q changes are considered. The resulting maps (Figs 3,4) are valuable, I think, not only for their consistency but also their differences between forcing data sets. One is not surprised to see the western US and Europe exhibit significant increases in diversions /extractions through the middle of the 20th Century. The growth of these impacts throughout southern Asia in the second half of the 20th Century is reasonably consistent across the four homogeneous data sets. Nevertheless there are significant regional differences in the estimated growth of anthropogenic effects in Australia, China and Mexico are discussed as examples where the forcing data set differences have significant interpretive consequences.

So, basically, my sense is that this paper is a valuable assessment of where our diagnostic modeling capabilities for water balance stand.

*AR: We are grateful for the assessment of this paper, thanks for this comprehensive summary and your thoughts about it.*

RC#3: That said, there are some aspects of the presentation that need improvement:

i. The discussion in section 2.4.2 on the construction of the indicators for anthropogenic effects was difficult to follow. The reasoning behind An and Bn seems clear enough, but I had a difficult time trying to understand how An and Bn were incorporated in Figs 3 and 4. What combinations of An >0, Bn<0 and Ivarpredoc,n >0 make up red or blue areas in those Figs? Presumably An and Bn are of opposite sign in both blue and read areas? Perhaps some schematic picture would be useful.

*AR: Thank you for pointing this out.*

**MM: We would restructure the description of the indicators for a better readability and integrate the indicator criteria to the legend of Figs. 3 and 4. In addition, we would include the An and Bn figure in Appendix B as well as add a new Table 1with some calculation examples.**

RC#3: ii. There were numerous places in the manuscript where I was unsure as to what Figures or Tables the discussion related to. Does section 4.2 refer to the information in Fig 2? Does the discussion in section 4.3 relate to Table 2 and 3? Does section 4.4 refer to Figures 3 and 4? Alluding to the appropriate graphic needs to be added.

*AR: Thank you.*

**MM: Based on the comments of referee #2 we would join results and discussion and emphasize for integrating related Tables or Figures where required. Even if some of the tables are referenced in different sections, we hope it is then more structured.**

RC#3: iii. I don't really see a lot of value of section 4.1. It largely discusses differences with the standard version WaterGAP 2.2. Perhaps mentioning earlier estimates from papers such as Oki and Kanae (2006), Haddeland et al, (2011) and Rodell et al, (2015) would give some context outside the WaterGAP model.

*AR: Thanks. In the paper Müller Schmied et al. (2014), Table 5, we compared WaterGAP output to many other global estimates and referred to it here because we do not want to simply reproduce this table. Anyhow, as we missed the global assessment of Wisser et al. (2010), Hanasaki et al. (2010) and there are some new numbers available (Rodell et al., 2015 as you mentioned).*

**MM: We would describe more clearer in Sect. 3.1, when we are comparing to earlier WaterGAP results or other studies. We would completely revise the previous section 4.1 and add the numbers of Wisser et al. (2010), Hanasaki et al. (2010) and Rodell et al. (2015) in a new table 6. For this comparison, we would analyze WaterGAP outputs for same time span and same spatial extent as it is described in the references. We feel that this part of Sect. 3.1 would now be more of value.**

RC#3:I recommend accpetance after attention to these these three areas of concern

*AR: Thank you again for your constructive assessment.*

References

Beck, C., Grieser, J. and Rudolf, B.: A new monthly precipitation climatology for the global land areas for the period 1951 to 2000, German Weather Service, Climate Status Report KSB 2004, Offenbach, Germany., pp. 181–190, 2005.

Biemans, H., Hutjes, R. W. A., Kabat, P., Strengers, B. J., Gerten, D. and Rost, S.: Effects of precipitation uncertainty on discharge calculations for main river basins, J. Hydrometeorol., 10(4), 1011–1025, doi:10.1175/2008JHM1067.1, 2009.

Döll, P., Fiedler, K. and Zhang, J.: Global-scale analysis of river flow alterations due to water withdrawals and reservoirs, Hydrol. Earth Syst. Sci., 13(12), 2413–2432, doi:10.5194/hess-13-2413-2009, 2009.

Fuchs, T., Rapp, J., Rubel, F. and Rudolf, B.: Correction of synoptic precipitation observations due to systematic measuring errors with special regard to precipitation phases, Phys. Chem. Earth, Part B Hydrol. Ocean. Atmos., 26(9), 689–693, doi:10.1016/S1464-1909(01)00070-3, 2001.

Hanasaki, N., Inuzuka, T., Kanae, S. and Oki, T.: An estimation of global virtual water flow and sources of water withdrawal for major crops and livestock products using a global hydrological model, J. Hydrol., 384(3-4), 232–244, doi:10.1016/j.jhydrol.2009.09.028, 2010.

Klein Goldewijk, K., Beusen, A., Van Drecht, G. and De Vos, M.: The HYDE 3.1 spatially explicit database of human-induced global land-use change over the past 12,000 years, Glob. Ecol. Biogeogr., 20(1), 73–86, doi:10.1111/j.1466-8238.2010.00587.x, 2011.

Müller Schmied, H., Eisner, S., Franz, D., Wattenbach, M., Portmann, F. T., Flörke, M. and Döll, P.: Sensitivity of simulated global-scale freshwater fluxes and storages to input data, hydrological model structure, human water use and calibration, Hydrol. Earth Syst. Sci., 18(9), 3511–3538, doi:10.5194/hess-18-3511-2014, 2014.

Rodell, M., Beaudoing, H. K., L'Ecuyer, T. S., Olson, W. S., Famiglietti, J. S., Houser, P. R., Adler, R., Bosilovich, M. G., Clayson, C. A., Chambers, D., Clark, E., Fetzer, E. J., Gao, X., Gu, G., Hilburn, K., Huffman, G. J., Lettenmaier, D. P., Liu, W. T., Robertson, F. R., Schlosser, C. A., Sheffield, J. and Wood, E. F.: The Observed State of the Water Cycle in the Early 21st Century, J. Clim., 150904104833007, doi:10.1175/JCLI-D-14-00555.1, 2015.

Schneider, U., Becker, A., Finger, P., Meyer-Christoffer, A., Rudolf, B. and Ziese, M.: GPCC Full Data Reanalysis Version 7.0 at 0.5°: Monthly land-surface precipitation from rain-gauges built on GTS-based and historic data, doi:10.5676/DWD_GPCC/FC_M_V7_050, 2015.

Stocker, T. F., Qin, D., Plattner, G.-K., Alexander, L. V., Allen, S. K., Bindoff, N. L., Bréon, F.-M., Church, J. A., Cubasch, U., Emori, S., Forster, P., Friedlingstein, P., Gillett, N., Gregory, J. M., Hartmann, D. L., Jansen, E., Kirtman, B., Knutti, R., Krishna Kumar, K., Lemke, Pl, Marotzke, J., Masson-Delmotte, V., Meehl, G. A., Mokhov, I. I., Piao, S., Ramaswamy, V., Randall, D., Rhein, M., Rojas, M., Sabine, C., Shindell, D., Talley, L. D., Vaughan, D. G. and Xie, S.-P.: Technical Summary, in: Climate Change 2013: The Physical Science Basis. Contribution of Working Group I to the Fifth Assessment Report of the Intergovernmental Panel on Climate Change, Stocker, T.F., Qin, D., Plattner, G.-K., Tignor, M., Allen, S. K., Boschung, J., Nauels, A., Xia, Y., Bex, V. and Midgley P.M. (eds.). Cambridge University Press, Cambridge, United Kingdom and New York, NY, USA., 84 pp., 2013.

Voisin, N., Wood, A. W. and Lettenmaier, D. P.: Evaluation of Precipitation Products for Global Hydrological Prediction, J. Hydrometeorol., 9(3), 388–407, doi:10.1175/2007JHM938.1, 2008.

Wisser, D., Fekete, B. M., Vörösmarty, C. J. and Schumann, A. H.: Reconstructing 20th century global hydrography: a contribution to the Global Terrestrial Network- Hydrology (GTN-H), Hydrol. Earth Syst. Sci., 14(1), 1–24, doi:10.5194/hess-14-1-2010, 2010.

Table 1: Examples of indicator calculation (Sect. 2.4.2) for four large river basins at grid cells located near the outflow to the ocean for the forcing GSWP3 as well as time steps 1941-1970 (t1) and 1971-2000 (t2). Values for latitude and longitude in degrees, other numbers in km$^3$ yr$^{-1}$; n.c.: not computed.

|  | Rhine River | Kongo River | Colorado River | Yellow River |
|---|---|---|---|---|
| Lat | 4.25 | 12.25 | -114.75 | 133.25 |
| lon | 52.25 | -6.25 | 31.75 | 48.25 |
| $P_{bas(n),t1}$ | 169.36 | 5735.52 | 191.24 | 771.92 |
| $P_{bas(n),t2}$ | 176.43 | 5469.11 | 206.56 | 771.91 |
| $Q_{natn,t1}$ | 69.27 | 1370.46 | 1.53 | 215.28 |
| $Q_{natn,t2}$ | 75.19 | 1251.09 | 1.92 | 209.94 |
| $Q_{n,t1}$ | 67.83 | 1370.46 | 0.62 | 213.41 |
| $Q_{n,t2}$ | 72.63 | 1250.67 | 0.10 | 203.68 |
| $A_n$ | 0.68 | 0.45 | -0.03 | 1163.26 |

| | | | | |
|---|---|---|---|---|
| $B_n$ | -4.29 | 286.86 | 0.57 | 2.21 |
| $I_{varprec,n}$ | n.c. | 57.49 | n.c. | 9.73 |
| $I_{varant,n}$ | n.c. | 118.96 | n.c. | 0.93 |
| $I_{varprecdom,n}$ | n.c. | 61.46 | n.c. | -8.81 |
| Dominant driver | $A_n > 0$ & $B_n < 0$: Precipitation | $I_{varprecdom,n} > 0$: Precipitation | $A_n < 0$ & $B_n > 0$: Human impact | $I_{varprecdom,n} < 0$: Human impact |

Table 6. Global and continental estimates of WaterGAP compared to literature values [km$^3$ yr$^{-1}$]. WaterGAP results are analysed for the same time span and spatial coverage as the reference and is comparable in terms of precipitation undercatch.

| Source | Coverage | time span | P | | AET | | Q | |
|---|---|---|---|---|---|---|---|---|
| | | | WaterGAP | reference | WaterGAP | reference | WaterGAP | reference |
| Wisser et al. (2010) | global, w G | 1901-1925 | 102 110[a] | 105 298 | 63 319[a,b] | 68 274 | 37 974[a] | 36 888 |
| | | 1926-1950 | 102 653[a] | 105 675 | 63 081[a,b] | 67 826 | 38 837[a] | 37 092 |
| | | 1951-1975 | 105 444[a] | 108 081 | 64 693[a,b] | 68 550 | 39 914[a] | 38 864 |
| | | 1976-2002 | 104 436[a] | 106 764 | 64 337[a,b] | 69 917 | 39 421[a] | 36 813 |
| | | 1901-2002 | 103 676[a] | 106 461 | 63 867[a,b] | 68 480 | 39 044[a] | 37 401 |
| Hanasaki et al. (2010) | global, w G, w A | 1984-1999 | 106 012[a,c] | 113 900 | 64 281[a,b,d] | 72 080 | 40 876[a,e] | 41 820 |
| Rodell et al. (2015) | global, w/o A | 2000-2010 | 113 341[f] | 114 300 | 71 554[b,f] | 70 500 | 41 309[f] | 43 800 |
| | NAmerica, w G | | 17 983[f] | 17 717 | 10 339[b,f] | 9911 | 6604[f] | 7894 |
| | SAmerica | | 29 153[f] | 29 587 | 17 573[b,f] | 17 286 | 11 579[f] | 12 301 |
| | Africa | | 21 323[f] | 20 629 | 17 307[b,f] | 16 809 | 4029[f] | 3820 |
| Müller Schmied et al. (2014) | global, w/o G, w/o A | 1971-2000 | 111 050[g] | 111 070[h] | 69 819[b,g] | 70 576[b,h] | 41 298[g] | 40 458[h] |

[a]PGFv2.1, [b]including WCa, [c]including Antarctica (as 2.1% of global value), [d]including Antarctica (as 4.9% of global value), [e]including Antarctica (as 0.2% of global value, all percentages based on Rodell et al., 2015), [f]WFDEI_hom, [g]WFD_WFDEI, [h]STANDARD model variant; G in column coverage: Greenland, A in column coverage: Antarctica.

**Appendix B: Consistency indicators A and B**

[Figure]

Figure B1: Indicator of consistency A between the change in P and the change in Q (Eq. 3) for the time periods 1911-1940 and 1941-1970. Red colours that the change in Q is zero such that A cannot be computed. In the yellow areas A is less than 0, i.e. change in P had the opposite sign of the change in Q; therefore, P was not the dominant driver for change in Q. Results are shown for WaterGAP as driven by the climate forcings GSWP3 (a), PGFv2.1 (b), WFDEI_hom (c) and WFD (d).

[Figure]

Figure B2: Indicator of consistency A between the change in P and the change in Q (Eq. 3) for the time periods 1941-1970 and 1971-2000. Red colours that the change in Q is zero such that A cannot be computed. In the yellow areas A is less than 0, i.e. change in P had the opposite sign of the change in Q; therefore, P was not the dominant driver for change in Q. Results are shown for WaterGAP as driven by the climate forcings GSWP3 (a), PGFv2.1 (b), WFDEI_hom (c) and WFD (d).

[Figure]

Figure B3: Indicator of consistency B between change in the anthropogenic impact on Q (i.e. Qnat-Q) and the change in Q (Eq. 4) for the time periods 1911-1940 and 1941-1970. Red colours indicate where the change in anthropocentric impact was zero such that B cannot be computed. In the yellow areas B is less than 0, and the change in anthropogenic impact on Q is not consistent with the change in Q. Therefore, the anthropogenic impact is not the dominant driver for change in Q. Results are shown for WaterGAP as driven by the meteorological forcings GSWP3 (a), PGFv2.1 (b), WFDEI_hom. (c) and WFD (d).

[Figure]

Figure B4: Figure B3: Indicator of consistency B between change in the anthropogenic impact on Q (i.e. Qnat-Q) and the change in Q (Eq. 4) for the time periods 1941-1970 and 1971-2000. Red colours indicate where the change in anthropocentric impact was zero such that B cannot be computed. In the yellow areas B is less than 0, and the change in anthropogenic impact on Q is not consistent with the change in Q. Therefore, the anthropogenic impact is not the dominant driver for change in Q. Results are shown for WaterGAP as driven by the meteorological forcings GSWP3 (a), PGFv2.1 (b), WFDEI_hom. (c) and WFD (d).

---

## Author Comment (AC2) · 17 Mar 2016

**Modification of the manuscript due to the updated climate forcing PGFv2.1**

During review time of the submitted manuscript, we were informed by the project coordinators of the Inter-Sectoral Impact Model Intercomparison Project (ISI-MIP), that there is a problem in the PGFv2 data for temperature during the period 1901-1947. "The problem was a mistake in the data provider's processing scripts for temperature and humidity" (https://www.pik-potsdam.de/research/climate-impacts-and-vulnerabilities/research/rd2-cross-cutting-activities/isi-mip/for-modellers/isi-mip-phase-2/input-data/input-data-issues, last access at 2016-03-10)). We have therefore assessed the difference between PGFv2 (which was used in the original submission) and the update PGFv2.1 and decided to use the updated variant in the revision of the manuscript. Here, we describe briefly the effect of the differences of the updated T (and its effect to water balance components) and the modifications that would result in the revised manuscript.

**Global means of climate forcing variables and model results**

As seen in Fig. 1, the correction of temperature T affects yearly or decadal aggregations of T by about 0.1 to 0.5 °C which is very large for global averages. The new T pattern fits well to those of the other forcings (comp. to Fig. 2 in the HESSD manuscript). SWD, LWD and P are not affected by the error. As for calculation of longwave outgoing radiation temperature is considered, NR differs by around 1 W m$^{-2}$. AET and Q are differs up to 500 km$^3$ yr$^{-1}$, whereas the influence of modified T on WCa is only marginal.

**Spatial differences and effect for calibration**

WaterGAP is calibrated to observed long-term average river discharge, and for PGFv2 preferably for the years 1971-2000 but years of calibration differs in basins with limited data availability. Within 21 river basins (shown in Fig. 2), (some) of the years which are affected by the T error are between 1901 and 1947. Thus, the error could have influenced the calibration approach. However, as shown in Fig. 2, these basins are located mainly outside of "hot spots" where discharge and other variables (except for PET, but these regions are mostly in water-limited areas) are strongly affected by the differences between PGFv2 (erroneous) and PGFv2.1. We therefore would not re-calibrated WaterGAP with PGFv2.1 climate input but use the calibration parameters of the calibration with PGFv2 for the new model runs with PGFv2.1.

**Consequences for the revised manuscript**

Fig. 2 will be updated showing model results computed with PGFv2.1 as climate input instead of PGFv2. At (nearly) all occurrences in the text, we changed PGFv2 into PGFv2.1. We would add a sentence at the end of the description of the forcing: "During review process of this manuscript, we were informed about an error in T data for the period 1901-1947. We therefore used the updated forcing PGFv2.1." Numbers will be updated where necessary.

[Figure]

Figure 1: Global means of climate forcing variables and model outputs (T: temperature, SWD: shortwave downward radiation, LWD: longwave downward radiation, NR: net radiation, P: precipitation, Q: river discharge, PET: potential evapotranspiration, AET: actual evapotranspiration, WCa: consumptive water use) for the climate forcings PGFv2 (light colors) and PGFv2.1 (with corrected error for T during 1901-1947, dark colours).

[Figure]

Figure 2: Spatial differences of simulated model outputs of PGFv2 compared to PGFv2.1 for the time span 1901-1947. PET: potential evapotranspiration, AET: actual evapotranspiration. Purple basin outlines included in right column show all the river basins where (some of) the calibration years are within the period 1901-1947 and therefore could be affected by the erroneous PGFv2 climate data.

---

## Author Response (AR1)

**Point by point response to the referees and modification of the manuscript**

First of all, we are grateful to the referees for their suggestions. We would like to answer each comment by providing first the referee's comment (normal font, introduced by RC#x), the author's response (italic font, introduced by AR) and the modification of the manuscript (bold font, introduced by MM).

RC#1 This m/s used alternative climate forcings in a global hydrological model to determine the resulting uncertainty in water resource estimates, and compares that uncertainty with the estimated impact of human modifications on water resources.

*AR: Thank you for the nice summary the manuscript.*

**MM: none**

RC#1: In summary, I am not convinced of the original contribution of this m/s.

The overall methodology seems sound enough and the first 2 questions asked (page 4) appear answered by the analysis. The 3rd question seems ill-formulated within the context of this m/s, as it does not clearly distinguish between real changes in precipitation and apparent changes that are artefacts in the data.

*AR:. We agree with the reviewer's comment that we do not distinguish between real temporal changes and apparent changes that are artefacts of the data. We actually are not in the position to do so as users of forcing data that have been produced by various data producers. This problem also concerns research questions 1 and 2, and is, in our opinion outside the scope of the manuscript. With the analysis of the (dominant) impact of precipitation changes or human impact on the temporal development of river discharge research question 3)  we provide a new combined analysis for distinguishing precipitation changes and human impact taking into account the uncertainty of precipitation data; this in our opinion fits well into the context of the manuscript (compare the title of the manuscript). The reviewer's comment, however, reminded us that we should mention in the manuscript that monthly precipitation observations used in producing the applied precipitation data sets are heterogeneous in time, i.e. that the number of stations is not constant over time, which makes them less suitable for the analysis of temporal trends. The applied GPCC and CRU monthly precipitation data sets are optimized for best spatial coverage.*

**MM: We have added the following sentence to section 2.3 (as last line of introductory paragraph):**

**"In all data sets, daily precipitation estimates were obtained by bias-correcting output of weather models by monthly precipitation data sets that had been derived from monthly precipitation observed at raingages. These monthly data sets were optimized for spatial coverage, i.e. using, for each month, the available number of gauging stations. The temporally variable number of considered precipitation observations makes the applied precipitation data sets less suitable for the analysis of temporal variations. While a temporally homogeneous data set of observation-based monthly precipitation exists at least for the time period 1950-2000, it is based on less than 10 000 gauging stations and therefore provides a spatially less accurate representation of global-scale precipitation (Beck et al. , 2005) than the data sets used in this study, which include up to 50 000 gauging stations (Schneider et al., 2015)."**

RC#1: However the concern I have with this m/s is the answers to those first 2 questions are already known and the results unsurprising. Regards (1): from numerous previous studies (including those cited but many more – including high level reports such as from IPCC etc) we already have a fair idea of temporal variations in global water cycle components.

*AR: Indeed, previous work was done to assess temporal variations in global water components, and we certainly have missed a study when preparing the first submission. Wisser et al. (2010) evaluated temporal variation in 25 year steps of modeled global water balance components (their Table 2). They used a model approach, CRU TS 2.1 as basis data and some precipitation products for the uncertainty analysis. In the submitted manuscript, we present a consistent assessment of more than a century with different temporal aggregation steps for 4 + 1 state-of-the-art climate forcings (with more than varying precipitation variable) that are frequently used in global hydrological modeling and this has, to our knowledge, not yet been done in this comprehensive manner. Most other previous work considered specific time steps (e.g. 1961-1990). Regarding knowledge assessed in the IPCC reports (mainly Chapter 2 in the WG I report 2013), the focus is on long-term trends, while we present different temporal aggregations. And while IPCC considers knowledge on each water balance component separately, we aim to provide a comprehensive and consistent picture of the temporal development of all components.*

**MM: We refer to the study of Wisser et al. (2010) in the introduction.**

**P4, line 17 reads now as: In addition to these uncertainties, water resources estimates differ due to different reference periods (Wisser et al., 2010).**

**And we relate our results regarding the long-term trend of Q to IPCC results.**

**P15, line 10 reads now as: The results of this study confirm the finding of the IPCC Fifth Assessment Report that "the most recent and most comprehensive analyses of river runoff do not support the IPCC Fourth Assessment Report (AR4) conclusion that global runoff has increased during the 20th century." (Stocker et al., 2013, p. 44)**

RC#1: Regards (2): we already know that global precipitation analysis in particular diverge considerably and of course that will propagate through a hydrological model.

*AR: There are studies available showing differences e.g. in precipitation products (e.g. Biemans et al., 2009; Voisin et al., 2008; Wisser et al., 2010) and also the correspondent changes in river discharge, but they mainly stay at the level of different precipitation data. To our knowledge, the information we present in Table 4 is novel as it compiles, for each precipitation product, the station density as well as the effect of undercatch correction on continental and global level. In addition, we did not only consider precipitation, but also other climate variables, which can have a huge impact on model results (e.g. short-wave radiation, see the combination of WFD and WFDEI) and may overwhelm the effect of different precipitation products.*

**MM: We have added the reference to Voisin et al., 2008 to the manuscript and modify the sentence slightly. P4, l11 reads now as: Simulated water balance components vary considerably due to various uncertainties of GHMs (Haddeland et al., 2011; Schewe et al., 2014) including GHM ability to include human water use, model improvements over time (e.g. see the different results of the Water Global Assessment and Prognosis (WaterGAP) model in Müller Schmied et al. (2014), their Table 5), climate forcing (Biemans et al., 2009, Voisin et al., 2008) as well as uncertainties in discharge observations (Coxon et al., 2015; McMillan et al., 2012).**

RC#1: Regards (3): This m/s shows that depending on the precipitation data set chosen we see the human impact on the water cycle less or more clearly, but always in places where we know it exists.

*AR: We agree in the sense that our analysis is based on data on irrigated areas and dams (where we know they exist). The dominance patterns in Figure 4 obviously reflect these data, but they are more complex than the underlying data (and partially surprising). It is true that precipitation is one driver for this method and we used precipitation as it is has probably the key role, but for the human impact indicator (now solely defined as $B_n$ ), we are using river discharge, where precipitation is not the only driver (but also radiation and temperature). The attempt was not to present new regions of human impact on water cycle, but to distinguish between climatic (data) changes and (simulated) human impact. We show this assessment for two time spans and four forcings, which gives an idea of the impact of forcing (which can be huge) as well as the considered time span(s). We believe that this assessment can be of value for future indicators / assessments.*

*We believe that the referee's comment "seems ill-formulated" regarding the 3rd question refers to the detailed research question. As our modeling approach is not able to e.g. reflect human changes in land use, we cannot simplify the question to " – either change of precipitation or change of human impact".*

**MM: none**

RC#1: Furthermore, on page 19/20 the authors refer to an earlier study in 2014 that sounds like it had very similar objectives to the present m/s. It is not clear to me what new insights this m/s adds to that previous study, given its title makes explicit reference to the sensitivity of the model to input data.

*AR: This section is also criticized by the other Referees, and was revised. Here we refer to the paper Müller Schmied et al. (2014) (referred from now on as MS14). MS14 evaluated the sensitivity of model outputs to climate forcing (comparing two forcings, of which one (WFD_WFDEI) is used here also), land cover (two land cover inputs), human water use (included, not included), model structure (two model versions) and calibration (or no calibration). MS14 is more general and showed that calibration has the largest effect, followed by climate forcing. In this manuscript, we are focusing on the effect of climate forcing (4 state-of-the-art forcings), applied a methodology for improving the combination of two timely consecutive forcings (WFD, WFDEI) plus applied the forcings to quantify the dominance indicator. Furthermore, the assessment of the used climate forcings is of interest for the many modeling groups involved in the Inter-Sectoral Impact Model Intercomparison (ISI-MIP) activities.*

**MM: We have joined the results and discussion section and have rewritten most of the content in the new section 3.1. which is called now "Water balance components as impacted by climate forcing uncertainty", also including now the new Table 6 with comparisons to Wisser et al. (2010), Hanasaki et al. (2010) and Rodell et al. (2015).**

RC#1: The above does not mean that the analysis presented cannot be used to draw some interesting new conclusions. In particular, the authors draw attention to the unexpectedly large uncertainty in North America and Europe, which they attribute to undercatch corrections. I thought that was a very interesting finding which could probably be the topic of a m/s in its own right.

*AR: Thank you for this statement and suggestion. We agree that addressing undercatch correction (or not) in a thorough way should be done in a separate study. In our manuscript we highlight the impact of precipitation undercatch (e.g. P23, l1; P23, l6), and in the revised manuscript we added a sentence in the conclusions that can be understood as a call in this direction.*

**MM: In the revised manuscript, we would add a sentence on P25, l24 of the initial manuscript:**

**Development of an improved method for correcting the global state-of-the-art precipitation products, by building on the work of Fuchs et al. (2001), would enable a better quantification of global precipitation.**

RC#1: page 6: It sounds like you essentially treat land use as unchanged during the model period. That has a precedent of course but is still a limitation, pls discuss.

*AR: We agree with the referee's comment. Land use / land cover has changed within the model time period (20th century and early 21st century). Within the WaterGAP model we do not consider dynamic land use as cropland or pasture (which could be covered by using e.g. the HYDE database (Klein Goldewijk et al. 2011)). Since MODIS satellites are in orbit (since the beginning of the 21st century), remote sensing based land cover classification can be done, and is included in the model as a snapshot. Currently, we do not see an advantage to change year-by-year the land cover e.g. based on MODIS, as this reflects only the last ~15 years, and consistent land cover classifications in for the 20th century (which fits to the current model structure) is not available to our knowledge. In addition, future research has to be done to include e.g. yearly changed land cover classification in a consistent manner. To change, for example, land cover at each 1st of January can be problematic e.g. for modeled leaf area index and rooting depths, which could then result in jumps of water storages (e.g. in canopy or soil).*

**MM: none, as we do not feel that a discussion of this model assumption would not fit into the scope of the manuscript.**

RC#1: page 12: This section requires proper statistical treatment, which some type of significance testing, not the type of 'binary' heuristics you use here. For starters, you clearly show that the forcing data are uncertain (not to mention the model) so that error needs to be considered in testing.

*AR: We think that given the large but unknown uncertainties (e.g. regarding the existence of irrigated areas, or net radiation), it is not possible to quantify errors in a meaningful way and to do significance testing.*

**MM: none**

RC#2: In this manuscript, the authors assessed the variations of global and continental water balance components as impacted by climate forcing uncertainty and human water use. In addition to that, the authors evaluated the impact of climate forcing uncertainty and the effect of temporal aggregations on the different water components at local, regional, and global scales.

The paper addresses relevant scientific questions within the scope of HESS. Although it is not the first paper that evaluates the impacts of human interventions versus hydrometeorological changes on hydrological parameters, the research angle chosen as well as the approach could be of added value to the research domain of water resources research.

Overall, the paper reaches some interesting conclusions- especially regarding the uncertainty of meteorological forcing data-sets and with respect to the dominant factors of change - backed-up with figures and numbers (results) and substantiated with some existing research (discussion). I would

therefore support the manuscript for publication but with substantial revisions taking into account the following general and technical comments/suggestions.

*AR: Thank you for the nice summary and the general evaluation.*

RC#2: 1. Locally redundant text, especially when it comes to the introductory parts and descriptions of data in different sections:

*AR: Thank you for pointing out redundancy so clearly.*

RC#2: a. The introduction of section 2 (Data and Methods) shows much overlap with the last paragraphs of section 1 (Introduction).

*AR: We have modified the last paragraph of the introduction.*

**MM: P5, first sentence reads now as: To answer these questions, we conducted a modeling experiment.**

RC#2: b. Page 6, line:2-26: Move the list of modifications to the appendix.

*AR: Thanks, done as suggested.*

**MM: We moved the bullet points at P6 and P7 to the appendix, and the sentence at P6,l2 reads now as: "The version used for this study is named WaterGAP 2.2 (ISI-MIP 2.1), and differences to WaterGAP 2.2 mainly consider requirements of the ISI-MIP project phase 2.1 as described in Appendix A." In addition, we added a bullet point that describes differences in the calibration setup for WFDEI_hom forcing: "For WaterGAP calibration, we used observed streamflow data from 30 years. For GSWP3, PGFv2 and WFD, we used data from 1971-2000 if available for the time period. Due to the offset in radiation of WFD_WFDEI forcing (and consequences for model results, see Müller Schmied et al., 2014), we calibrated WFDEI_hom using using preferably the period 1980-2009 and used these calibration parameters for the WFD_WFDEI simulation." We also have reformulated the bullet point regarding the reduction factor of lake evaporation to: "For lakes, reduction of evaporation due to decreasing lake area is calculated according to Eq. 1 in Hunger and Döll (2008), resulting in a lower but more realistic lake area and thus evaporation reduction with decreasing lake storage."**

RC#2:c. Page 16, line:7-9: Here the authors refer to Müller Schmied et al (2015) for a comparison of AET, Q, WCa, and total water storage to the five climate forcings. Since the different climate forcings are already described in this paper, the authors can shorten all text with reference to these climate forcings (section 2.3).

*AR: Thanks for this suggestion. In the submitted conference proceeding, we are referring to this HESSD-manuscript and providing only one table (Table 1) with the main characteristics of the climate forcings.*

**MM: As we now feel that this sentence does not fit to the content of this paragraph, we removed it.**

RC#2:d. The results and discussion section show a lot of overlap. In its current form, the discussion section is basically a repetition of/part of the results section but with some possible explanations for

the observations given. Please avoid any unnecessary repetition and try to substantiate/discuss the found results with previous research where possible.

*AC: When separating results and discussion, at least some overlap cannot be avoided. We decided to join the results and discussion in order to reduce the overlap and included some more references.*

**MM: The results section is now structured as follows: Sect 3.1: "Water balance components as impacted by climate forcing uncertainty" with subsections 3.1.1 "Uncertainty of global climate forcings", 3.1.2 "Uncertainty of simulated water balance components due to climate forcing uncertainty" and 3.1.3 "Comparison with other studies". This section includes initial sections 3.2, 3.3, 4.1 and 4.3 (answering research question 2, now 1). Sect. 3.2 reads now as: "Variation of estimated global water balance components across temporal aggregation and reference periods" (includes sections 3.1, 4.2 and answering research question 1 and 2). Sect. 3.3 would be now: "Dominant drivers of temporal variations of 30-year mean annual river discharge: precipitation or human water use and dam construction" (includes sections 3.4, 4.4 and answering research question 3). We additionally have re-arranged the research questions (interchanged the position of 1 and 2). Consequently, we modified the conclusion section according to the rearranged research questions. We feel that this revising improves the readability and shapes the manuscript significantly.**

RC#2: e. The authors use their research questions as titles for the discussion section, please choose shorter title names.

*AR: We intended to allow a quick search for the answers to the research questions and have chosen therefore the research questions as titles for the discussion section. Anyhow, we tried to find titles which match to the research questions in the newly joined results and discussion section.*

**MM: The new section titles are written in the previous MM.**

RC#2: 2. The results presented in the results section do not follow an intuitive order. Section 3.1 (temporal aggregations) is not a very logical section to start with. Given the order GCM(-impact)-GHM(-impact)-outputs I would start with the GCM uncertainty discussion (3.1 - make a separate section for this topic), then continue with the human impacts section (3.2), and finally end with the impact of aggregation section (3.3). Results for the global and continental scales could be discussed in sub-sections. Change the order of the research questions and the discussion accordingly.

*AR: Thank you for this suggestion. We do not have included any GCM (if Global Climate Models are meant) in this study, but we believe that the reviewer refers to forcing data uncertainty. We do not agree that the human impacts section should be placed before the global and continental scale as well as the temporal variation section as it is affected by the temporal variation. But we agree that it might be better to describe the global and continental scale first, followed by temporal aggregation. In order to separate the global and continental assessment we will stay with the subdivision in sections.*

**MM: We have modified the order of research questions (changed position of 1 and 2) and start the results and discussion section with describing the uncertainty of climate forcing to model results, followed by a section about the effect of temporal variations and finally the human impacts.**

RC#2: 3. In the discussing the impact of climate forcing uncertainty clearly distinguish between the modeling spread in forcing data (P, T, LWD) and the modelling spread in WaterGAP output (Q, AET,

WCa) and elaborate a bit more on the discussion whether the initial spread in forcing data increases/decreases when feeding it into WaterGAP.

*AR: Thank you for pointing out that we need to separate both uncertainties. Regarding the second point (if initial spread in forcings increases/decreases within WaterGAP), WaterGAP reduces initial spread of the forcings due to the calibration approach. Using observed river discharge for ~54% of the land surface, the model balances out spread in forcing data (see Table 2 (now 5), e.g. row 2 (river discharge) for calibration regions), but for non-calibration regions, spread is basically transformed to model outputs. We have described this using percentage of spread at several positions in the manuscript, e.g. at P2, l19, P25, l3, Table 2 (now 5)). As translation of climate forcing (and its spread) into model outputs is non-linear, we feel that it is not possible to further discuss that point on a solid basis.*

**MM: As written in earlier answers, we combined the results and discussion section and followed in the new Sect. 3.1 the order 1) climate forcing uncertainty, 2) impact of climate forcing uncertainty to model outputs, 3) comparison to other estimates. We are consistently going from global to continental to calibration/non calibration regions and to grid cell scale, so we hope the reader is now guided better through the results/discussion section. In addition, we have added climate forcing variables to the continental assessment in Table 3.**

RC#2: Specific/Technical comments:

4. Page 3, line: 13-16: "The international .. data-sparse regions": Would leave this sentence out. Does not add much to the introduction

*AR: Thanks for the suggestion.*

**MM: We deleted this sentence.**

RC#2: 5. Page 7, line: 5-6: "The initial .. well-being": Incomplete sentence

*AR: Thanks, we modified the sentence*

**MM: This sentence reads now as: "The purpose of WaterGAP has been to provide a best estimate of renewable water resources worldwide."**

RC#2: 6. Page 7, line: 24: Please specify (or with reference) why you applied this minimum and maximum catchment area size

*AR: Both numbers are minimum sizes. The 9000 km² are minimum upstream area which corresponds to at least 3 grid cells at the Equator as minimum. In many cases, more than one discharge observation station is located within a basin. With the 30000 km² minimum area between two calibration stations in one basin, we want facilitate an efficient calibration of WaterGAP. This was introduced by Hunger and Döll 2008, and we have increased the interstation area in MS14.*

**MM: We have modified the sentence to: "Observation stations were selected such that the upstream area had a minimum of 9000 km². To avoid including stations that are located very close to each other along a river, the minimum interstation catchment area was set to 30 000 km². Furthermore, a station was selected only if a minimum of four complete years of data was available."**

RC#2: 7. Page 12, line: 13-14: I would say that -despite A being negative- change in P might still be a significant driver to the change in Q (although not the largest/dominant driver of change). E.g. without a positive change in P, Q might have been even lower than already observed due to human activity.

*AR: Thank you for pointing this out, we agree, P can still be significant. However, it cannot be the dominant driver according to our approach.*

**MM: We have changed the word "significant" to "dominant" in this sentence.**

RC#2: 8. Page 12, line: 21-24: Isn't it technically possible that there is (+/-) change from natural to human whilst there is a change in the other direction when comparing period 1 with period 2? Again, although anthropogenic factors would in that case not be the major driving factor, the influence can still be significant.

*AR: Same answer to previous comment.*

**MM: We have changed the word "significant" to "dominant" in this sentence.**

RC#2: 9. Page 12, equations: I think it would be good to provide the reader with a map (appendix) that shows the results of the consistency indicators individually. From figures 3 & 4 I cannot deduct for the red/green areas which share of this land area is red/green simply due a negative result for the inconsistency indicator of the other parameter.

*AR: Thanks for this suggestion. While revising the manuscript, we found out that calculation of Ivarant can be reduced to the difference of the values of Qnat which cannot be an indicator of human impacts. We therefore re-thought the overall indicator approach and came up with a very simplified form.*

**MM: We re-thought the indicator approach and included the maps of An and Bn for the four forcings and the two time steps in Appendix B. We found out that we interchanged the figure caption of Fig. 3 and 4 (now 4 and 5). c) is WFDEI_hom and d) is WFD. We updated the positions in the text where it is necessary and revised Figure captions in order to improve readability. In addition (and according to Referee#3 we re-formulated and re-structured the section 2.4.2 and added a more descriptive legend into Figs 3 and 4 (now 4 and 5).**

RC#2: 10. Page 13, line: 7-8: I had to read this a few times before understanding. With the 0.5 you basically take the mean of the ratio between Q and Precip over period 1 and period 2. Would be more clear if you show something with the SUM/AVERAGE symbol in the equation. Moreover, the line afterwards "assuming that the runoff coefficient remains constant over the two time-periods" is a bit strange in this context: the runoff coefficient is namely estimated using data from the two time-periods.

*AR: Thanks for the suggestion regarding the equation. The reviewer is right, runoff coefficient is the mean between two periods, but is then applied for both periods (and thus assumed to be constant throughout the time).*

**MM: We reorganized this section and modified P13, line 6 in the initial manuscript to: "…calculated as the averaged mean runoff coefficient of the two periods under consideration.". In addition, we modified P 13 line 8 to: "Runoff coefficient is applied for the two time periods." Furthermore, we used the avg symbol in the equation to address the reviewer's concern.**

RC#2:11. Page 13, line: 7-8: Wouldn't it –w.r.t. to comment 3- be more logical to estimate the Ivarprec,n with the use of the runoff coefficients (Cqp,n) under both natural and human conditions in time period 1 and 2 only, rather than combining it again with Q and P?

*AR: We have revised the indicator approach.*

**MM: none**

RC#2: 12. Page 13, line: 7-8: I would say that it is especially this runoff coefficient that is changing due to human interventions. Wouldn't in that sense the change in runoff coefficient be responsible for the share of Q that is impacted by changes in P under a human impacts run?

*AR: The runoff coefficient can be influenced by both, climate variation or human interventions. To exclude human interventions, we are calculating the runoff coefficient by using the naturalized river discharge. Taking the change of runoff coefficient as indicator of human change is an interesting thought, but we will stay at the current form of usage.*

**MM: none**

RC#2: 13. Page 13, line: 15: Would be good to show a figurative example (with numbers) to show how/whether different P and Q scales end up to fall within the same range in Ivarant/Ivarprec.

*AR: Thank you for this suggestion.*

**MM: We have added a new Table 1 showing examples of four grid cells for the GSWP3 forcing for time steps 1941-1970 and 1971-2000. Section 2.4.2 ends now with: "To illustrate the indicator of relative dominance approach, Table 1 lists indicator values and underlying data for the example of four grid cells representing discharge of large rivers near the outlet to the ocean."**

RC#2: 14. Page 15, line: 22: "1971-2000": The values in the table refer to long-term means. Please mention in text.

*AR: Thanks for this suggestion*

**MM: After restructuring the results and discussion section, we feel it is now not necessary to introduce the tables in the text. From the Table caption, it should be clear that long-term means are meant.**

RC#2: 15. Page 16, line: 7-9: "Müller Schmied .. Table 2": Is this correct? It seems to me that the numbers are swapped. Moreover, could you think of an explanation why the homogeneous forcing performs worse than all forcings (although prob. no significant difference)?

*AR: Thanks, as Müller Schmied et al., 2015 is not yet available for the reader and to avoid confusion, we have deleted this sentence.*

**MM: We have deleted this sentence.**

RC#2: 16. Page 16, line: 22: "10.5%": Where does this number come from?

*AR: We calculated the percentage difference in this case as (max(GSWP3, WFD, WFDEI_hom) − min(GSWP3, WFD, WFDEI_hom)) / mean(GSWP3, WFD, WFDEI_hom) * 100 (same methodology as described in P16, l13).*

**MM: none, calculation method already provided in P16, l13.**

RC#2: 17. Page 17, line: 7: "of": should be "in"

*AR: Thanks!*

**MM: fixed**

RC#2: 18. Page 17, line: 8-14: This piece of text is a bit fuzzy. Starts with anthropogenic water use, then on to Q, finally back to WCa again. Please reformulate.

*AR: Thank you.*

**MM: We have reformulated this in the new section 3.1 and tried to avoid such fuzzy paragraphs by separating the variable into paragraphs.**

RC#2: 19. Page 17, line: 15: "different .. AET": Can you relate AET also with differences in T or Radiation? And how could this possibly influence your irrigation water demand estimates?

*AR: Sure, AET is also related to differences in T and radiation. T is relatively consistent among the forcings (Fig. 2 (new Fig. 3)). Radiation is for sure one large driver for AET. As global Q is well constrained by observations in calibration regions, and the water balance is closed, differences in meteorological forcings need to be balanced by the other water balance components. As P is obviously the most important variable of the water balance, differences in P are translated to AET differences.*

**MM: This sentence (and the following) reads now as: "As Q within the calibration region is forced to be nearly equal for all climate data sets, different values of P (as well as T and radiation) lead to large differences in aggregated AET (with higher absolute differences than the P differences, or 12.2%). In contrast, AET differs by only 8.8% (and lower absolute differences than the P differences) in non-calibrated regions (both numbers for all forcings, Table 5)."**

RC#2: 20. Page 18, line: 8: "leads" should be "lead"

*AR: Thanks.*

**MM: fixed**

RC#2: 21. Page 18, line: 24: "More likely" is not equal to "more impotant", please apply terminology consistently

*AR: Thanks.*

**MM: We have modified the "either more likely due to" to "caused mainly by the" at P18, l23, l24 of the initial manuscript; in addition we modified it at P11, l 20, l 21 in the initial manuscript.**

RC#2: 22. Page 19, line: 10-11: "This is.. 20th century": At first glance, this statement seems contradictory to the previous sentence. Please add some text about the changes towards T3 (1971-2000) to clarify this.

*AR: Thanks for pointing this out. We have written differences at P19 l16-21 but we would shortly introduce it here.*

**MM: We have modified this sentence to: "Anthropogenic impact increases in the time period 1941-1970 and 1971-2000, which is consistent with the …"**

RC#2: 23. Page 19, line: 14-15: "The fraction… later period". Had to read this sentence a few times before I understood what was meant. Please clarify what is meant with the later period and specify that with 1911-1940/1941-1970 is mean between/from period T1 to period T2.

*AR: Thanks, you interpreted it correctly, we need to be more specific.*

**MM: We have modified this sentence to: "Human water use and dam construction is the dominant driver for changes in long-term Q averages on 9-13% of land area for the time period 1911-1940 and 1941-1970, and increases to 11-18% of land area for the time period 1941-1970 and 1971-2000."**

RC#2: 24. Page 19, line: 16-21: I thought that the areas "which cannot be calculated"(l:19) and "where both, P and human water use is not the dominant driver"(l:20) are the same areas. But here they have different fractions associated.

*AR: The fraction "which cannot be calculated" refers the area were the denominator of indicators An and/or Bn are zero. In contrast, "where both, P and human water use is not the dominant driver" relates to the grid cells, where An and Bn are both negative. We have simplified the indicator approach and feel that this is now better described.*

**MM: The whole paragraph is restructured as follows: "Human water use and dam construction is the dominant driver for changes in long-term Q averages on 9-13% of land area for the time period 1911-1940 and 1941-1970, and increases to 11-18% of land area for the time period 1941-1970 and 1971-2000. The fraction with P domination increases, too, from 53–54% to 58–65%. At the same time, the area for which the indicators An and Bn cannot be calculated (due to similar long-term Q averages and thus zero in the denominators of Eqs. 3 and 4) is rather constant (1.1% to 0.9%). The land fractions where neither driver is dominant decreases slightly from 6% to 5%. Figs. B1 and B2 in Appendix B shows An and Figs. B3 and B4 shows Bn."**

RC#2: 25. Page 20, line: 2: "STANDARD": I haven't seen this model reference before whilst you refer to this specific version of model in earlier paragraphs. Please use consistent namings.

*AR: In earlier paragraphs we are referring to the whole set of model variants of MS14, and in this specific paragraph we refer to this specific model variant.*

**MM: none**

RC#2: 26. Page 20, line: 13-16: "In addition…in STANDARD". Incomplete sentence, please reformulate

*AR: Thanks.*

**MM: We would split up this sentence and modify it to: "In the applied WaterGAP 2.2 (ISIMIP 2.1) version, reservoirs are filled up with water in their construction year. This leads to a net increase of reservoir storage (53 km³ yr$^{-1}$) compared to a decrease of 43 km³ yr$^{-1}$ in STANDARD, where reservoirs are assumed to have been in operation over the entire simulation period. Thus, total water storage decreased less than in STANDARD, with 47 km³ yr$^{-1}$ instead of 215 km³ yr$^{-1}$."**

RC#2: 27. All of section 4.2 and many parts of 4.3/4.4 are results, no discussion. Please reshuffle.

*AR: Thanks.*

**MM: In the revised manuscript we have joined results and discussion.**

RC#2: 28. Page 21, line: 21-22: "the effect .. time aggregation". Was this specifically evaluated? If yes, where (I cannot find the associated results section)

*AR: Thanks for pointing this out. We based this evaluation on Fig. 2.*

**MM: We have revised this part.**

RC#2: 29. Page 23, line: 26: Would dam construction indeed lead to significant decreases in long-term Q? I would think that dam construction would mainly influence the timing of runoff peaks/lows. Could you give a reference for this statement?

*AR: Due to increased evaporation from the reservoir surface (which was not there before dam construction), a decrease in mean Q is expected. Döll et al. (2009) calculated the decrease of mean Q solely due to dam construction to be 0.8% globally. In conjunction with dam construction, water is used often e.g. for irrigation, which also reduces mean Q. Anyhow, the reviewer is right in that the effect on timing is more substantial.*

**MM: We have added the reference Döll et al. (2009) and revised the sentence to "While dam construction leading to new reservoirs decreases long-term average Q (e.g. due to additional evaporation), human water consumption is expected to be more important in most grid cells (see also Döll et al., 2009)".**

RC#2: 30. Page 24, line: 2:8: "For example…"anthropogenic effects": Do you expect that under the different climate forcings the absolute value of the anthropogenic impacts (mainly irrigation I would say) also changes or is this difference in outcome (relative contribution) only determined by changes in P)

*AR: Yes, irrigation water use depends on the climate forcings (all variables).*

**MM: We have added a sentence to P24, l2 (initial manuscript): "They lead to different changes of P and different changes of human water use as the globally dominant irrigation water use is computed as a function of climate."**

RC#2: 31. Page 25, line: 20-21: "e.g. ISI-MIP2.1". Could you mention more model intercomparison projects? Think of Agmip, Earth2Observe.

*AR: Thanks for this suggestion.*

**MM: we have included AgMIP, eartH2Observe and LS3MIP to the list of current model intercomparison projects.**

RC#3: This manuscript presents a quantitatively useful update on our ability to marry terrestrial models of water and energy balance & their uncertain physical processes and tuning parameters, with records of hydrometeorological forcing that are fragmentary and have significant time-dependent biases. These forcing biases result from an amalgam of diverse satellite and in situ data as well as inputs from atmospheric reanalyses.

*AR: Thank you for this very positive general statement.*

RC#3: The analysis presented here covers time scales from annual to century and spatial scales from 0.50 degree lat/lon grids to global land means. While the formalism here is a largely a standard approach there are two specific aspects I feel are noteworthy:

1) The first is the identification of how specific differences in forcing data propagate through the water balance and, by virtue of tuning / calibration on observed discharge affect gauged and ungauged contributions differently. Both the low precipitation in the PGFv2 data (as a result primarily of no snowfall undercatch correction) and the low values of downward SW radiation in the WATCH forcing (resulting from the older NCEP reanalysis values) are cases in point. In gauged regions, discharge (Q) is controlled by calibration and so AET responds to the differences in precipitation and radiative forcing. (Note the large spread in global AET in Table 3, focused in Europe and N. America where snowfall is significant.) Conversely, in ungauged regions the resulting uncertainty from forcing data sets in discharge is over 18% (Table 2). These results quantify the effects of forcing data quality and (un)availability on regional and global results.

2) The other noteworthy aspect of this paper is the attempt to identify the relative roles of climate forcing versus anthropogenic effects on changing water balance. Diagnostic indicators involving the relative changes in P versus Q and changes in the actual versus "naturalized" P and Q changes are considered. The resulting maps (Figs 3,4) are valuable, I think, not only for their consistency but also their differences between forcing data sets. One is not surprised to see the western US and Europe exhibit significant increases in diversions /extractions through the middle of the 20th Century. The growth of these impacts throughout southern Asia in the second half of the 20th Century is reasonably consistent across the four homogeneous data sets. Nevertheless there are significant regional differences in the estimated growth of anthropogenic effects in Australia, China and Mexico are

discussed as examples where the forcing data set differences have significant interpretive consequences.

So, basically, my sense is that this paper is a valuable assessment of where our diagnostic modeling capabilities for water balance stand.

*AR: We are grateful for this assessment of the paper, thanks for this comprehensive summary and your thoughts about it. During revising of the manuscript / abstract we have included some of the main messages. Thanks again.*

RC#3: That said, there are some aspects of the presentation that need improvement:

i. The discussion in section 2.4.2 on the construction of the indicators for anthropogenic effects was difficult to follow. The reasoning behind An and Bn seems clear enough, but I had a difficult time trying to understand how An and Bn were incorporated in Figs 3 and 4. What combinations of An >0, Bn<0 and Ivarpredoc,n >0 make up red or blue areas in those Figs? Presumably An and Bn are of opposite sign in both blue and read areas? Perhaps some schematic picture would be useful.

*AR: Thank you for pointing this out.*

**MM: We have completely revised the indicator approach as we found out that Ivarant can be reduced to the difference of the naturalized Q thus this approach is not useful for anthropogenic impacts. We have now simplified the indicators, and modified the numbers / figures in the manuscript. We also have revised the description of the indicators for a better readability and have integrated the indicator criteria to the legend of Figs. 3 and 4 (now 4 and 5). In addition, we have included the An and Bn figure in Appendix B as well as added a new Table 1 with some calculation examples.**

RC#3: ii. There were numerous places in the manuscript where I was unsure as to what Figures or Tables the discussion related to. Does section 4.2 refer to the information in Fig 2? Does the discussion in section 4.3 relate to Table 2 and 3? Does section 4.4 refer to Figures 3 and 4? Alluding to the appropriate graphic needs to be added.

*AR: Thank you.*

**MM: Based on the comments of referee #2 we have joined results and discussion and emphasized for integrating related Tables or Figures where required. Even if some of the tables are referenced in different sections, we hope it is now better structured.**

RC#3: iii. I don't really see a lot of value of section 4.1. It largely discusses differences with the standard version WaterGAP 2.2. Perhaps mentioning earlier estimates from papers such as Oki and Kanae (2006), Haddeland et al, (2011) and Rodell et al, (2015) would give some context outside the WaterGAP model.

*AR: Thanks. In the paper Müller Schmied et al. (2014), Table 5, we compared WaterGAP output to many other global estimates and referred to it here because we do not want to simply reproduce this table. Anyhow, as we missed the global assessment of Wisser et al. (2010), Hanasaki et al. (2010) and there are some new numbers available (Rodell et al., 2015 as you mentioned), so we made new analyses and revised this part of the manuscript*

**MM: We have described it now more clearer in (new) Sect. 3.1, when we are comparing to earlier WaterGAP results and to other studies. We have completely revised the previous section 4.1 and added the numbers of Wisser et al. (2010), Hanasaki et al. (2010) and Rodell et al. (2015) in a new table 6. For this comparison, we analyzed WaterGAP outputs for same time span and same spatial extent as it is described in the references. We feel that this part of Sect. 3.1 is now more of value.**

RC#3:I recommend accpetance after attention to these these three areas of concern

*AR: Thank you again for your constructive assessment.*

References

Beck, C., Grieser, J. and Rudolf, B.: A new monthly precipitation climatology for the global land areas for the period 1951 to 2000, German Weather Service, Climate Status Report KSB 2004, Offenbach, Germany., pp. 181–190, 2005.

Biemans, H., Hutjes, R. W. A., Kabat, P., Strengers, B. J., Gerten, D. and Rost, S.: Effects of precipitation uncertainty on discharge calculations for main river basins, J. Hydrometeorol., 10(4), 1011–1025, doi:10.1175/2008JHM1067.1, 2009.

Döll, P., Fiedler, K. and Zhang, J.: Global-scale analysis of river flow alterations due to water withdrawals and reservoirs, Hydrol. Earth Syst. Sci., 13(12), 2413–2432, doi:10.5194/hess-13-2413-2009, 2009.

Fuchs, T., Rapp, J., Rubel, F. and Rudolf, B.: Correction of synoptic precipitation observations due to systematic measuring errors with special regard to precipitation phases, Phys. Chem. Earth, Part B Hydrol. Ocean. Atmos., 26(9), 689–693, doi:10.1016/S1464-1909(01)00070-3, 2001.

Hanasaki, N., Inuzuka, T., Kanae, S. and Oki, T.: An estimation of global virtual water flow and sources of water withdrawal for major crops and livestock products using a global hydrological model, J. Hydrol., 384(3-4), 232–244, doi:10.1016/j.jhydrol.2009.09.028, 2010.

Klein Goldewijk, K., Beusen, A., Van Drecht, G. and De Vos, M.: The HYDE 3.1 spatially explicit database of human-induced global land-use change over the past 12,000 years, Glob. Ecol. Biogeogr., 20(1), 73–86, doi:10.1111/j.1466-8238.2010.00587.x, 2011.

Müller Schmied, H., Eisner, S., Franz, D., Wattenbach, M., Portmann, F. T., Flörke, M. and Döll, P.: Sensitivity of simulated global-scale freshwater fluxes and storages to input data, hydrological model structure, human water use and calibration, Hydrol. Earth Syst. Sci., 18(9), 3511–3538, doi:10.5194/hess-18-3511-2014, 2014.

Rodell, M., Beaudoing, H. K., L'Ecuyer, T. S., Olson, W. S., Famiglietti, J. S., Houser, P. R., Adler, R., Bosilovich, M. G., Clayson, C. A., Chambers, D., Clark, E., Fetzer, E. J., Gao, X., Gu, G., Hilburn, K., Huffman, G. J., Lettenmaier, D. P., Liu, W. T., Robertson, F. R., Schlosser, C. A., Sheffield, J. and Wood, E. F.: The Observed State of the Water Cycle in the Early 21st Century, J. Clim., 150904104833007, doi:10.1175/JCLI-D-14-00555.1, 2015.

Schneider, U., Becker, A., Finger, P., Meyer-Christoffer, A., Rudolf, B. and Ziese, M.: GPCC Full Data Reanalysis Version 7.0 at 0.5°: Monthly land-surface precipitation from rain-gauges built on GTS-based and historic data, doi:10.5676/DWD_GPCC/FC_M_V7_050, 2015.

Stocker, T. F., Qin, D., Plattner, G.-K., Alexander, L. V., Allen, S. K., Bindoff, N. L., Bréon, F.-M., Church, J. A., Cubasch, U., Emori, S., Forster, P., Friedlingstein, P., Gillett, N., Gregory, J. M., Hartmann, D. L., Jansen, E., Kirtman, B., Knutti, R., Krishna Kumar, K., Lemke, Pl, Marotzke, J., Masson-Delmotte, V., Meehl, G. A., Mokhov, I. I., Piao, S., Ramaswamy, V., Randall, D., Rhein, M., Rojas, M., Sabine, C., Shindell, D., Talley, L. D., Vaughan, D. G. and Xie, S.-P.: Technical Summary, in: Climate Change 2013: The Physical Science Basis. Contribution of Working Group I to the Fifth Assessment Report of the Intergovernmental Panel on Climate Change, Stocker, T.F., Qin, D., Plattner, G.-K., Tignor, M., Allen, S. K., Boschung, J., Nauels, A., Xia, Y., Bex, V. and Midgley P.M. (eds.). Cambridge University Press, Cambridge, United Kingdom and New York, NY, USA., 84 pp., 2013.

Voisin, N., Wood, A. W. and Lettenmaier, D. P.: Evaluation of Precipitation Products for Global Hydrological Prediction, J. Hydrometeorol., 9(3), 388–407, doi:10.1175/2007JHM938.1, 2008.

Wisser, D., Fekete, B. M., Vörösmarty, C. J. and Schumann, A. H.: Reconstructing 20th century global hydrography: a contribution to the Global Terrestrial Network- Hydrology (GTN-H), Hydrol. Earth Syst. Sci., 14(1), 1–24, doi:10.5194/hess-14-1-2010, 2010.

**List of relevant changes of the manuscript**

- We have joined (and partly rewritten) the results and discussion section (new Sect. 3) in order to reduce overlap (criticized esp. by referee#2) and to improve general manuscript structure.
- We modified the order of the first two research questions and subsequent re-organized the following subsections.
- We updated Fig. 5 (in the revised manuscript Fig. 3) and now show discharge instead of runoff.
- We updated PGFv2 forcing to PGFv2.1 as during review process, errors in temperature between 1901-1947 were reported (and corrected) by the data provider.
- We now include in Fig. 2 global assessments of net radiation and potential evapotranspiration as this enables us to better explain differences in the water balance components among the model variants.
- We have strongly simplified the indicator in Sect. 2.4.2 (research question 3) as well as updated the figures and show the single indicators An and Bn as figures in the appendix.
- We have enhanced the section that compares our results with previous estimates by adding a new Table (6) and the description in the text.
- We have revised the abstract to be more focused on the study results.

**Manuscript in track-change view**

We have carefully revised several parts of the manuscript in order to improve structure, the presentation and language. Furthermore, major changes were made due to the joining of the results and discussion section as well as due to the simplification of the indicators. In correspondence with the Publishing Service of Copernicus, we came to the decision that it would not be of benefit for the reader to provide a "track-change" version of the manuscript.